



# CoupModel (v6.0): an ecosystem model for coupled phosphorus, nitrogen and carbon dynamics – evaluated against empirical data from a climatic and fertility gradient in Sweden

Hongxing He[1], Per-Erik Jansson[2], Annemieke Gärdenäs[1]

**1**Department of Biological and Environmental Sciences, University of Gothenburg, Po Box 460, Gothenburg 40530, Sweden

**2**Department of Land and Water Resources Engineering, Royal Institute of Technology

(KTH), 100 44 Stockholm, Sweden

*Correspondence to:* Hongxing He (hongxing-he@hotmail.com); and Annemieke Gärdenäs (annemieke.gardenas@bioenv.gu.se)

**Abstract**

This study presents the integration of the phosphorus (P) cycle into CoupModel (Coup-CNP). The extended Coup-CNP enables simulations of coupled carbon (C), nitrogen (N) and P dynamics for terrestrial ecosystems which explicitly consider mycorrhizal interactions. The model was evaluated against observed forest growth and measured leaf C/P, C/N and N/P

ratios in four managed forest regions in Sweden. The four regions form a climatic and fertility gradient from 64°N in the North to 56°N in South Sweden with the mean annual temperature varying between 0.7-7.1 °C and the soil C/N and C/P ratios between 19.8-31.5 and 425-633, respectively. The growth of the southern forests was found to be P-limited, with harvested biomass representing the largest P loss over the studied rotation period. The

simulated P budgets revealed that southern forests are losing P while northern forests are close to a steady state in P availability. Mycorrhizal fungi account for half of the total plant P uptake across all four regions, which highlights the importance of fungal-tree interactions in Swedish forests. Sensitivity analysis results demonstrated that the highest forest growth occurs at a soil N/P ratio of 15 to 20. A soil N/P ratio above 15-20 resulted in decreased soil

C sequestration and total P leaching, but significantly increased N leaching. The development and evaluation of the new Coup-CNP model demonstrate that P fluxes need to be further



considered in studies of how climate change will influence C turnover and ecosystem responses. We conclude that the potential P-limitation of terrestrial ecosystems highlights the need of a proper consideration of the P cycle in biogeochemical models. The inclusion of the

P cycle is necessary in order to make models reliable tools for assessing long-term impacts of climate change and N deposition on C sequestration and N leaching.

## 1 Introduction

Phosphorus (P) is an essential element for all life, with the P cycle coupled to Carbon (C) and

Nitrogen (N) fluxes through biochemical reactions such as photosynthesis and nutrient cycling in terrestrial ecosystems (Lang et al., 2016; Vitousek et al., 2010). A steep increase in the anthropogenic release of C and N to the atmosphere relative to P release has altered plant and soil nutrient stoichiometry, leading to new forcing conditions (Elser et al., 2007; Penuelas et al., 2013). For instance, numerous monitoring studies have revealed increasing N/P ratios

in plants and soils, especially in forests from North America (Crowley et al., 2012; Gress et al., 2007; Tessier and Raynal, 2003) and Central and Northern Europe (Braun et al., 2010; Jonard et al., 2015; Talkner et al., 2015). Such trends are generally assumed to indicate that these ecosystems are shifting from being N limited to either co-limited by both N and P or P limited (Elser et al., 2007; Saito et al., 2008; Vitousek et al., 2010). Human activities are

expected to continue increasing the N inputs to the atmosphere and, as such, P availability and its dynamics will become progressively more important to regulating the biogeochemistry of terrestrial ecosystems and amplifying feedback mechanisms relevant to climate change, e.g. limiting the growth of plants (Deng et al., 2017; Fleischer et al., 2019; Goll et al., 2017).

Nevertheless, the P cycle is seldom incorporated into ecosystem model structures (Flato et al., 2013; Reed et al., 2015). The few models that include a P module vary greatly with respect to scale, scope and ecosystem type. Most of the process-based models that can simulate P cycling were specifically developed for agricultural systems and focus on the soil ecosystem, e.g., EPIC (Jones et al., 1984), ANIMO (Groenendijk and Kroes, 1999), and GLEAMS

(Knisel and Turtola, 2000). A few catchment-scale models that focus on surface water, e.g. SWAT (Arnold et al., 2012), HYPE (Arheimer et al., 2012), and INCA-P (Jackson-Blake et al., 2016), aim to simulate how crop management influences P leaching and thus, consider processes such as nutrient retention, leaching, and transport. However, none of these models explicitly consider plant litter inputs, nutrient mineralization, or how nutrient uptake

mechanisms influence photosynthesis. Until recently, global vegetation models that were





integrated into Earth System Models (ESMs) largely ignored carbon-phosphorus cycle interactions, e.g. present Coupled Model Intercomparison Project (CMIP 5) ESMs do not include the P cycle (Flato et al., 2013). However, the C response to P limitation has recently been studied through several empirical and field studies. For example, Van Sundert et al.
(2019) showed that there is a strong correlation between the productivity of European beech forests and soil organic and mineral C/P ratios. A few global vegetation models have included a simplified P cycle to study how it affects the C cycle (Goll et al., 2012; Wang et al., 2010; Yang et al., 2014). However, these P enabled models differ in how they describe soil P dynamics, plant P use and acquisition strategies, which results in considerable
uncertainty in the C response (Fleischer et al., 2019; Medlyn et al., 2016; Reed et al., 2015; Zaehle et al., 2014). Medlyn et al. (2016) applied six ESMs including two coupled Carbon-Nitrogen-Phosphorus (CNP) models (CABLE and CLM4.0-CNP) to study the response to elevated $CO_2$ (e$CO_2$) of the C cycle of the Eucalyptus-Free-Air $CO_2$ Enrichment experiment and found large variations, ranging from 0.5 to 25%, in predicted net primary productivity.
The CNP models that considered how P can limit photosynthesis predicted the lowest e$CO_2$ response. The P cycle is assumed to be relatively closed, meaning that the input (i.e. deposition and weathering) and output (i.e. leaching) fluxes are small. In other words, the vegetation is rather inflexible to increase P uptake. Yu et al. (2018) developed the field-scale biogeochemical model – ForSAFE and applied the model to study the P budget of a southern
Swedish Spruce forest site. They concluded that the total P flow into the forest was small compared to the internal turnover. Plant P uptake was attributed more to mineralization of soil organic matter than the weathering of minerals. Fleischer et al. (2019) demonstrated that four CNP models, when applied to the Amazon forest, provide up to 50% lower estimates of the e$CO_2$-induced biomass increment than the 10 coupled C-N models. They attributed these
variations to the contrasting plant P use and acquisition strategies considered in the models, and suggested that the inclusion of flexible tissue stoichiometry and enhanced plant P acquisition could improve C-P cycle coupling in terrestrial ecosystem models.

Most terrestrial plants live in symbiosis with mycorrhizal fungi to increase uptake capacity of P, among other nutrients (Smith and Read, 2008). Several studies have shown that the
depletion zone around plant roots, which is caused by plant uptake and the immobile nature of mineral P, increases when a plant interacts with mycorrhizal fungi (Bolan, 1991; Schnepf and Roose, 2006; Smith, 2003). It is also well known that mycorrhizal fungi acquire P from organic sources that are not directly available to the plant roots, e.g., phytic acid and nucleic acids (Schachtman et al., 1998). Global meta-analysis studies highlighted that interaction



with soil mycorrhizal fungi strongly influences how P availability affects plant growth (Terrer et al., 2016; Terrer et al., 2019). Previous research showed that mycorrhizal fungi can receive between 1 and 25% of the plants' photosynthates and constitute as much as 70% of the total soil microbial biomass; this symbiont thus has a major impact on soil C sequestration (Averill et al., 2014; Clemmensen et al., 2013; Staddon et al., 2003). Even

though there is a well-established link between mycorrhizal fungi and plant P nutrition (Bucher, 2007; Read and Perez-Moreno, 2003; Rosling et al., 2016), this factor is seldom included in ecosystem models (Smith and Read, 2008). To the best of our knowledge, only Orwin et al. (2011) have presented a model of how mycorrhizal fungi influence soil C dynamics that considers both N and P. They found that considering organic nutrient uptake

by mycorrhizal fungi in an ecosystem model can significantly increase soil C storage, with this effect more pronounced under nutrient-limited conditions. However, in this model, plant growth was static; thus plant-soil or plant-environment interactions were largely ignored. He et al. (2018) integrated the MYCOFON model (Meyer et al., 2009) into CoupModel v5 to ensure that mycorrhizal interactions would be sufficiently considered, and compared the

results with a previous implicit representation of N uptake in forest ecosystems with limited N availability. CoupModel v5 assumes that carbohydrates provided by plants are the primary driver of mycorrhizal responses to N availability, and that fungal uptake of N will influence host plant photosynthesis. We argued that terrestrial ecosystem models which explicitly consider mycorrhizal interactions should also take into account P cycling, especially due to

the significant role of symbiont mycorrhiza for P uptake in P-limited environments and thus, developed a new version of CoupModel that includes the P cycle.

     The overall objective of this study was to improve the current understanding of C, N and P cycle interactions in forest ecosystems by presenting a new scheme for modelling P dynamics. More specifically, the study had the following aims: 1) to present the new

CoupModel v6.0 that explicitly includes the P cycle and interactions between the N and P cycles; 2) to estimate the regional C, N, and P budgets of Swedish forests along a climatic and fertility gradient; and 3) to demonstrate how soil N and P availability influence growth, soil C and leaching. For this reason, we present a new version of CoupModel (v6.0), hereafter referred to as Coup-CNP, which explicitly simulates the P cycle. The key features of the new

Coup-CNP model are: 1) coupled C, N, and P dynamics; 2) explicit and implicit plant-mycorrhizal representations to describe N and P uptake from the soil; 3) flexible CNP stoichiometry for plant components, soil organics, and mycorrhizal fungi; 4) dynamic nutrient demand and uptake, photosynthesis and growth rates regulated by N and P



availability; and 5) simultaneous uptake of nutrients to roots or mycorrhizae from both
organic and inorganic pools. The Coup-CNP model was evaluated using four forest regions
situated along a climatic and fertility gradient in Sweden that has earlier been considered by
He et al. (2018) and Svensson et al. (2008).

## 2 Model structure and description of processes linked to the phosphorus pool

**2.1 Brief description of CoupModel (v5)**

The CoupModel platform (coupled heat and mass transfer model for soil–plant–atmosphere
systems) is a flexible process-based model designed to simulate water and heat fluxes, along
with C and N cycles, in terrestrial ecosystems (Jansson, 2012). The main model structure is a
one-dimensional, vertical layered soil profile including plants. The core of the model consists
of five sets of coupled partial differential equations, one for each of water, heat, C, N, and P
fluxes (the later one in v6.0). They are numerically solved using an explicit forward
difference model scheme (Euler integration). In this application, we used a default common
time step for all five, but a smaller time step was applied for the water and heat calculations
during specific events crucial to ensure the numerical stability and accuracy. The model is
driven by climatic data – precipitation, air temperature, relative humidity, wind speed, and
global radiation – and can simulate ecosystem dynamics with daily resolution. Vegetation is
described using the "big leaf" concept, i.e. two vegetation layers, trees and understory plants,
are simulated taking into account mutual competition for light interception, water uptake and
soil N (Jansson and Karlberg, 2011). The model is freely available at www.coupmodel.com.
A general structural and technical overview of the CoupModel can be found in Jansson and
Karlberg (2011) and Jansson (2012). He et al. (2018) introduced an explicit plant-mycorrhizal
representation scheme into CoupModel v5.

**2.2 Phosphorus cycle representation in CoupModel (v6.0)**

This study describes the inclusion of the P cycle in the CoupModel, resulting in v6.0. We
developed the P model in a way that 1) concentrates on P processes that are most relevant for
biogeochemical assessments, e.g., dynamic plant growth and P leaching, and 2) follows the
conceptual structure of CoupModel as closely as possible. The P model runs at the same time
step as the models for C and N cycles, which can differ from the time step of the models for
water and heat. The discretization of the soil includes common compartments that are linked





to all elements and abiotic conditions. For simplicity, the following description of the model concerns one of the layers that represent the entire soil profile.

### 2.2.1 Inorganic and organic soil phosphorus

Soil phosphorus can be divided into inorganic P ($P_i$, phosphate ions, e.g., $H_3PO_4$, $H_2PO_4^-$, $HPO_4^{2-}$, $PO_4^{3-}$), soil mineral P ($P_m$) and soil organic P ($P_o$, P that is bound to organic C). Overall, three inorganic P pools (soluble ($P_{isol}$), labile ($P_{ilab}$), and soil mineral ($P_m$)) and three organic P pools (litter ($P_{olit}$), humus ($P_{ohum}$), and dissolved organic ($P_{dop}$)) are used to represent the soil P (Fig. 1). In Coup-CNP, the soil inorganic-phosphorus pools are defined

according to the classic Hedley fractionation method (Hedley and Steward, 1982), in which labile $P_{ilab}$ determines the available $P_i$ for plants including the readily $P_i$ exchanges with soil solutions, $P_{isol}$. The soil mineral $P_m$ is defined as the total soil P without organic $P_o$ and labile $P_{ilab}$ (Hedley and Stewart, 1982). Thus, soil mineral $P_m$ is a lumped pool containing primary and secondary mineral P (and occluded P) (Smeck, 1985; Wang et al., 2007), and can be

estimated using total P content and bulk density data. The three organic P pools follow the division of the C and N cycles. Soil litter consists of fresh plant residues and non-symbiotic microbes while humus is the organic residue from litter decomposition. In CoupModel, soil litter could be further divided into two litter pools: one which contains readily decomposing materials (e.g., plant leaves and fine roots) and another for decomposition-resistant litter (e.g.,

stems and coarse roots). For simplicity, only one soil litter pool was used in this study.

Soil mineral $P_m$, which mostly exists in the form of apatite, may be transformed into labile $P_{ilab}$ through the weathering process (Fig. 1). Part of the $P_{ilab}$ pool would quickly be adsorbed by soil water and colloids (Buendía et al., 2010; Stewart and Tiessen, 1987). These $P_i$ ions are normally loosely adsorbed to surfaces and can thus easily re-enter the $P_{ilab}$ pool through

the desorption process (McGechan and Lewis, 2002). The adsorption/desorption of $P_i$ in the $P_{ilab}$ pool is thus modelled as a continuum of fast reversible exchange reactions of the phosphate groups with water or hydroxyl groups (Bünemann, 2015). Both Cole et al. (1977) and Olander and Vitousek (2005) showed that when P is added to a soil ecosystem, the soluble ($P_{isol}$) and labile ($P_{ilab}$) P pools reach equilibrium in less than 1 hour. As the

CoupModel provides daily resolution, we can assume that the $P_{isol}$ and $P_{ilab}$ pools are always in equilibrium (Barrow, 1979). In our model, the slow $P_i$ diffusion process into the soil matrix (e.g. adsorption/desorption to the secondary minerals) is implicitly included in the weathering flux.



If mineral $P_i$ fertilizer is applied at the soil surface, the $P_i$ first enters an undissolved fertilizer
pool, after which $P_i$ from this pool gradually dissolves into the labile P pool following a
decay-type function. P could also be added as an external organic substrate (faeces or
manure). In this case, P moves from the surface faeces ($P_{ofae}$), litter ($P_{olit}$), and labile ($P_{ilab}$) P
pools according to the composition of the manure. Atmospheric P deposition is assumed to
directly flow to the $P_{ilab}$ pool in the uppermost soil layer. $P_i$ in the $P_{isol}$ pool can be
transported by water flows between layers or from a layer to a drainage outlet. The soil
surface layer may also lose mineral $P_m$ by erosion, which is driven by surface runoff.

### 2.2.2 Phosphorus fluxes in plant and symbiotic fungi

Plants take up $P_i$ from the $P_{ilab}$ pool through roots and symbiotic microbe association and then
partition the $P_i$ into grain, leaf, stem, coarse root, fine root and symbiotic microbe pools (Fig.
1). In this paper, we use mycorrhizal fungi as the role model of plant-microbe symbiosis, the
same concept is also applicable for other symbiosis microbes. Through plant litterfall, P is
recycled and released back into the soil through mineralization. During certain seasons, plants
can also capture mobile P (as well as mobile N) to prepare for rapid growth in the spring.
This will increase the litter C/P and C/N ratios, respectively. In the Coup-CNP, P
mineralization is conceptually divided into biological and biochemical mineralization
following McGill and Cole (1981). Biological mineralization, which is regulated by
temperature and moisture, represents microbe-mediated oxidation of organic matter, during
which nutrients (P and N) are immobilized by implicit non-symbiotic microbes or transferred
from litter to humus (Fig. 1). Biochemical mineralization, on the other hand, describes the
release of $P_i$ through extracellular enzymatic processes (e.g., phosphatases), which are driven
by plant demand for nutrients (Richardson and Simpson, 2011). In Coup-CNP, biochemical
mineralization is defined as organic uptake and assumed to be driven by the unfulfilled plant
P demand after $P_{ilab}$ root uptake. The assumption is that plant roots and symbiotic fungi
bypass the $P_{ilab}$ pool, and obtain mineralized $P_i$ directly from the $P_{olit}$ and $P_{ohum}$ pools (Fig. 1).
A fraction of the $P_o$ in the $P_{olit}$ and $P_{ohum}$ pools may also form $P_{dop}$. This reaction has the same
kinetics as having been observed for the C and N cycles (DOC, DON). The dissolved
organics are characterized by a mobile and an immobile fraction, and thus, can be
redistributed between layers. In addition to harvest, P could leave the ecosystem through
leaching of $P_{isol}$ and DOP (Fig. 1).

### 3 Equations describing key phosphorus processes/fluxes and their parameterization



The following section outlines the processes that are specific to the P cycle. Processes that are analogous to those of the N cycle, e.g., atmospheric deposition, fertilization, mineralization-

immobilization, plant growth and uptake, litterfall, leaching and surface runoff and removal of plant harvest, are detailed described in Appendix A.

### 3.1 Weathering

During weathering, $P_i$ is released from minerals through the dissolution of apatite

($Ca_5(PO_4)_3(F/Cl/OH)$) depending on the soil acidity. Organic acid exudates from plant roots also contribute to the release of $P_i$ (Schlesinger, 1997). In our model, the weathering process is modelled as a dissolution reaction (Brantley et al., 2008; Lasaga, 1998) in which the reaction rate at a given temperature is proportional to the pH of the soil water solution (Guidry and Machenzie, 2000). The weathering rate is calculated as,

$$P_{m \to ilab} = k_w \times f_w(T) \times f_w(pH) \times P_m \qquad (1)$$

Where $P_{m \to ilab}$ is the flux rate of weathering (g P m$^{-2}$ day$^{-1}$), $k_w$ is a first-order integrated weathering rate coefficient (day$^{-1}$) which depends on lithology, rates of physical erosion and soil properties, $f_w(T)$ and $f_w(pH)$ are response functions of soil temperature and soil $pH$, $P_m$ is the size of the $P_m$ pool (g P m$^{-2}$), determined by,

$$P_m = \delta_P \times \rho_{bulk} \times \Delta z_{layer} \times 10^6 \qquad (2)$$

Where $\delta_P$ is the prescribed $P_m$ content for each soil layer (g P g dry soil$^{-1}$), with reported ranges from 0.1 to $1.5 \times 10^{-3}$ g P g soil$^{-1}$ (Yang et al., 2014), $\rho_{bulk}$ is the dry bulk density for each soil layer (g cm$^{-3}$), and $\Delta z_{layer}$ is the thickness of the simulated soil layer (m).

The temperature effect could be expressed as an Arrhenius function (3), where $E_{a,wea}$ is the

activation energy parameter (J mol$^{-1}$) for minerals (i.e., apatite), available from empirical studies, $R$ is the gas constant (J K$^{-1}$ mol$^{-1}$), $T_s$ is the simulated soil temperature in °C, $T_{s,0}$ is a parameter (°C) which normalize the function $f_w(T)=1$ and $T_{abszero}$ is -273.15 °C.

$$f_w(T) = e^{\left( -\frac{E_{a,wea}}{R} \times \left( \frac{1}{T_s + T_{abszero}} - \frac{1}{T_{s,0} + T_{abszero}} \right) \right)} \qquad (3)$$

Alternatively, the existing Ratkowsky function, O'Neill function or Q$_{10}$ method can be used

to determine the temperature response in CoupModel.

The effect of soil pH on weathering can be calculated through (4), where $n_H$ is a parameter that describes the sensitivity soil pH when it differs from an optimal value $pH_{opt}$ for weathering.



$$f_w(pH) = 10^{n_H \times |pH_{opt} - pH|} \tag{4}$$

### 3.2 Inorganic soluble phosphorus dynamics

The sizes of the $P_{isol}$ and $P_{ilab}$ pools are largely determined by chemical soil properties, e.g., anion exchange capacity and pH. The dynamics of these pools are regulated by physiochemical, e.g., adsorption/desorption, as well as biochemical processes, e.g., mineralization/immobilization. The relationship between the $P_{isol}$ and $P_{ilab}$ pools is normally represented by empirical equations, i.e., Freundlich and Langmuir isotherms (McGechan and Lewis, 2002). In this study, the modified Langmuir isotherm (Barrow, 1979) was used to model the fast and reversible sorption process.

$$P_{ilab,con} = p_{max,ads} \times \frac{P_{isol}}{c_{50,ads} + P_{isol}} \tag{5}$$

Where $P_{ilab,con}$ is the concentration of labile pool (g P g soil$^{-1}$) calculated by using equation (2) with $P_{ilab}$ (g P m$^{-2}$), $\rho_{bulk}$ (g cm$^{-3}$) and $\Delta z_{layer}$ (m), $p_{max,ads}$ is the maximum sorption capacity of the labile pool (g P g soil$^{-1}$), and $c_{50,ads}$ is an empirical parameter corresponding to 50% of saturation (g P m$^{-2}$).

### 3.3 Plant growth under phosphorus and nitrogen stress

Plant photosynthesis is modelled by a "light use efficiency" approach (Monteith, 1965). We adopted Liebig's law of minimum to simulate the effects of multiple nutrient stress on plant growth (Liebig, 1840). This approach assumes that the nutrient (N, P) which has a smaller supply relative to the corresponding plant demand will limit growth. Plant demand was estimated through defined optimum ratios.

$$C_{a \rightarrow plant} = \varepsilon_L \times f(T_{leaf}) \times f(nutrient) \times f(\frac{E_{ta}}{E_{tp}}) \times R_S \tag{6}$$

$$f(nutrient) = min\left(f(C/N_{leaf}); f(C/P_{leaf})\right) \tag{7}$$

Where $C_{a \rightarrow plant}$ is the plant carbon assimilation rate (g C m$^{-2}$ day$^{-1}$), $\varepsilon_L$ is the coefficient for radiation use efficiency (g C J$^{-1}$), $f(T_{leaf})$, $f(nutrient)$ and $f(E_{ta}/E_{tp})$ are response functions of leaf temperature, leaf nutrient status (N$_{leaf}$, P$_{leaf}$) in proportion to its C content, and water, respectively, and $R_s$ represents radiation absorbed by the canopy (J m$^{-2}$ day$^{-1}$). Details





concerning $f(T_{leaf})$, $f(E_{ta}/E_{tp})$, as well as growth and maintenance respiration, can be found in Jansson and Karlberg (2011). The nutrient response function *f(nutrient)* which includes P is described below.

As is the case with N, the photosynthesis process responds to the leaf C/P ratio was modelled

according to the work of Ingestad and Ågren (1992). Hence, below an optimum C/P ratio ($p_{CP,opt}$) the photosynthesis is not limited by P, between $p_{CP,opt}$ and $p_{CP,th}$ the response decrease as a linear function from one to zero,

$$f(C/P_{leaf}) = \begin{cases} 1 & C/P_{leaf} < p_{CP,opt} \\ 1+(\dfrac{C/P_{leaf} - p_{CP,opt}}{p_{CP,opt} - p_{CP,th}}) & p_{CP,th} \leq C/P_{leaf} \geq p_{CP,opt} \\ 0 & C/P_{leaf} > p_{CP,th} \end{cases} \qquad (8)$$

Where C/P$_{leaf}$ is the actual leaf C/P ratio and $p_{CP,opt}$ and $p_{CP,th}$ are parameters that vary

between plant species.

### 3.4 Symbiotic mycorrhizal fungi growth and phosphorus dynamics

The mycorrhiza describes a symbiotic association between fungus and the plants' fine roots: as such, it consists of C, N, and P pools that are separate from those of the plant. The

mycorrhiza is further distinguished into the mycelia, which is responsible for N and P uptake (both in inorganic and organic forms), and the fungal mantle, which covers the fine-root tips (He et al., 2018).

The following described the fungal processes that are specific to P (P processes analog to N processes (He et al., 2018) are found in appendix A). Plant C allocation to mycorrhizal fungi

is influenced by soil $P_i$ concentrations. We thus introduce a response function $f_{a \to fungi}(P_i)$ to account for reductions in plant C allocation to mycorrhizal fungi when soil $P_i$ concentrations are high analog to the N response function in He et al. (2018),

$$f_{a \to fungi}(P_i) = e^{(-P_{avail} \times P_{isol}{}^2)^3} \qquad (9)$$

Where $P_{isol}$ is the total soluble $P_i$ in the soil (g P m$^{-2}$), and $p_{avail}$ is a reduction parameter (m$^4$ g$^{-2}$

P). According to Bahr et al., (2015), mycorrhizal fungi biomass decrease already when either N or P wais added, but most significantly when both N and P were added. These multiple responses were integrated into the model so that potential fungal growth would decline as a result of either increasing soil N or P.

$$C_{a \to fungi,max} = C_{a \to root} \times p_{fmax} \times (f_{a \to fungi}(P_i) \times f_{a \to fungi}(N)) \qquad (10)$$



Where $C_{a\rightarrow fungi,max}$ is the defined maximum C flow that plants allocate to fungi (g C m$^{-2}$ day$^{-1}$), $C_{a\rightarrow root}$ is the total C allocated to both root and mycorrhiza (g C m$^{-2}$ day$^{-1}$), $p_{fmax}$ is a parameter that defines the maximum C fraction allocated to mycorrhiza from the total root and mycorrhiza C pool, and $f_{a\rightarrow fungi}(P_i)$ and $f_{a\rightarrow fungi}(N)$ are response functions which regulate maximum mycorrhizal fungi growth due to soil N and P availability.

The actual growth of mycorrhizal fungi, $C_{a\rightarrow fungi}$ (g C m$^{-2}$ day$^{-1}$), is limited by the defined maximum growth, $C_{a\rightarrow fungi,max}$ calculated as,

$$C_{a\rightarrow fungi} = \min\left\{\left[((C_{root} \times p_{fopt}) - C_{fungi}) \times \min(f(N_{supply}); f(P_{supply}))\right]; C_{a\rightarrow fungi,max}\right\} \quad (11)$$

Where $C_{root}$ is the total root C content (g C m$^{-2}$), $p_{fopt}$ is the defined optimum ratio parameter between fungi and root C content, $C_{fungi}$ is the total C content of fungi (g C m$^{-2}$), and $f(N_{supply})$

and $f(P_{supply})$ are response functions of fungal growth to the amount of N and P (both mineral and organic N) which are transferred from fungi to plant. In this way, mycorrhizal fungi growth is also influenced by how efficiently the fungi transfer nutrients to the host plant. The model follows the assumption that plants provide fungi with C as long as their investment is outweighed by the benefits (i.e., acquired N or P) (Nasholm et al., 2013; Nehls, 2008). We

further assume the C investment will be limited by the minimum nutrient supply efficiency provided by fungi. $f(P_{supply})$ is calculated as,

$$f(P_{supply}) = \begin{cases} 1 & P_{fungi\rightarrow plant,th} \leq P_{fungi\rightarrow plant} \\ \dfrac{P_{fungi\rightarrow plant}}{P_{fungi\rightarrow plant} + P_{ilab\rightarrow root}} & P_{fungi\rightarrow plant,th} > P_{fungi\rightarrow plant} \end{cases} \quad (12)$$

$$P_{fungi\rightarrow plant,th} = p_{fth} \times (P_{fungi\rightarrow plant} + P_{ilab\rightarrow root}) \quad (13)$$

Where $P_{fungi\rightarrow plant,th}$ is the defined threshold rate of fungal P supply (g P m$^{-2}$ day$^{-1}$), below

which plant C investment is limited, $p_{fth}$ is a threshold fraction determined by fungal and plant species. $P_{fungi\rightarrow plant}$ is the actual mycorrhizal fungal P supply to the plant (g P m$^{-2}$ day$^{-1}$), $P_{ilab\rightarrow root}$ describes plant uptake by roots (g P m$^{-2}$ day$^{-1}$).
P in the fungal biomass, $P_{fungi}$ (g P m$^{-2}$), is calculated as,

$$P_{fungi} = P_{soil\rightarrow fungi} - P_{fungi\rightarrow litter} - P_{fungi\rightarrow plant} \quad (14)$$

Fungal P litter production ($P_{fungi\rightarrow litter}$, g P m$^{-2}$ day$^{-1}$) is estimated from a first-order rate equation,

$$P_{fungi\rightarrow litter} = P_{fungi} \times p_{lrate} \times (1 - p_{fret}) \quad (15)$$





Where $P_{fungi}$ stands for fungal P content (g P m$^{-2}$), $p_{lrate}$ is the litterfall rate parameter (day$^{-1}$), and $p_{fret}$ is a parameter describing the fraction of P retained in fungal tissue during senescence.

P transfer from mycorrhizal fungi to plant, $P_{fungi \rightarrow plant}$ (g P m$^{-2}$ day$^{-1}$), is driven by plant P demand after root uptake, but regulated by P availability to fungi.

$$P_{fungi \rightarrow plant} = \begin{cases} P_{Demand} - P_{ilab \rightarrow root} & P_{Demand} - P_{ilab \rightarrow root} \leq P_{fungiavail} \\ P_{fungiavail} & P_{Demand} - P_{ilab \rightarrow root} > P_{fungiavail} \end{cases} \qquad (16)$$

Where $P_{fungiavail}$ is the P that can be acquired by fungi and transferred to the plant (g P m$^{-2}$), calculated as,

$$P_{fungiavail} = P_{fungi} - \frac{C_{fungi}}{p_{cp\,fungi\,\max}} \qquad (17)$$

Where $P_{fungi}$ is fungal P content (g P m-2), $p_{cpfungimax}$ is a parameter describing the predefined maximum C/P ratio of fungal tissue. This is based on the assumption that mycorrhizal fungi will only supply the plant with P as long as fungal C demand is fulfilled (Nehls, 2008).

### 3.5 Mycorrhizal fungi phosphorus uptake

Total and partial mycorrhizal fungal P uptake is calculated analog to the mycorrhizal fungal N uptake (He et al., 2018) as,

$$P_{soil \rightarrow fungi} = P_{ilab \rightarrow fungi} + P_{olit \rightarrow fungi} + P_{ohum \rightarrow fungi} \qquad (18)$$

Uptake from the $P_{ilab}$ pool to fungi is first limited by a potential uptake rate $P_{ilabpot \rightarrow fungi}$ (g P m$^{-2}$ day$^{-1}$), determined by the biomass of fungal mycelia.

$$P_{ilabpot \rightarrow fungi} = p_{i,rate} \times C_{fungi} \times p_{fmyc} \qquad (19)$$

Where $P_{ilabpot \rightarrow fungi}$ stands for potential fungi $P_i$ uptake rate (g P m$^{-2}$ day$^{-1}$), and $p_{i,rate}$ is a parameter which describes the mycorrhizal fungal potential uptake rate of $P_i$ per unite $C_{fungi}$ (g P g C$^{-1}$ day$^{-1}$), $p_{fmyc}$ is the fraction of fungal mycelia in total fungal biomass.

The actual uptake from the $P_{ilab}$ pool to fungi, $P_{ilab \rightarrow fungi}$ (g P m$^{-2}$ day$^{-1}$), is calculated by the potential uptake rate, further regulated by soil $P_{ilab}$ availability,

$$P_{ilab \rightarrow fungi} = \begin{cases} P_{ilabpot \rightarrow fungi} \times f(P_{fungidef}) & P_{ilabpot \rightarrow fungi} \leq P_{ilab} \times f(P_{fungiavail}) \\ P_{ilab} \times f(P_{fungiavail}) & P_{ilabpot \rightarrow fungi} > P_{ilab} \times f(P_{fungiavail}) \end{cases} \qquad (20)$$

Where $f(P_{fungiavail})$ is the availability function that calculates the fraction of $P_{ilab}$ that fungi can directly obtain (21), and $f(P_{fungidef})$ is the function that calculates the deficiency fraction that fungi can possibly uptake, which is determined by the fungi C/P ratio (eq. 22)



$$f(P_{fungiavail}) = p_{iavail} \times upt_{f,enh} \tag{21}$$

Where $p_{iavail}$ defines the fraction of $P_{ilab}$ that can be directly obtained by roots (also see eq. A.8 in Appendix A), $upt_{f,enh}$ is an enhanced uptake coefficient to account for the fact that fungal mycelia have higher uptake efficiency than roots.

The function of uptake deficiency fraction, $f(P_{fungidef})$ scales the unfulfilled capacity of fungi for P uptake, is calculated as,

$$f(P_{fungidef}) = 1 - \frac{p_{cpfungimax}}{C_{fungi} / P_{fungi}} \tag{22}$$

Where $p_{cpfungimin}$ is the defined minimum fungal C/P ratio parameter.

In our model, we assume that $P_i$ derived from the enzymatic hydrolysis of organic $P_o$ is directly taken up by fungi (termed organic uptake in this study). Similar to $P_{ilab \rightarrow fungi}$, uptake from the $P_{olit}$ pool to fungi is first limited by a potential uptake rate $P_{olitpot \rightarrow fungi}$ (g P m$^{-2}$ day$^{-1}$), determined by the biomass of fungal mycelia.

$$P_{olitpot \rightarrow fungi} = p_{olit,rate} \times C_{fungi} \times p_{fmyc} \tag{23}$$

Where $p_{olit,rate}$ is a parameter that describes the potential rate at which fungi mycelia acquire P from soil litter (g P g C$^{-1}$ day$^{-1}$). The actual uptake from the $P_{olit}$ pool to fungi, $P_{olit \rightarrow fungi}$ (g P m$^{-2}$ day$^{-1}$), is calculated by,

$$P_{olit \rightarrow fungi} = \begin{cases} P_{olitpot \rightarrow fungi} \times f(P_{fungidef}) \times frac_{P,lit} & P_{olitpot \rightarrow fungi} < p_{olitf} \times P_{olit} \\ p_{olitf} \times P_{olit} \times frac_{P,lit} & P_{olitpot \rightarrow fungi} \geq p_{olitf} \times P_{olit} \end{cases} \tag{24}$$

Where $p_{olitf}$ is the fungi organic uptake parameter that describes the uptake rate of soil litter $P_{olit}$ that can be hydrolyzed and directly acquired by fungi (day$^{-1}$), $frac_{P,lit}$ is introduced to ensure that fungi organic uptake is less than the missing plant demand after $P_{ilab}$ uptake, as well as to avoid uptake from only one organic pool, calculated as,

$$P_{of,max} = P_{olit} \times p_{olitf} + P_{ohum} \times p_{ohumf}$$

$$frac_{P,lit} = \min \left\{ \frac{P_{Demand} - P_{ilab \rightarrow plant}}{P_{of,max}} ; \frac{P_{olit} \times p_{olitf}}{P_{of,max}} \right\} \tag{25}$$

Where $p_{ohumf}$ is the fungi organic uptake parameter that describes the uptake rate of soil humus $P_{ohum}$ that can be hydrolyzed and directly acquired by fungi (day$^{-1}$), The same approach can be used to quantify fungal P uptake from the humus pool by replacing terms that include the litter P pool with the humus P pool in (23), (24) and (25).



The fungal mantle prevents contact between roots and the soil, and thereby limits the rate at which roots can directly acquire nutrients from the soil. Plant root $P_i$ uptake response to P availability and fungal mantle is calculated as,

$$f(P_{iavail}) = p_{iavail} \times e^{(-fm \times m)} \tag{26}$$

Where $p_{iavail}$ is a parameter that describes the maximum fraction of $P_{ilab}$ that is available for uptake by plant roots (i.e., not covered by the fungal mantle), and $fm$, which is an uptake reduction parameter that describes cover by the fungal mantle. $m$ is the mycorrhization degree, see He et al. (2018).

## 4 Description of the region used for simulation and model setup

### 4.1 Description of the region

The Coup-CNP model was tested on four managed forest regions, situated along a climatic, N and P deposition, and fertility gradient across Sweden – Västerbotten 64°N, Dalarna 61°N, Jönköping 57°N, and Skåne 56°N – the same four regions as in Svensson et al. (2008) and He et al. (2018). An overview of the climatic, geological, plant and soil characteristics of the four regions is provided in Table 1. In general, the four regions represent a North-South transect characterized by increasing mean air temperature (from 0.7 to 7.1 °C), precipitation (613-838 mm), and atmospheric N deposition (1.5 – 12.5 kg N ha$^{-1}$). The measured annual P deposition ranges from 0.06 to 0.28 kg P ha$^{-1}$, with the lowest and highest deposition rates observed in the 61°N and 57°N regions, respectively. For comparison, sites dominated by Scots pine (*Pinus sylvestris*) and/or Norway spruce (*Picea abies*) and Podzol soils (FAO, 1990) were chosen for all four regions (Table 1). Soil fertility, indicated by C-to-nutrient ratios, exhibited an increasing trend from North to South, however, the highest soil organic C/P ratio (thus the poorest P content) was measured in the 61°N region (Table 1). Soil mineral P content varied with geology (Table 1). The aqua regia extraction method was used to determine total soil mineral P content from regional till samples collected by the Geological Survey of Sweden (SGU) (Andersson et al., 2014). Samples were taken from the C-horizon at a depth of approximately 0.8 m, where the till is generally not disturbed by weathering. In general, Swedish till soils belong to the youngest and least weathered soils in Europe. High total mineral P contents can be found in southern (i.e. 57 and 56°N) and northern parts of the country (i.e. 64°N), which include apatite-iron ore districts (Table 1). Total mineral P content in central Sweden (e.g., 61°N) is much lower than in other parts due to the occurrence of marine and postglacial clays that cover, for example, the Mälaren region.



### 4.2 Model design and setup

The development of managed forests was simulated in daily resolution over a rotation period from stand age class 10 until 10 years after final harvesting to cover the potential nutrient
leaching during the regeneration phase as in Gärdenäs et al. (2003). The trees in all regions were assumed to be planted in 1961; thus, the period 1961 to 1970 represented the spin-up period. The harvesting intensities and rotation lengths were set specifically for each region following recommendations from SLU (2012). The management of forest stands ranged from two thinnings during a rotation period of 120 years for the least productive stands in northern
regions to four thinnings during a rotation period of 70 years in the most productive stands in southern Sweden (Table 1). According to general forest management guidelines, it was assumed that during thinnings 20% of the stem is removed while 5% transforms into litter (Swedish Forest Agency, 2005). For leaves and roots, it was assumed that 25% transforms into litter. For all of the regions, one clearance – during which 60% of the stands is removed -
was applied at the end of the first year after spin-up. During final felling, 5% of trees are remained intact, 90% of the stem is removed and 5% becomes litter, while it is assumed that 95% of all other plant components become litter. The surface cover parameters and litterfall rates of understory vegetation were modified from Svensson et al. (2008) to achieve a more realistic understory in those regions (Table 2).

Historical weather data were derived from the nearby SMHI weather station data through spatial interpolation for each region. Projections of future weather data were generated by the climate change and environmental objective (CLEO) project, using ECHAM5 projections and bias correction of regional climatic data (Personal communication to Thomas Bosshard, SMHI). N-concentration in the deposition was kept constant for each region throughout the
simulation as in Svensson et al. (2008) and Gärdenäs et al. (2003).

Literature data, both soil and biomass, were compiled from sites with coniferous forests on Podzols soil within the major moisture classes (mesic and moist), according to the Swedish National Forest Soil Inventory (NFSI) (Olsson et al. 2009, Stendahl et al., 2010.) The corresponding forest biomass data were based on measured standing stock volumes of
different age classes presented in the Swedish Forest Inventory (SFI) data (SLU, 2003), for more details, see Svensson et al. (2008).

Part of the model design and setup such as soil physical properties, soil depth (1 m), initial soil C content and C/N ratio followed what was reported by Svensson et al. (2008), who in turn had NSFI as the main source. He et al. (2018) additionally described explicit mycorrhizal



fungi settings. The following section will only chronicle the setup for the newly developed P

model, as well as describes parts of the model that differed from the aforementioned studies

(Svensson et al., 2008, He et al., 2018).

The P content in soil organic matter, i.e., soil organic matter C/P ratios, was based on

measurements performed at the sites during a Swedish Throughfall Monitoring Network

(SWETHTRO) project (Pihl Karlsson et al., 2015). Only the organic C/P ratio at the O

horizon was measured at most sites. Thus, in our calculations of the total stock of soil organic

P in the soil profile, we assumed that the C/P ratio measured for the O horizon also extends to

the other horizons (uncertainties associated with this assumption will be assessed by

including a range in the soil N/P ratios, 10-25, in the sensitivity analysis). The initial labile $P_i$

concentrations were set according to previous data from similar Swedish forest sites (Kronnäs

et al., 2019; Fransson and Bergkvist, 2000). Soil pH was set according to the NSFI data.

Initial soil organic P pools were partitioned between soil litter (5%) and humus pools (95%)

analog to N partition in Svensson et al. (2008), and decreased exponentially with depth

(Fransson and Bergkvist, 2000).

The sensitivity of plant growth, soil C and leaching loss responses to soil N and P availability

was assessed by varying the soil N/P ratio from 10 to 25 for the study regions (see Table 2

and Figure 5). These ranges were set according to previously published Swedish forest soil

data (Lagerström et al., 2009; Giesler et al., 2002; Kronnäs et al., 2019). Previous modelling

studies (Eckersten et al., 1995; He et al., 2018) have reported that the parameter 'fungal

organic N uptake rate' strongly affects N availability. For this reason, both fungal N and P

uptake rates were included in the sensitivity analysis so that we could determine how fungal

N and P uptake influence the response to soil N and P availability between soil N/P ratio 10

to 25 (Table 2). The range between the regional lowest and highest values of fungal organic

uptake rates for the four regions was used for the sensitivity analysis (Table 2). The newly

introduced parameters of P processes were mostly based on values from the literature (Table

3). For instance, the optimal leaf C/P ratios for forest growth, C/P ratios of individual plant

components were obtained from empirical measurements from Swedish forests (e.g. Thelin et

al., (1998; 2002)). The weathering and surface runoff parameters were defined according to

laboratory empirical data (e.g. Guidry and Machenzie, (2000)). The fungi related parameters

were mainly obtained from the previous CoupModel calibrations for the same regions (He et

al., 2018).

### 4.3 Datasets for model evaluation



Data used to evaluate the newly developed model include SFI biomass values from forest

stands aged 10 to 100 years (SLU, 2003). The measured leaf nutrient data used in the evaluation were obtained from managed forests sites within the SWETHTRO project (Pihl Karlsson et al., 2011; Pihl Karlsson et al., 2015) in the studied regions (some forest sites are also part of the ICP FOREST LEVEL II monitoring program, www.icp-forests.org). Data used in the North 64°N region include two Scots pine stand sites, Gransjö (N 64°30',

E17°24') and Brattfors (N64°29', E18°28'). For the 61°N region, two sites with Scots pine stands - Kansbo (N61°7', E 14°21') and Furudalsbruk (N61°12', E15°11') were used. Data describing the Fagerhult (N57°30', E15°20') site, dominated by Norway spruce, and the Gynge Scots pine stand (N57°52', E14°44') were used in the 57°N region. Three sites, including a Scots pine stand in Bjärsgård (N56°10', E13°8'), a Norway spruce stand in Västra

Torup (N56°8', E13°30') and a European Beech stand in Kampholma (N56°6', E13°30'), represented the 56°N region.

To compare model outputs with measured P leaching, $PO_4$ and total P data in stream water were obtained from the open database of environmental monitoring data (MVM, https://miljodata.slu.se/mvm/). DOP data were not available for the regions thus it was

calculated as the difference between total P and $PO_4$. Thus the measured DOP contains our simulated fractions DOP but also particular phosphorus. We used measured water outflow rates from the regional outlet from the Swedish Meteorological and Hydrological Institute (SMHI, https://vattenwebb.smhi.se/station/) to convert the concentrations into fluxes.

**5 Results**

**5.1 Model assessment**

The new Coup-CNP model was able to reproduce the observed development of forest tree biomass (SLU, 2003) over the rotation period well (Fig. 2). Note that the dips in the simulated biomass are related to the timing forestry operations in the model that is not

represented in the measured. The regional biomass data show an increasing trend from North to South, which the model captured clearly (Fig. 2). However, when the predictions were compared with observed plant biomass prior to final harvesting, the model showed a slight underestimation (12%) for the northern 61°N region and slight overestimations for the other regions (7%, 13% and 1% for the 64, 57 and 56 °N regions, respectively).

The simulated leaf C/P ratios agree fairly well with the available SWETHTRO data (Pihl Karlsson et al., 2011; Pihl Karlsson et al., 2015), despite general overestimation of 10%,



32%, 30% and 21% from North to South. The average measured leaf C/P ratio in the four regions was 396 (standard deviation, 48), 398 (59), 355 (45) and 396 (72), respectively. The model found that the 56 and 61 °N regions have higher C/P ratios than the other regions,

which was also noticed in the observational data (Fig. 2). The average measured leaf C/N ratios were 44 (4), 41 (3), 36 (5) and 31 (7), respectively. The model was accurate in simulating leaf C/N ratios, and identified a similar decreasing leaf C/N trend from North to South. The exception was a slight leaf C/N overestimation for the 57 °N region (Fig. 2). For leaf N/P ratios, the average of the observations from North to South were 9.1 (1), 9.6 (1.3),

9.9 (1.4) and 13.4 (3.8), respectively. The Coup-CNP model was also able to accurately reproduce the measured leaf N/P ratios, as well as reveal an increasing leaf N/P trend with decreasing latitudes (Fig. 2). Of the climate variables, the radiation absorbed by tree canopy increased from the North to the South, while the temperature and water limitation of Gross Primary Production (GPP) declined from the North to the South (Table 4). N was the most

limiting nutrient at the 64 °N and 57 °N, while P was the most limiting nutrient at the 61 °N region and 56 °N (Table 4). The limiting effect of P availability could be seen in the predicted relatively high N/P ratios, as the 56 °N region – and, to a lesser extent, the 61 °N region as this region is also N limited (Fig. 2).

Total annual plant N and P uptake rates in the northernmost region were modelled to be 3.7 g

N m$^{-2}$ year$^{-1}$ and 0.4 g P m$^{-2}$ year$^{-1}$, respectively. The southernmost region demonstrated three times higher N uptake rates and two times higher P uptake rates than the northernmost region (Table 4). Uptake of the organic fraction of total N decreased from North to South (Table 4). The modelling results also indicated that the uptake of organic P is necessary to satisfy the demands of the plant. However, uptake of the organic fraction of total P was not found to be

associated with latitude or the C/N ratio. Instead, it is regulated by soil C/P ratio and geology (Tables 1 and 5). The amount of N in the fungi litter fraction of total plant litter decreased from North to South, but this was not the case for P, as the amount of P remained stable in the corresponding fraction (Table 4).

The simulated annual soil C sequestration rates were 2, -2, 9, and 15 g C m$^{-2}$ year$^{-1}$ from

North to South (Figs. 3a, 4a, Table 5). Thus, the soil C stock was generally in a steady-state over the forest rotation period, with slightly higher C sequestration rates predicted for the southern regions (Figs. 3a, 4a). The soil C/N ratios of all of the regions were in a steady-state over the forest rotation period. In contrast, the C/P ratios and N/P ratios showed a slightly increasing trend over the rotation period, with the exception of the soil N/P ratio in the 64 °N

region (Fig. 3b, c, d).



The modelled P leaching generally reflected the observational data, however, the mean estimated concentrations were often lower than the measurements available for each region (MVM, https://miljodata.slu.se/mvm/, Table 6). The data show that P losses through leaching have a small effect on the system, i.e., they account for approximately one-third of the annual

deposition input, while DOP losses were more dominant in the northern systems (Table 6). However, the simulated proportion of DOP in total losses through leaching was much lower than what had been measured, and the decreasing trend from North to South identified in the simulations was not supported by the observational data (Table 6).

**5.2 Modelled forest C, N and P budgets**

Regarding C assimilation, the average plant growth over the rotation period was predicted to be three times higher in the southernmost region than that in the northernmost region (Fig. 4a). Most of the forest productivity was harvested, the change in plant C was small as simulation started when plants were 10 years old and ended when they were 10 years old.

Regarding the N budget, the northernmost ecosystem showed a slight loss while the southern ecosystems showed N gains. The N sequestration rates generally increased towards the southern latitude (Fig. 4b). The P budget shows an opposite pattern, as the northernmost ecosystem was in balance while the other three ecosystems showed P losses, with total losses increasing from North to South (Fig. 4c).

Most of the C captured from the atmosphere was the harvested plants (Fig. 4a). Our model predicted small losses of DOC through leaching, and the forest soil in all of the regions was found to be in a quasi-steady-state with generally low sequestration. An exception was the region with the lowest P-availability (61 °N), which showed soil C losses (Tables 1, 5).

Our results identified atmospheric deposition as the main N input. When accounting only for

harvested N, 60%, 53%, 35% and 36% of the deposited N was removed from the 64, 61, 57 and 56 °N regions, respectively (Fig. 4b). The N accumulated in standing plants and harvested plants accounted for 104%, 80%, 54% and 55% of the annual N deposition in the 64, 61, 57 and 56 °N regions, respectively. The model results show that soils in the two northern regions will lose N while soils in the two southern regions will accumulate N.

Annual average losses through leaching were predicted to increase from North to South, and ranged from 0.09, 0.19, 0.27, 0.47 g N m$^{-2}$ from North to South, which corresponds to 60%, 45%, 21%, 41% of the annual N deposition, respectively (Fig. 4b).

The simulated annual P weathering fluxes ranged from 0.009 to 0.025 g P m$^{-2}$ year$^{-1}$, and showed similar magnitudes as the deposition inputs (Fig. 4c). The most significant source of





P losses over the rotation period was plant harvest, which removed 89%, 255%, 108% and 167% of the deposited P from the 64, 61, 57 and 56 ºN regions, respectively (Fig. 4c). When the P that accumulated in standing plants and harvested plants is considered together, this accounts for 85%, 147%, 90% and 114% of the total P input through deposition and weathering for the 64, 61, 57 and 56 ºN regions, respectively. The simulation showed that

soils from all four studied regions are slightly losing P, with the annual losses ranging from 0.01 to 0.03 g P m$^{-2}$ from North to South (Fig. 4c).

### 5.3 Impacts of forest growth, soil C and leaching on soil N and P levels

Forest growth, measured through harvested biomass, increased as the soil N/P ratios

increased from 10 to 15, but decreased once an optimum soil N/P ratio of around 15-20 was reached. This trend was noted for three studied regions (Fig. 5a), however getting less pronounced moving north and it was almost not detectable for the northernmost region where GPP was strongly limited by radiation (average absorbed radiation 3.89×10$^6$ J m$^{-2}$ day$^{-1}$ at 64 °N and 6.57×10$^6$ J m$^{-2}$ day$^{-1}$ at 56 °N, Table 4). The lowest air temperature and precipitation

in 64 °N out of the four regions also contribute to the GPP limitation (temperature/water limitation of GPP 0.47/0.45 at 64 °N and 0.67/0.65 at 56 °N, Table 4). Soil C sequestration between 56-61 °N latitudes was found to be highly sensitive to soil N/P ratios, with the model predicting that soil C sequestration would consistently decrease as the soil N/P ratio increases (Fig. 5b). In addition, total P losses through leaching generally decreased as soil N/P ratios

increased; an exception was the 57 ºN region where they increased again for soil N/P ratios above 15-18, the same range with maximum plant harvest (Fig 5a). In contrast, total N losses through leaching were found to be positively correlated with the soil N/P ratio above 15, with this relationship more pronounced for the southern regions (Fig. 5c, d). Thus, the sensitivity analysis results indicate that strong C-N-P interactions are also prevalent in forest ecosystems

(Fig. 5). Forest soils with soil N/P ratios above 15-18 were predicted to exhibit slower forest growth rates, lower soil C sequestration (potentially even losses), and high N leaching risk.

## 6 Discussion

### 6.1 Modelled P budgets and comparison with published data

It is important to compare our modelled P fluxes with previously reported values. In a study that applied the PROFILE model and empirical data, Akselsson et al. (2008) estimated the average weathering rate in Swedish forests (down to a depth of 0.5 m) to be 0.009 g P m$^{-2}$



year$^{-1}$, ranging from 0.001 to 0.024 g m$^{-2}$ year$^{-1}$ (5% to 95% percentile). This can be compared with our simulated P weathering rate (down to a depth of 1 m soil depth) range of 0.009 to 0.025 g P m$^{-2}$ year$^{-1}$. Both our estimations and those by Akselsson et al. (2008) were far lower than the 0.071 g P m$^{-2}$ year$^{-1}$ (0.5 m depth) reported by Yu et al. (2018) for a spruce forest on Podzol soil in southern Sweden. It is important to mention that Yu et al. (2018) suggested that the weathering rate they provided was an overestimation.

The modelled total plant P uptake rates in this study ranged from 0.4 to 1 g P m$^{-2}$ year$^{-1}$ (Table 4), which is slightly higher than the $0.5 \pm 0.4$ to 0.96 g P m$^{-2}$ year$^{-1}$ reported by Johnson et al. (2003) and Yanai (1992) for temperate forests, and the 0.5 g P m$^{-2}$ year$^{-1}$ reported for a southern Swedish forest by Yu et al. (2018). One explanation for this discrepancy could be that Coup-CNP explicitly considers mycorrhizal processes related to P uptake, e.g., the presented estimates revealed that mycorrhizal fungi accounted for more than half of total plant P uptake (Table 4). This highlights that mycorrhizal fungi are crucial to plant P acquisition in forest ecosystems. The estimated P uptake by fungi was - to a large extent - proportional to the rates estimated for N (Table 4). He et al. (2018) compared explicit and implicit models and found that CoupModel v5.0 predictions of plant N uptake were higher when mycorrhizal fungi were explicitly included in the model. Furthermore, it is important to note that previous accounts of empirical data (Johnson et al., 2003; Yanai, 1992), as well as the ForSAFE model (Yu et al., 2018), did not account for P uptake by understory vegetation. In this study, understory vegetation was estimated to contribute to c.a. one-third of total P uptake in northern regions and one-sixth of total P uptake in southern regions (data not shown).

Akselsson et al. (2008) reported that, in Swedish forests, whole-tree harvesting causes average P removal of 0.054 g P m$^{-2}$ year$^{-1}$ with a range from 0.016 to 0.13 g P m$^{-2}$ year$^{-1}$. This agrees well with our modelled range (0.012 to 0.038 g P m$^{-2}$ year$^{-1}$) as well as the value reported by Yu et al. (2018), 0.037 g P m$^{-2}$ year$^{-1}$.

The P balances estimated for the ecosystem in this study ranged from 0 to -0.02 g P m$^{-2}$ year$^{-1}$ (with the negative value representing P losses). This agrees with what has been reported by Akselsson et al. (2008), i.e., an average P balance of -0.029, ranging from 0.008 g P m$^{-2}$ year$^{-1}$ in the North to -0.1 g P m$^{-2}$ year$^{-1}$ in the South. The modelling by Yu et al. (2018) yielded P accumulation of 0.004 g P m$^{-2}$ year$^{-1}$ over a 300-year period in South Sweden. This predicted gain in P over the simulation period, which was very low, could have been due to relatively high P inputs via weathering. Our modelled regional P budget implies that clear-felling harvesting will result in a negative P balance for most Swedish forests even when P uptake by



mycorrhizal fungi in nutrient-poor forests is accounted for, with an exception being the northernmost region.

### 6.2 Implications of P availability on forest C and N dynamics

Our results demonstrate that Swedish forests are increasingly P-limited, a trend that was especially noticeable at southern latitudes (Table 4). N limitation was even more severe than P limitation at 64 and 57 °N regions. Furthermore, the northernmost region had much less radiation intercepted by the canopy, which partly masks the response to nutrient limitation, they may appear less sensitive to nutrient limitation. This was supported by the observed leaf N/P ratios (average values between ca. 9-14), which are recognized to reflect the state of nutrient limitation in forest trees (e.g. Jonard et al. 2015). In Swedish forests, leaf N/P ratios below 7 are normally considered an indicator of N limitation, while ratios above 12 signal P limitation (Rosengren-Brinck and Nihlgård, 1995; Yu et al., 2018). Linder (1995) has previously reported an optimal N/P ratio of 10 for forests in northern Sweden. Our leaf N/P ratio estimates were within these ranges, with the exception of the southernmost region (Fig. 1). The ratio of total plant P uptake to total N uptake in the southernmost region was much lower than what was measured for the other regions (Table 4), which further suggests P limitation in the southernmost region. The 61 °N region, which was characterized by the lowest P inputs among the studied regions due to geology and deposition (Table 1) (Fig. 4), was also shown to be P-limited (Table 4). This low P input also explains why this region showed the highest simulated fraction of organic P uptake (Table 4). Our modelling suggests that northern regions, which have traditionally been conceived as N-limited (Högberg et al., 2017), may experience P limitation or co-limitation by N and P. For instance, the 57 °N region showed an overall N limitation as the average value of GPP response to N, 0.30 is lower than the GPP response to P, 0.34 over the rotation period (Table 4). However, our model results further showed a lower GPP response to P, thus P limitations during the initial c.a. 10 years of stand development (data not shown). This suggests co-limitation could still occur since the nutrient limitations could potentially shift during forest development stages. For example, Tarvainen et al. (2016) reported a decrease in needle P following N fertilization in a Scots pine forest in northern Sweden. Several groundwater discharge areas were also shown to be P-limited (Giesler et al., 2002). Sundqvist et al. (2014) and Vincent et al. (2014) reported that alpine ecosystems in northern Sweden may also be P-limited.

The removal of harvest residues from final fellings for use in biofuel production is common, and expected to increase in southern and central regions of Sweden (Cintas et al., 2017, Ortiz


et al., 2014; Stendahl et al., 2010). Our modelling indicates that clear-cutting or final-felling will significantly impact the forest P balance and soil C sequestration (Figs. 4c, 5b). Furthermore, it is important to note that this practice was found to affect P availability more than N availability, especially in southern Sweden (Figs. 4b, 4c). Simulations with earlier

versions of CoupModel have also revealed N depletion for final-felling/clear-cutting scenarios in northern Sweden, but reported N gains for southern Sweden (He et al., 2018; Svensson et al., 2008; Gärdenäs et al., 2003).

The soil C sequestration simulated by the Coup-CNP model is generally comparable with what has been reported in previous studies (Table 5). Plants in the north will need to acquire

nutrients to meet demands for growth, but the Coup-CNP model showed that plants acquire a smaller fraction of total nutrients than what was previously estimated (Coup-CN model; see Table 8 in Svensson et al. (2008)). Our results further suggest that P regulates SOC, as an increasing soil N/P ratio will decrease soil C sequestration rates (Fig. 5b).

The sensitivity analysis results found the optimum soil N/P ratio for forest production to be

15 to 20 on podzol soils for 61-56 ºN regions (Fig. 5). Manzoni et al. (2010) reviewed the forest litter decomposition process and found that litter C-to-nutrient ratios decreased - towards a C/N ratio of 20 and C/P ratio of 350 (thus an N/P ratio of 17.5) - as decomposition proceeded. A synthesis of long-term decomposition studies in northern forests also showed that the N/P ratio of both fine litter and woody residues converges to c.a. 20 (Laiho and

Prescott, 2004). The optimum range identified by the Coup-CNP model is thus similar to these observed convergence ratios, which generally represent the shift from immobilization during the initial decomposition phase to net mobilization (Penuelas et al., 2013; Güsewell, 2004; Cleveland and Liptzin, 2007). Lagerström et al. (2009) measured soil and microbial nutrient contents in 30 diversified forest islands in northern Sweden that vary considerably in

terms of fertility. Surprisingly, they found that microbial biomass N/P ratios remained unchanged across the gradient, suggesting that nutrient availability is mainly determined by soil organic N/P ratios. The identified bell shape response of plant growth to the soil N/P ratio thus highlights the importance of nutrient stoichiometry. This implies that forests with N/P below the optimal range can benefit from N fertilization, which will stimulate forest

growth and reduce the P leaching risk. In contrast, P fertilization in forests with N/P above the optimal range will stimulate forest growth, promote soil C sequestration and reduce N leaching (Figure 5). A synthesis of long-term water quality measurements from forest streams in the geochemical monitoring network (GEOMON) found total N fluxes to be tightly linked to DON/TP ratios (Oulehle et al., 2017). As such, total N leaching increased with the


DON/TP ratio, a finding which agrees with the results obtained in this modelling study. The presented modelling predictions thus corroborate that decreased P availability can profoundly affect the N cycle and catchment retention.

To summarize, the presented model (CoupModel v6.0) demonstrated that considering the P cycle in ecosystem models can significantly impact estimations of forest C and N dynamics.

This is an important finding in the context of climate change and forest management, as researchers need to have tools that will reliably model the C-N-P dynamics in an ecosystem. Climate change research strives to maximize C accumulation in terrestrial ecosystems, but this may currently be limited by P availability, which will be further jeopardized by the removal of forest residues for bioenergy production. The presented results show that forest

growth in southern regions, which are characterized by high N deposition and already show limitation by P, will be most affected (Fig. 4c, Table 4) (Akselsson et al., 2008; Yu et al., 2018; Almeida et al., 2018).

**7 Conclusions**

This paper describes the most recent version (6.0) of CoupModel, which explicitly considers the phosphorus cycle and mycorrhizal interactions. The simulations of the C, N, and P budgets for four forest regions were complete and accurate based on evaluation with empirical forest biomass, leaf nutrient ratio, and P leaching data. The development and evaluation of this new model demonstrate that P availability needs to be considered when

studying how climate change will influence C turnover and ecosystem responses, otherwise important feedback mechanisms may be overseen and the potential land sink of C overestimated. Thus, the detailed description of all the Coup-CNP components and their interactions between the water, heat, C, N, and P cycles - are highly relevant to future studies. Our model results showed that N was the most limiting nutrient at the 64 °N and 57 °N, while

P was the most limiting nutrient at the 61 °N region and 56 °N (Table 4). The N limitation at 64 and 57 °N regions was more severe than P limitation. Furthermore, the northernmost region had less radiation intercepted by the canopy and lower temperature and precipitation, which may mask or make them less sensitive to nutrient limitation. During the simulated rotation period, southern forests showed P losses, mainly through harvest and changes in soil

storage, while northern forests were close to a steady-state in P availability. Mycorrhizal fungi accounted for half of total plant P uptake in all of the regions, which highlights the crucial role of the mycorrhiza in Swedish forests. A sensitivity analysis determined that a soil



N/P ratio of 15 to 20 is optimal for forest growth. Furthermore, soil N/P ratio above 15-20 decreased soil C sequestration and total P leaching, while significantly increased N leaching.

The largest P outflow over the rotation period was found to be removal via final-felling.

We conclude that the potential P-limitation of terrestrial ecosystems highlights the need of a proper consideration of the P cycle in biogeochemical models. The inclusion of the P cycle enable to account for possible feedback mechanisms of importance for prediction of C sequestration and N leaching under climate change and/or elevated N deposition.


## 8 Code and data availability

The model and extensive documentation, including tutorial exercises, are freely available from the CoupModel home page: http://www.coupmodel.com/ (CoupModel, 2019). CoupModel is written in the C++ programming language and runs with a GUI under the

Windows systems, but can also be run on other platforms without GUI. Version 6.0, from 03 July 2019, was used for the presented simulations. This version is archived on Zenodo (https://zenodo.org/record/3547628#.Xn3Bc0F7lEZ), as are the simulation files including the model and calibration set-up, parameterization settings, and corresponding input and validation files.






**Appendix A: Equations and parameterization regarding phosphorus processes that are analogous to those for nitrogen**

The following section provides the equations for P processes that are analogous to those of N, as well as discusses parameterization aspects. The inclusion of the N cycle in CoupModel was previously described by Gärdenäs et al. (2003), Jansson and Karlberg (2011), and He et al. (2018).

**A1 Deposition and fertilization**

Atmospheric deposition, $P_{dep \to ilab}$ is treated as a model input using the parameter $p_{dep}$. In contrast to N deposition, only dry P deposition is considered since wet deposition is generally neglectable. Fertilization $P_{fert \to ilab}$ is also treated as a model input and calculated as,

$$P_{fert \to ilab} = p_{kfert} \; P_{fert} \tag{A.1}$$

Where $P_{fert \to ilab}$ is the rate of fertilizer P addition (g P m$^{-2}$ day$^{-1}$) and $p_{kfert}$ is the specific dissolution rate of commercial fertilizer (day$^{-1}$). The value of $p_{kfert}$ depends on fertilizer type and moisture conditions, e.g., in our model, a value of 0.15 corresponds to a half-time of 5 days, and that 90% of the fertilizer is dissolved into the $P_{ilab}$ pool within 15 days. If manure fertilizer is used, the organic P$_o$ in the manure is added into a separated organic pool $P_{ofae}$,
termed faeces. Fecal processes are similar to those of soil litter, described below.

According to a global compilation of published data, the average annual global P deposition is 0.027 g P m$^{-2}$ year$^{-1}$ (0.033 for Europe), which equals to 0.000074 g P m$^{-2}$ day$^{-1}$ (Tipping et al., 2014; Schlesinger, 1997).

**A2 Mineralization-Immobilization & decomposition**

The $P_{ilab}$ pool is also controlled by biological demand and turnover (Olander and Vitousek, 2005). The P flux of biological mineralization-immobilization is calculated precisely as for N, in that C fluxes from litter (or faeces) to humus or from humus to atmosphere are driven by the microbial need for energy. The non-symbiotic microbes are implicitly simulated using
a fixed microbe C/P ratio parameter. The C/P ratio for microbes ($cp_m$) can vary widely, ranging from approximately 25–400 (see review by Manzoni et al. 2010).



$$C_{DecomL} = k_l \times f(T) \times f(\theta) \times C_{Litter}$$

$$P_{olit \rightarrow ilab} = C_{DecomL} \left( \frac{1}{C_{litter}/P_{litter}} - \frac{f_{e,l}}{cp_m} \right) \quad \text{(A.2)}$$

$$P_{olit \rightarrow ohum} = \frac{C_{litter \rightarrow humus}}{cp_m}$$

Where $k_l$ is the decomposition coefficient for soil litter (day$^{-1}$), $C_{litter}$ is the size of the litter pool (g C m$^{-2}$), $f(T)$ and $f(\theta)$ are common temperature and water content response functions for decomposition, for more details see Jansson and Karlberg (2011). Humus decomposition is calculated by changing pool size and the decomposition coefficient in the previous equation into terms that describe humus. $P_{olit \rightarrow ilab}$ is the mineralization flux from the soil litter pool to the $P_{ilab}$ pool (g P m$^{-2}$ day$^{-1}$). $P_{olit \rightarrow ohum}$ is the humufication flux rate. $C_{DecomL}$ is the C decomposition flux of soil litter (g C m$^{-2}$ day$^{-1}$), whereas $C_{litter}/P_{litter}$ and $cp_m$ are the C to P ratio in the litter pool and microbes. $f_{e,l}$ is a microbial efficiency parameter which represents the fraction of mineralized C that remains in the soil. Corresponding fluxes are calculated by changing the efficiency parameter to $f_{e,f}$ or $f_{e,h}$, along with changing the litter C/P ratio to a fecal C/P ratio or humus C/P ratio, gives the corresponding flow from the fecal pool, $P_{ofae \rightarrow ilab}$, or the humus pool, $P_{ohum \rightarrow ilab}$, respectively. A negative value means that net immobilization takes place.

The total biological mineralization is calculated as,

$$P_{biomin} = P_{olit \rightarrow ilab} + P_{ohum \rightarrow ilab} + P_{ofae \rightarrow ilab} \quad \text{(A.3)}$$

The biochemical mineralization process includes the release of root exudates, e.g., efflux of protons and organic anions, phosphatase and cellulolytic enzymes required for the hydrolysis or mineralization of $P_o$ (Richardson and Simpson, 2011; Bünemann, 2015; Hinsinger, 2001). This additional mineralization process is driven by plant demand for P (Richardson et al., 2009). Bünemann (2008) reviewed the existing enzyme addition experiments and showed, for example, that the phosphatase enzyme has low substrate specificity and that up to 60% of total organic $P_o$ in soil can be hydrolyzed and mineralized. We, therefore, assume that biochemical mineralization can occur from both the soil litter and humus pools. The flux rate is calculated as a first-order function regulated by pool size and uptake rate. Furthermore, it is assumed that the flux rate will not exceed the remaining plant demand after root $P_i$ uptake (equ A.8). The following equation (A. 4) is used when symbiotic microbes are implicitly simulated.





$$P_{olit \to plant} = frac_{P,litter} \times o_{uptPlitter} \times P_{olit}$$
$$P_{ohum \to plant} = frac_{P,humus} \times o_{uptPhumus} \times P_{ohum}$$
$$P_{bioche,max} = P_{olit} \times o_{uptPlitter} + P_{ohum} \times o_{uptPhumus}$$


$$frac_{P,litter} = \min\left\{\frac{P_{Demand} - P_{ilab \to plant}}{P_{bioche,max}}; \frac{P_{olit} \times o_{uptPlitter}}{P_{bioche,max}}\right\}$$ (A.4)

$$frac_{P,humus} = \min\left\{\frac{P_{Demand} - P_{ilab \to plant}}{P_{bioche,max}}; \frac{P_{ohum} \times o_{uptPhumus}}{P_{bioche,max}}\right\}$$

Where $P_{olit \to plant}$ and $P_{ohum \to plant}$ represent the biochemical mineralization fluxes from the litter and humus pools (g P m$^{-2}$ day$^{-1}$), assuming immediate uptake by the plant roots after mineralization. $o_{uptPlitter}$ and $o_{uptPhumus}$ are coefficient parameters that define the maximum plant uptake rates from the soil litter and humus pools, respectively. $P_{olit}$ and $P_{ohum}$ are the

pool sizes (g P m$^{-2}$), $frac_{P,litter}$ and $frac_{P,humus}$ are introduced to ensure that biochemical mineralization is less than the missing plant demand after $P_{ilab}$ uptake, as well as to ensure proportional uptake from the $P_{olit}$ and $P_{ohum}$. In this modelling framework, the inorganic $P_i$, when released by enzymatic activities acquired directly by the plants rather than entering the $P_i$ pool.

Total biochemical mineralization is calculated as,

$$P_{biochem} = P_{olit \to plant} + P_{ohum \to plant}$$ (A.5)

The total mineralization-immobilization flux is calculated as,

$$P_{totmin} = P_{biochem} + P_{biomin}$$ (A.6)

As is the case with DOC/DON, in Coup-CNP, organic P dissolution is described as a

microbial decomposition process. The dissolved organic matter can be fixed by humus via adsorption, precipitation, etc. A fixation coefficient, $d_{DOD}$, which varies between layers, was introduced (Kalbitz et al., 2000; Kaiser and Kalbitz, 2012). The equation for DOP is similar to that for DOC, and is calculated as,

$$P_{olit \to dop} = d_{DO,l} \times f(T) \times f(\theta) \times P_{olit}$$
$$P_{ohum \to dop} = f(T) \times f(\theta) \times (d_{DO,h} \times P_{ohum} - d_{DOD}(z) \times P_{dop})$$ (A.7)

Where $d_{DO,l}$ and $d_{DO,h}$ are the dissolution rate coefficients (day$^{-1}$) for the litter and humus, $f(T)$ and $f(\theta)$ are common response functions for soil temperature and water content, and identical to those used for the decomposition process (equ A.2).

**A3 Plant growth and P uptake**



Plants can acquire $P_i$ through both the roots and mycorrhizal fungi; for this reason, both of these processes were simulated. We assume that uptake of $P_i$ by roots is driven by net photosynthesis and determined by plant demand, yet constrained by the $P_{ilab}$ pool size.

$$P_{ilab \rightarrow root} = min\ (p_{iavail} \times P_{ilab};\ P_{demand})  \tag{A.8}$$

Where $P_{demand}$ is the plant P demand, based on the C/P ratios of various plant compartments ($_{iplant}$ includes leaf, stem, fine roots and coarse roots),

$$P_{demand} = \sum_{iplant} \frac{C_{a \rightarrow iplant} - C_{iplant \rightarrow a}}{cp_{iplant\,min}}  \tag{A.9}$$

Where $C_{a \rightarrow iplant}$ is the photosynthesis assimilation for each compartment $_{iplant}$ (g C m$^{-2}$ day$^{-1}$), $C_{iplant \rightarrow a}$ is the respiration of each compartment, and $cp_{iplantmin}$ is the defined minimum C/P ratio for each plant compartment. Empirical measurements show that the C/P ratio of leaves generally varies between 200-600, while the stem requires C/P between 1000-3000 and roots require C/P between 500-1500 (Bell et al., 2014; Tang et al., 2018). It should be noted that the compartment C/P ratio is calculated for each time step; thus, the model provides flexible stoichiometry.

In addition, increasing soil P abundance, particularly when P fertilizer is added, is known to decrease belowground C allocation (Ericsson, 1995). We assume that an increasing C/P ratio (i.e. decreasing P content) in the leaf, $C/P_l$, will increase belowground allocation (e.g., $frac(root)$).

$$frac_{a \rightarrow root}(C/P_l) = r_{cpc1} + r_{cpc2} \times C/P_l$$
$$frac(root) = frac_{a \rightarrow root}(C/P_l) \times frac_{a \rightarrow root}(C/N_l)  \tag{A.10}$$
$$C_{a \rightarrow root} = C_{a \rightarrow plant} \times frac(root)$$

Where $r_{cpc1}$ and $r_{cpc2}$ are the plant allocation pattern parameters, determined by plant species and a similar equation as what was used to calculate $frac_{a \rightarrow root}(C/N_l)$ (He et al. 2018). CoupModel can additionally account for the effects of water stress on plant allocation. In this study, C allocation to roots is assumed to be constrained by both N and P contents in the leaves, i.e., $frac_{a \rightarrow root}(C/N_l)$ and $frac_{a \rightarrow root}(C/P_l)$.

**A4 Plant litterfall**

Plant litterfall P fluxes are proportional to the corresponding C fluxes, and determined by the C/P ratio of each compartment $_{iplant}$ ($_{iplant}$=leaf, steam, grain, fine roots, and coarse roots), calculated as,





$$P_{plant \to soil} = \sum_{iplant} \frac{c_{liplant} \times C_{iplant} \times (1 - c_{iplantret})}{C/P_{iplant}}$$ (A.11)

Where $c_{liplant}$ is the litterfall rate (day$^{-1}$) for plant compartment $_{iplant}$, $C_{iplant}$ is the C stock in that compartment (g C m$^{-2}$), and $c_{iplantret}$ is a parameter defined as the fraction that was retained before litterfall. Total litterfall also includes inputs from mycorrhizal fungi. The litterfall flux is directly added to the surface soil litter pool, or to the layer in which it formed when it was produced by roots and fungi. The average C/P ratio of fresh litter varies widely,

100 - 4100 (Manzoni et al. 2010). The retention of nutrients prior to leaf senescence is one of the main factors that affect the C/P ratio of fresh litter.

**A5 Leaching and surface runoff**

The losses of soluble $P_{isol,loss}$ (g P m$^{-2}$ day$^{-1}$) are modelled through the transport of water,

$$P_{isol,loss} = \sum_{jlayer} P_{isoldrainage,j} + P_{isolpercolation}$$

$$P_{isoldrainage,jlayer} = \frac{P_{isol,j}}{\theta_{,j} \times \Delta z_{,j}} \times q_{drainage,j}$$ (A.12)

$$P_{isolpercolation} = \frac{P_{isol}}{\theta \times \Delta z} \times q_{percolation}$$

Where $q_{drainage}$ is the water flow (mm day$^{-1}$) due to drainage, and $q_{percolation}$ is the deep percolation flow (mm day$^{-1}$), $\theta_{,j}$ is the water content (volume %) at the soil layer $j$, and $\Delta z_{,j}$ is the layer thickness (m) at soil layer $j$. The vertical $P_i$ flow between layers is calculated through a similar equation.

DOP losses from the system is calculated as,

$$P_{dop,loss} = \frac{P_{dop}}{\theta \times \Delta z} \times \left( \sum_{jlayer} q_{drainage,j} + q_{percolation} \right)$$ (A.13)

In addition, we also accounted for particulate phosphorus (*PP*) losses, e.g., due to soil erosion, subsidence and lateral losses of secondary minerals and occluded P due to surface runoff. We assume the *PP* loss is proportional to the water flow. When surface runoff occurs,

for example, during snow melting, the loss is assumed to occur only for the first soil layer (soil surface).

$$P_{m,loss} = q_{surfacerunoff} \times k_{scale}$$

$$k_{scale} = \min(1, \frac{q_{surfacerunoff}}{q_{thr}}) \times (P_{\Delta,i} - P_{base,i}) + P_{base,i}$$ (A.14)





Where $q_{surfacerunoff}$ is the surface runoff flow (mm day$^{-1}$). An empirical scale factor $k_{scale}$ is
introduced to account for the concentration of erodible $P_m$ over the flow rate of surface

runoff. $P_\Delta$ (mg P l$^{-1}$), $P_{base}$ (mg P l$^{-1}$) and $q_{thr}$ (mm day$^{-1}$) are empirical coefficients.

Therefore, the total P losses are calculated as,

$$P_{totloss} = P_{isol,loss} + P_{dop,loss} + P_{m,loss} \tag{A.15}$$

**A6 P removal during plant harvest**

The removal of P during plant harvesting was calculated in a similar way as C losses through

harvesting, and depends on the C/P ratio of the plant compartment.

**Appendix B: Simulated annual mean P, N and C budgets, generated by varying three
regional key parameters, including soil N/P ratio and the fungal organic uptake rates**

**for N and P**

**Table B1 Simulated annual P budget, with the associated uncertainty range (mean ±
SD, g P m$^{-2}$ year$^{-1}$)**

| P budget | 64°N | 61°N | 57°N | 56°N |
|---|---|---|---|---|
| Weathering | 0.014 (0.0002) | 0.0094 (0.0002) | 0.024 (0.0002) | 0.025 (0.001) |
| Deposition | 0.013 | 0.0065 | 0.028 | 0.023 |
| Leaching | 0.0025 (0.0003) | 0.0015 (0.0003) | 0.004 (0.0004) | 0.006 (0.0009) |
| Harvest export | 0.01 (0.002) | 0.018 (0.003) | 0.03 (0.004) | 0.045 (0.01) |
| Change in plant | 0.0125 (0.003) | 0.007 (0.002) | 0.018 (0.006) | 0.017 (0.006) |
| Change in soil | -0.012 (0.005) | -0.02 (0.002) | -0.024 (0.005) | -0.045 (0.01) |
| Change in ecosystem | 0.0005 (0.0004) | -0.013 (0.003) | -0.006 (0.003) | -0.028 (0.007) |





**Table B2 Simulated annual N budget, with the associated uncertainty range (mean± SD, g N m⁻² year⁻¹)**


| N budget | 64ºN | 61ºN | 57ºN | 56ºN |
|---|---|---|---|---|
| Deposition | 0.15 | 0.35 | 0.78 | 1.26 |
| Leaching | 0.10 (0.004) | 0.15 (0.02) | 0.16 (0.05) | 0.45 (0.15) |
| Harvest export | 0.08 (0.01) | 0.19 (0.02) | 0.27 (0.04) | 0.5 (0.19) |
| Change in plant | 0.09 (0.009) | 0.10 (0.008) | 0.15 (0.03) | 0.17 (0.01) |
| Change in soil | -0.12 (0.03) | -0.09 (0.02) | 0.20 (0.05) | 0.14 (0.03) |
| Change in ecosystem | -0.03 (0.005) | 0.01 (0.008) | 0.35 (0.03) | 0.31 (0.09) |

**Table B3 Simulated annual C budget, with the associated uncertainty range (mean± SD, g C m⁻² year⁻¹)**

| C budget | 64ºN | 61ºN | 57ºN | 56ºN |
|---|---|---|---|---|
| Net ecosystem productivity | 63 (7) | 90 (10) | 170 (19) | 237 (21) |
| Leaching | 0.8 (0.07) | 0.6 (0.06) | 0.4 (0.1) | 0.3 (0.05) |
| Harvest export | 50 (6) | 81 (9) | 145 (17) | 201 (19) |
| Change in plant | 10 (2) | 12 (1) | 18 (2) | 20.7 (3) |
| Change in soil | 2 (1) | -3.6 (6) | 6.5 (4) | 15 (7) |
| Change in ecosystem | 12.2 (1) | 8.4 (1) | 24.5 (3) | 35.7 (5) |




**Appendix C: Sensitivity of annual harvested biomass response to the varying fungal organic uptake rates for N and P**

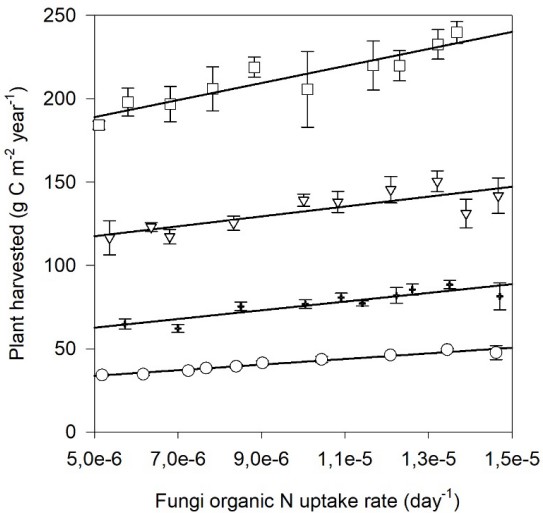

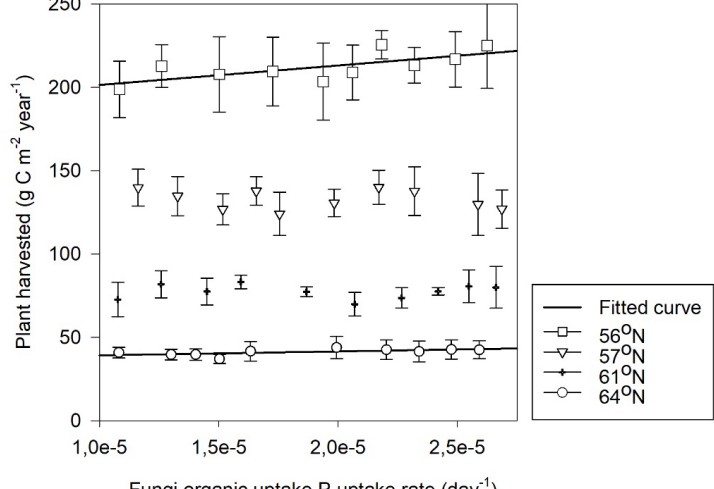


**Fig. C1 Simulated annual mean (symbol) of harvested biomass response to varying fungal organic uptake rates for N and P among the four regions. The bar indicates the standard deviation created by changes in the soil N/P ratio and fungal uptake rates (Table 2)**





*Author contributions.* AG formulated the project, HH conducted the literature review and developed the phosphorus model concepts with AG and PEJ, HH and PEJ implemented the phosphorus cycle code into CoupModel. HH performed the simulations and analysis. HH drafted the manuscript, to which AG and PEJ contributed.

*Competing interests.* The authors declare that they have no conflict of interest.

*Acknowledgements.* The authors express their gratitude to all persons providing data and background information. Gunilla Pihl-Karlsson and Per-Erik Karlsson at IVL Göteborg provided data on the needle P and N contents, soil P data, and regional P deposition data for

the studied regions. Magnus Svensson provided the regional simulation files with CoupModel v4, and Thomas Bosshard at SMHI for the regional climate files. The Department of Aquatic Sciences and Assessment (SLU) contributed P concentrations from open water data, while the Swedish Geological Survey (SGU) provided regional mineral P content data. The Swedish Forest Soil Inventory provided soil C/N data. Cecilia Akeselson collected data for

the SWETHRO project.

The Department of Biological and Environmental Sciences, University of Gothenburg and the strategic research area, along with ModElling the Regional and Global Earth system (MERGE), funded the study. MERGE provided SP project funding for "incorporating phosphorus cycle into ecosystem models". The Formas-funded strong research environment

IMPRESS. Further, we acknowledge the comments from participants of the N-P interaction workshops for the new model concepts funded by MERGE and BECC - Biodiversity and Ecosystem services in a Changing Climate.



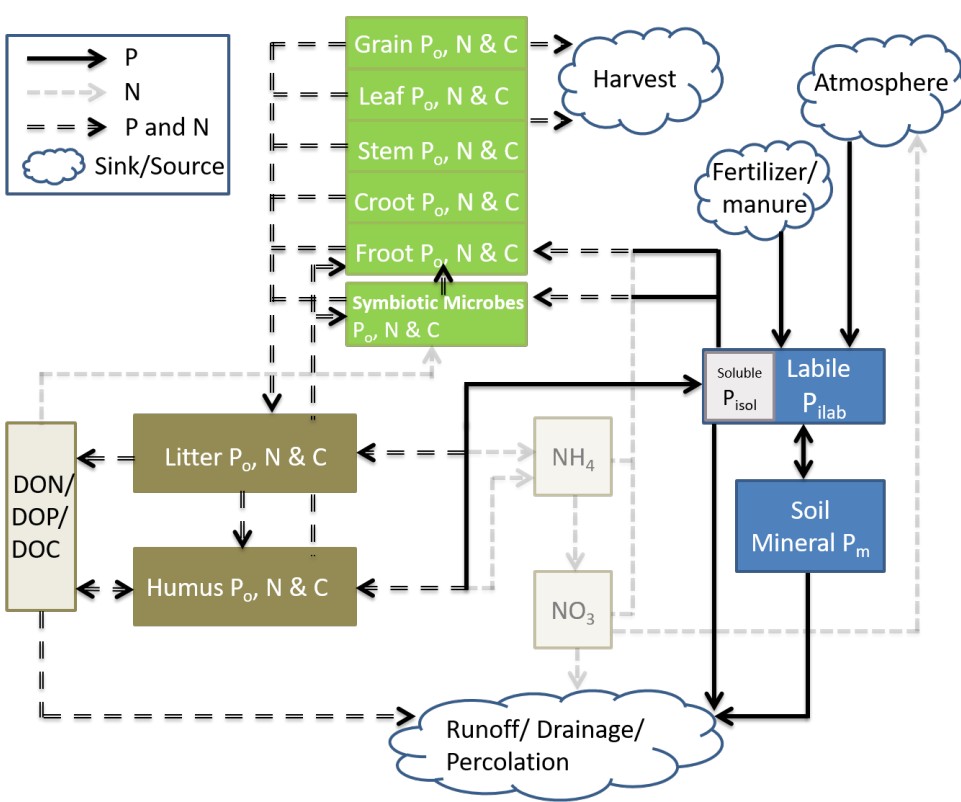


Fig. 1 Conceptual figure of simplified Coup-CNP and its link with the N cycle. Pools in green represent pools of plant-symbiotic microbes (e.g., mycorrhiza fungi), while brown represents soil organic matter, grayish-yellow represents water solutions and blue indicates the soil mineral P pools. Within the pools, Croot stands for coarse root and Froot stands for fine root.




Table 1 Overview of climatic, geological, plant and soil characteristics of the four forest regions

| Regional characteristics | Västerbotten | Dalarna | Jönköping | Skåne |
|---|---|---|---|---|
| Latitude | 64°N | 61°N | 57°N | 56°N |
| Mean annual air temperature (°C)[a] | 0.7 | 3.3 | 5.2 | 7.1 |
| Mean annual precipitation (mm)[a] | 613 | 630 | 712 | 838 |
| Annual N deposition (kg N ha$^{-1}$)[b] | 1.5 | 3.5 | 7.5 | 12.5 |
| Annual P deposition (kg P ha$^{-1}$)[b] | 0.13 | 0.06 | 0.28 | 0.23 |
| Studied soil type FAO (1990) | Podzol | Podzol | Podzol | Podzol |
| Quaternary deposit, SGU [c] | Glacial till | Glacial till | Glacial till | Glacial till |
| Bedrock geology, SGU [c] | Gneiss | Sandstone, Rhyolite | Gneiss | Gneiss |
| Mineral P content of till (mg kg$^{-1}$)[d] | 881 | 428 | 859 | 773 |
| Major tree species Pine/Spruce/broadleaved trees (%)[f] | 45/37/16 | 49/40/9 | 31/54/13 | 12/46/41 |
| Rotation period, year | 120 | 110 | 90 | 70 |
| Thinning conducted at forest age, year (1st /2nd /3rd thinning)[h] | 50/100 | 40/90 | 25/40/70 | 25/40/55 |
| Measured plant biomass at 100 age class (g C m$^{-2}$) [f] | 5371 | 7815 | 10443 | 11501 |
| Soil organic matter C/N (-)[e] | 31.5 | 29.1 | 27.2 | 19.8 |
| Soil organic matter C/P (-)[b] | 494 | 633 | 425 | 425 |
| Soil organic matter N/P (-) | 15.7 | 21.8 | 15.6 | 21.5 |
| Initial Soil C (g C m$^{-2}$) [f] | 7006 | 8567 | 9995 | 10666 |
| Litter C (g C m$^{-2}$) [f] | 350 | 428 | 500 | 533 |
| Humus C (g C m$^{-2}$) [f] | 6655 | 8139 | 9495 | 10133 |
| Initial Soil N (g N m$^{-2}$) | 223 | 295 | 367 | 539 |
| Litter N (g N m$^{-2}$) [f] | 11 | 15 | 18 | 27 |
| Humus N (g N m$^{-2}$) [f] | 212 | 280 | 349 | 512 |
| Initial Soil P (g P m$^{-2}$) | 14.2 | 13.5 | 23.5 | 25.1 |
| Litter P (g P m$^{-2}$) [g] | 0.7 | 0.7 | 1.2 | 1.3 |
| Humus P (g P m$^{-2}$) [g] | 13.5 | 12.8 | 22.3 | 23.8 |
| Soil pH [e] | 5.1 | 5.1 | 5.1 | 4.9 |

[a] 30-year (1961 to 1991) annual average of regional SMHI stations

[b] N and P deposition data and soil organic C/P ratio were obtained from the SWETHTRO project

[c] Geological Survey of Sweden (SGU), https://apps.sgu.se/kartvisare/



[d] according to Geochemical Atlas of Sweden. 2014, measured till samples at C horizon, c.a. 0.8 m below the soil surface

[e] calculated based on the Swedish Forest Soil Inventory data (SFSI. https://www.slu.se/en/Collaborative-Centres-and-Projects/Swedish-Forest-Soil-Inventory/)

[f] Svensson et al. (2008)

[g] assumption that 5% of the total organic pool is litter and 95% is humus, as reported for N in Svensson et al. (2008)

[h] Skogsdata (2012)





Table 2 Parameters with specific values for the different region

| Region | Humus decomposition rate, $k_h$ (day$^{-1}$)[a] | Fungal organic P uptake rate from humus pool through biochemical mineralization (day$^{-1}$)[b] | Fungal organic N uptake rate from humus pool (day$^{-1}$)[a] |
|---|---|---|---|
| Västerbotten 64°N | 0.00048 | $1.5\times10^{-5}$ | $1.5\times10^{-5}$ |
| Dalarnas 61°N | 0.00042 | $2.75\times10^{-5}$ | $1.2\times10^{-5}$ |
| Jönköpings 57°N | 0.0004 | $1.0\times10^{-5}$ | $1.0\times10^{-5}$ |
| Skåne 56°N | 0.00038 | $1.5\times10^{-5}$ | $0.5\times10^{-5}$ |

[a]: from He et al. (2018)

[b]: a high fungal P uptake rate was assumed for high soil organic matter C/P ratios




Table 3 Parameters for the P processes with common values of all four studied regions. Note that the same parameter values were applied for tree and understory layers if otherwise not specified

| Symbol | Parameter | Equation | Value | Unit | Reference |
|---|---|---|---|---|---|
| *Soil mineral P processes* | | | | | |
| $k_w$ | Integrated weathering rate | (1) | $8\times10^{-7}$ | day$^{-1}$ | Guidry and Machenzie, (2000); Sverdrup and Warfvinge, (1993) |
| $n_H$ | Weathering pH response coefficient | (4) | 0.27 | - | |
| $pH_{opt}$ | Weathering pH response base coefficient | (4) | 7 | - | |
| $p_{max,ads}$ | Langmuir max sorption capacity | (5) | 0.0002 | g P g soil$^{-1}$ | Adjusted from Wang et al. (2007) |
| $c_{50,ads}$ | Langmuir half saturation coefficient | (5) | $5\times10^{-5}$ | g P m$^{-2}$ | |
| *Photosynthesis* | | | | | |
| $p_{cp,opt}$ | C/P optimal (leaf) | (8) | 250 | gC gP$^{-1}$ | Thelin et al. (1998; 2002) |
| $p_{cp,th}$ | C/P threshold (leaf) | (8) | 600 | gC gP$^{-1}$ | |
| *Fungal uptake N and P processes* | | | | | |
| $p_{avail}$ | P availability reduce C allocation coefficient | (9) | 0.0009 | - | Assumed |
| $p_{fopt}$ | The optimum ratio between C allocation between fungi and root | (11) | 0.22 | - | He et al. (2018); Orwin et al. (2011) |
| $k_{rm}$ | Respiration coefficient of fungi | | 0.01 | day$^{-1}$ | |
| $p_{lrate}$ | Fungi litterfall rate | (15) | 0.0045 | day$^{-1}$ | |
| $n_{avail}$ | N availability reduce C allcation coefficient | | 0.00039 | - | |
| $p_{i,rate}$ | Potential unit fungal mycelia uptake rate PO$_4$ | (19) | 0.0001 | g P g C$^{-1}$ m$^{-2}$ day$^{-1}$ | Smith and Read, (2008) |
| $n_{NH4rate}$ / $n_{NO3rate}$ | Potential unit fungal mycelia uptake rate NH$_4$/NO$_3$ | | 0.0004 | g N g C$^{-1}$ m$^{-2}$ day$^{-1}$ | He et al. (2018) |
| $n_{olit,rate}$ / $n_{ohum,rate}$ | Potential unit fungal mycelia uptake rate organic N, | | 0.00002 | g N g C$^{-1}$ m$^{-2}$ day$^{-1}$ | |
| $p_{cpfungimax}$ | Fungi maximum C/P | (17) | 200 | gC gP$^{-1}$ | Wallander et al. (2003); Zhang and Elser, (2017) |
| $p_{iavail}$ | Maximum PO$_4$ uptake fraction for roots | (21) | 0.008 | - | |
| $p_{cpfungimin}$ | Fungi minimum C/P | (22) | 100 | - | |
| $p_{olit,rate}$ / $p_{ohum,rate}$ | Potential unit fungal mycelia uptake rate organic P | (23) | 0.00002 | g P g C$^{-1}$ m$^{-2}$ day$^{-1}$ | Assumed to be the same as N |
| *Soil organic P processes* | | | | | |
| $cp_m$ | C/P of non symbiotic microbes | (A.3) | 350 | gC gP$^{-1}$ | Manzoni et al. |





| | | | | | (2010) |
|---|---|---|---|---|---|
| *Uptake demand of P* | | | | | |
| $cp_{leaf, min}$ | Minimum C/P (leaf) | (A.9) | 220 | - | Bell et al. (2014); Tang et al. (2018) |
| $cp_{stem, min}$ /$cp_{croot, min}$ | Minimum C/P, for stem and coarse roots | (A.9) | 4000/ 800 | - | |
| $cp_{root, min}$ | Minimum C/P ratio (fine roots) | (A.9) | 400 | - | |
| *Plant Litterfall processes* | | | | | |
| | Leaf litterfall rate for understory | | 0.0015 | $day^{-1}$ | Calibrated |
| *Plant surface cover* | | | | | |
| | Surface maximum canopy cover, forest | | 0.8 | $m^2\ m^{-2}$ | Assumed |
| | Surface maximum canopy cover, understory | | 1 | $m^2\ m^{-2}$ | Assumed |
| *Erosion* | | | | | |
| $P_{base}$ | P concentration scaling coefficient for surface erosion 1 | (A.14) | $2.7 \times 10^{-6}$ | $mg\ l^{-1}$ | Assumed |
| $P_{\Delta}$ | P concentration scaling coefficient for surface erosion 2 | (A.14) | $7 \times 10^{-6}$ | $mg\ l^{-1}$ | |
| $q_{thr}$ | Critical surface flow rate for erosion | (A.14) | 10 | $mm\ day^{-1}$ | |



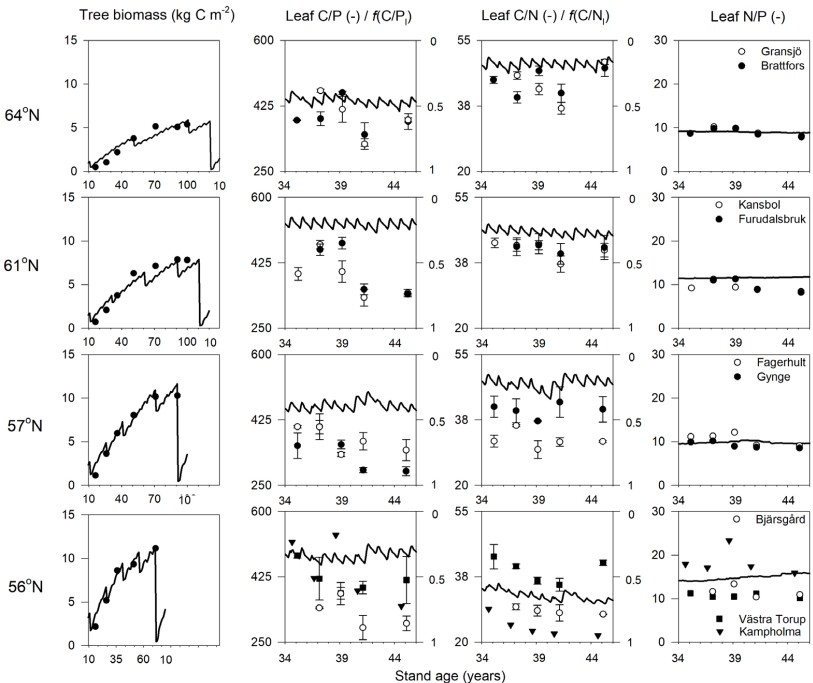


Fig. 2 Simulated (lines) and measured (symbols) plant biomass and leaf C/P, C/N and N/P ratios over the rotation period across the four regions. The x-axis is the stand age in years. The right axis in leaf C/P and C/N figure shows the minimum (*f*(nutrient)=0) and optimum (*f*(nutrient)=1) responses to gross primary production (GPP), respectively. Biomass data and leaf nutrient data were from SFI (SLU, 2003) and the SWETHTRO project (Pihl Karlsson et al., 2015)





Table 4 Summary of the plant-fungal internal C, N and P variables (shown as average values over the rotations period) of the simulated forest ecosystems. Bold values indicate a limiting response for GPP, according to the Liebig's law of minimum. The scale of response for GPP including temperature, water, N and P ranges from 0 (meaning none assimilation) to 1 (meaning optimal growth conditions).

| Variable | Unit | 64ºN | 61 ºN | 57 ºN | 56 ºN |
|---|---|---|---|---|---|
| Net primary production, tree layer | g C m$^{-2}$ year$^{-1}$ | 205 | 302 | 486 | 600 |
| Radiation adsorbed, tree layer | ×10$^6$ J m$^{-2}$ day$^{-1}$ | 3.89 | 5.35 | 6.50 | 6.57 |
| Temperature response for GPP, tree layer | - | 0.47 | 0.52 | 0.63 | 0.67 |
| Water response for GPP, tree layer | - | 0.45 | 0.50 | 0.63 | 0.65 |
| Response N for GPP, tree layer | - | **0.22** | 0.45 | **0.30** | 0.80 |
| Response P for GPP, tree layer | - | 0.56 | **0.23** | 0.34 | **0.33** |
| Total plant uptake, N | g N m$^{-2}$ year$^{-1}$ | 3.67 | 5.76 | 9.00 | 13.8 |
| Total plant uptake, P | g P m$^{-2}$ year$^{-1}$ | 0.42 | 0.49 | 0.87 | 1.08 |
| Organic N uptake fraction (of total) | - | 0.34 | 0.21 | 0.17 | 0.05 |
| Organic P uptake fraction (of total) | - | 0.14 | 0.23 | 0.10 | 0.14 |
| Fungal N uptake fraction (of total) | - | 0.68 | 0.69 | 0.66 | 0.65 |
| Fungal P uptake fraction (of total) | - | 0.56 | 0.57 | 0.56 | 0.56 |
| Fungal N transfer to plant (of total) | - | 0.31 | 0.31 | 0.33 | 0.34 |
| Fungal P transfer to plant (of total) | - | 0.44 | 0.43 | 0.43 | 0.43 |
| Total plant litter, N | g N m$^{-2}$ year$^{-1}$ | 3.50 | 5.47 | 8.61 | 13.2 |
| Total plant litter, P | g P m$^{-2}$ year$^{-1}$ | 0.40 | 0.47 | 0.82 | 1.02 |
| Fungi N litter (of total plant litter) | - | 0.38 | 0.40 | 0.35 | 0.33 |
| Fungi P litter (of total plant litter) | - | 0.13 | 0.15 | 0.14 | 0.13 |





Table 5 Simulated annual average soil C changes (g C m$^{-2}$ year$^{-1}$, positive mean
sequestration, negative mean losses), with comparisons to previous studies. Values in
parentheses indicate uncertainties due to certain model parameters.

| Studies | Approach | 64°N | 61°N | 57°N | 56°N |
|---|---|---|---|---|---|
| Svensson et al. (2008) | Coup-CN implicit mycorrhiza | -5 | -2 | 9 | 23 |
| He et al. (2018) | Coup-CN implicit mycorrhiza | -6 (10) | -5 (11) | 3 (13) | 13 (13) |
| | Coup-CN explicit mycorrhiza | -8 (11) | -9 (12) | -5 (15) | -1 (19) |
| This study | Coup-CNP explicit mycorrhiza | 2 | -2 | 9 | 15 |





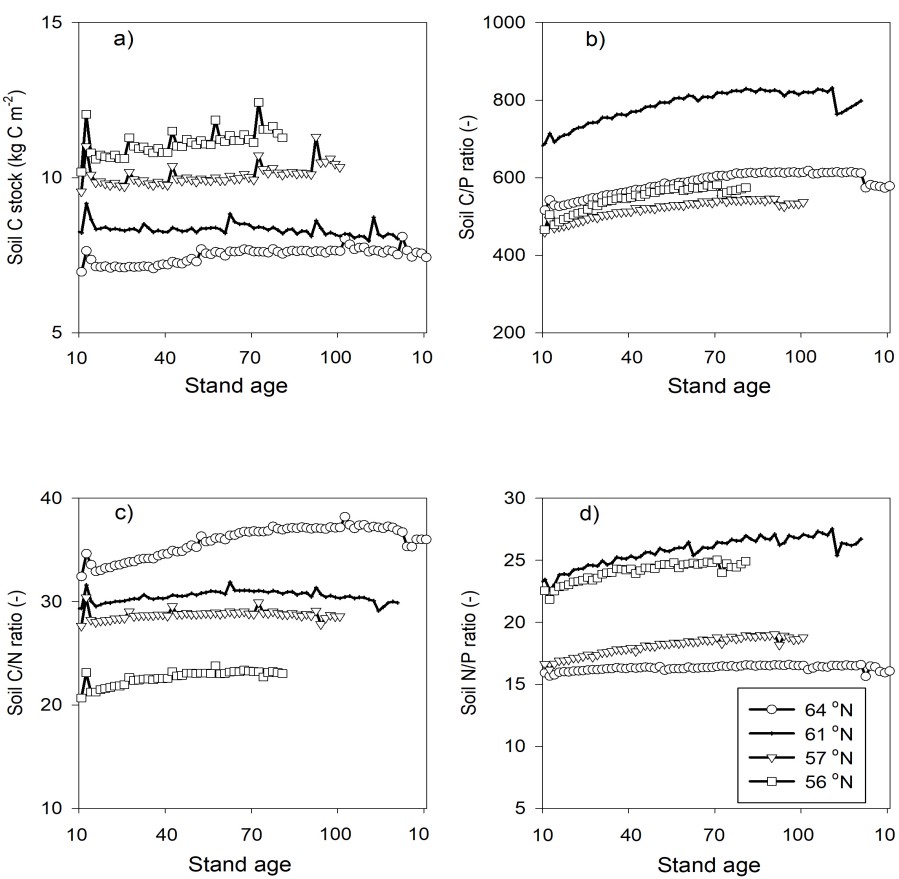


Fig 3. Simulated a) soil C stocks, b) soil C/P, c) soil C/N and d) soil N/P ratios over the
rotation period from relative age 10 to 10 years after the final harvest. Rotation period
increases from South to North Sweden and the small peaks in soil C were related to forest
operations, which were more frequent in southern Sweden. At all latitudes, a clearance at
year 10 was conducted. Thinnings varied from four in southern to two in northern Sweden.





Table 6 Simulated and measured annual P losses through leaching. Note that TP measured is
more than the simulated model fraction DOP and PO4 due to the presence of particulate
phosphorus

| P leaching | 64°N | 61°N | 57°N | 56°N |
|---|---|---|---|---|
| Annual regional total P leaching, measured (kg P ha$^{-1}$) | 0.04 | 0.02 | 0.09 | 0.08 |
| Annual regional total P leaching, simulated (kg P ha$^{-1}$) | 0.03 | 0.01 | 0.05 | 0.07 |
| Average TP concentration, measured (mg l$^{-1}$) | 0.0067 | 0.0066 | 0.03 | 0.02 |
| Average PO$_4$+DOP concentration, Simulated (mg l$^{-1}$) | 0.0056 | 0.002 | 0.003 | 0.006 |
| The fraction of dissolved organic P of total leaching, measured | 63% | 64% | 83% | 61% |
| The fraction of dissolved organic P of total leaching, Simulated | 56% | 74% | 15% | 12% |



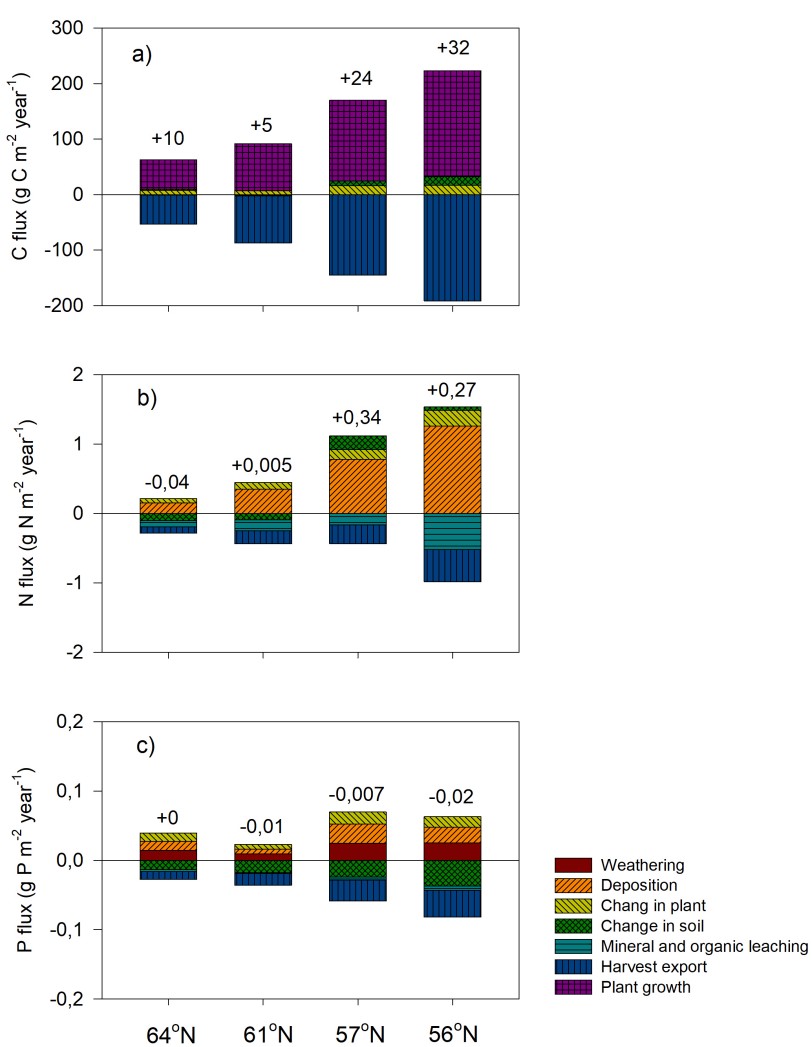

Fig. 4 Simulated annual mean major fluxes in a) C, b) N and c) P in the four regions. The numbers above the stacks indicate the annual mean change in the ecosystem. Note the simulation period starts from year 10 and ends 10 years after final felling.



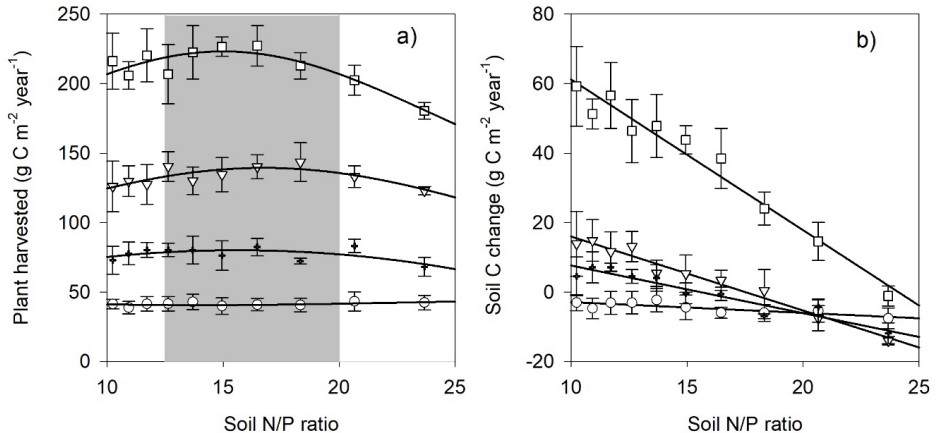

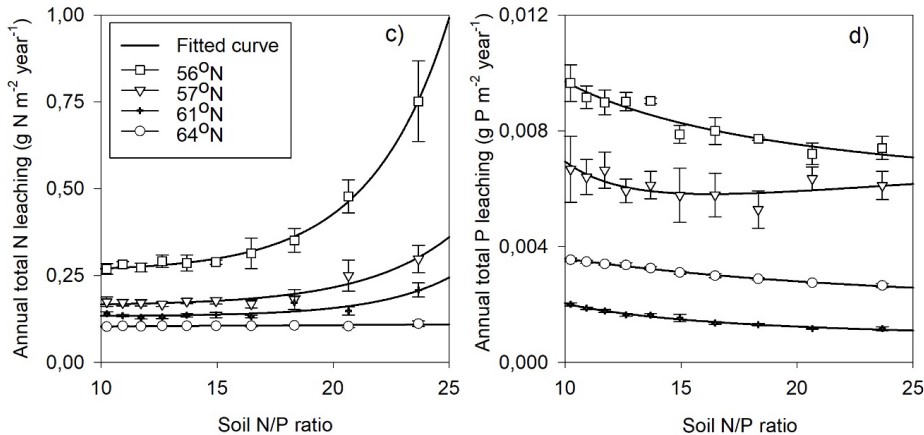

Fig. 5 Simulated annual mean (symbol) a) harvested biomass, b) soil C change (positive
        mean sequestration, negative mean losses), c) total N leaching and d) total P leaching
        response to changing soil organic N/P ratio in the four regions. The bar indicates standard
        deviation created by changes in fungal uptake rates of N and P (see Table 2).




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
