# Peer review of "CoupModel (v6.0): an ecosystem model for coupled phosphorus, nitrogen, and carbon dynamics – evaluated against empirical data from a climatic and fertility gradient in Sweden"

_Geoscientific Model Development, 2020_

## Referee Comment (RC1) · Anonymous Referee #1 · 27 May 2020

The paper by He et al. presented the integration of the phosphorus (P) cycle into the CoupModel and evaluation of the new model, Coup-CNP, against four regions in Sweden that differ in climate and fertility. It is overall a very interesting paper, particularly with the novel setup of both NP cycles and mycorrhiza. The results are well presented, and the discussion is clear and well organized; the authors have put much effort into compiling the information of the model development both in the paper and appendix. Overall I think the paper is in good shape and contributes to advances in the modeling community.

[Figure]

However, the current quality of the paper needs to be improved before final acceptance. First of all, I found the quality of English writing an obstacle for me to keep focusing on the scientific content of the paper. I would recommend the authors to go for a professional editing service with the paper. I listed some obvious mistakes in the detailed comments, mostly before the results section, since I stopped to do that for the rest of the paper simply due to the heavy load of scientific information. Secondly, this study lacks a proper sensitivity analysis. The authors did a simple sensitivity test on the fungal organic uptake rates of N and P and presented the result in the appendix. As far as I see from the description in this paper, Coup-CNP is a heavily parameterized model with a huge number of parameters. It is extremely important to run a proper sensitivity analysis with multiple parameters, not only to see the effects of parameterization on model outputs but also to test the stability and robustness of the model. Thirdly, apart from a sensitivity test, I would also recommend the authors to conduct a few model experiments to see the model responses to alternative model assumptions or changing climatic/environmental conditions. For example, the authors introduced the plant growth response to P stress based on leaf C/P ratio (Eq.9), which is novel and interesting but at the same time debatable. I personally would really like to see the effect of this mechanism on the predicted GPP/NPP and biomass. Another example is the role of mycorrhiza uptake and the so-called organic uptake of N and P. I found that the authors made some very strong assumptions regarding the uptake competition (the sequence of uptake) between plant, fungal and adsorption, and it would be interesting (and fundamental) to see the effect of these strong assumptions.

Detailed comments

1. Abstract and Introduction

Line 18: make "which explicitly consider mycorrhizal interactions" a relative clause after "The extended Coup-CNP"

Line 26: what is "a steady state in P availability"? I don't find "P availability" from the P

budget

Line 40: "nutrient cycling" is not a biochemical reaction

Line 50: it is true that N inputs to the atmosphere increased due to human activity, but for terrestrial ecosystems, the important process is the N deposited from the atmosphere

Line 52: mechanisms can not be amplified, right?

Second paragraph: I think it is a brilliant idea to review the literature of the P cycle in current models, but the organization of information needs to be much improved in this paragraph. I also have some disagreements with the authors about the interpretations of some cited publications, and would like to discuss with the authors about them.

Line 56-65: I think this part is irrelevant to the overall discussion and conclusion of this study. I would recommend to remove or to shorten it.

Line 72: there are some more P-enabled ESMs, e.g. Zhu et al. 2016 Biogeosciences, Goll et al. 2017 GMD, Thum et al. 2019 GMD.

Line 75: Zaehle et al. 2014 does not support your statement here

Line 76-92: The interpretation of these studies is a bit imprecise and vague. I found it difficult to jump from one study to another one; maybe it is better to reorganize all the studies with some intrinsic links, such as common problems or findings. What I will recommend is to focus on the role and effect of plant P uptake in different model studies.

Yu et al. only included the P cycle into the ForSAFE model. I would not phrase it as "developed the model", which causes confusion

Line 99: whose interaction with soil mycorrhizal fungi?

Line 100: I don't fully agree with the interpretation of the references here. These datadriven meta-analyses do not really explain "how P availability affects plant growth", and if this mechanism is influenced by mycorrhizae-plant interactions. They are more of "a proof" than "an explanation" to me

Line 109-111: please restructure the sentence

Line 132: soil organic matter is a more commonly used term than "soil organics"

Line 134: there is little evidence for organic P uptake of plants and microbes, as far as I know

2. Model structure and description of processes linked to the phosphorus pool

Please rename the title, maybe "Model structure and phosphorus process description"? Another piece of advice is linking the process description of Section 2.2 with the equation number in Section 3 and Appendix A. It is much easier for the readers to track information in this way.

Line 142: what does "flexible" model mean?

Line 145: please check the grammar

Line 147: maybe already mention the normal time step and the smaller time step here?

Line 149: "crucial"??? what and why?

Line 151: why the radiation forcing has to be "global"???

Line 153: compete for light??? Not "light interception"?

Line 161: strange sentence structure, please consider adjusting it

Line 164: "can differ" => differs, or do you mean that there are two options for time step???

Line 166: difficult to understand the sentence

Line 171: there is not a common definition of "mineral P", please distinguish it from

other inorganic P forms

Line 174: "inorganic-phosphorus", why a hyphen here?

Line 176-180: I don't see the connection between the model definition and Hedley fractionation. Please elaborate.

Line 183: "which contains"=> "for"

Line 185: which decomposition rate is used for the combined litter pool?

Line 185-200: If I understand correctly, Coup-CNP applied a three-pool structure for soil inorganic P, which is different from most other P models. One thing that is particularly different in this study is that the role of adsorption/desorption is greatly neglected by most biochemical processes since Pisol is only relevant to transport and Pilab is relevant for other processes, such as deposition, weathering, plant/fungal uptake and etc.. It is a very interesting setup, but I think it needs to be better explained.

Particularly, the statement that "These Pi ions are normally loosely adsorbed to surfaces and can thus easily re-enter the Pilab pool through the desorption process (McGechan and Lewis, 2002)." is wrong. There is plenty of evidence for the strong adsorption of phosphate, which is also the main reason for the extremely low soluble inorganic P concentration in the soil water. The main reason that plant and microbe can take up enough P in such a low P concentration is probably the fast replenishing of soluble P in soil water, which are the consequences of desorption/diffusion and biological mobilization (mineralization). Please see Buenemann et al. 2016, SBB and Pistocchi et al. 2018, SBB, and the references therein for more information.

Line 214: what is mobile P and N? this is a very strong assumption that plants can capture nutrients from litterfall, and I wonder how sensitive are the model outputs to this assumption.

Line 221: what are the enzymatic processes? Please be specific. Btw, phosphatase is not a process

Line 222-225: well, this is another astonishing assumption, which needs to be properly tested. And the hidden hypothesis that it only occurs after inorganic P uptake when plant P demand is not fully met is also quite strong from my personal feeling. It basically means that there are no interactions (feedback/competition) between soil organic and inorganic P cycling processes, all the feedback mechanisms have to go through the plant growth&litterfall pathway. I wonder how the model will perform in an extremely P limited ecosystem.

Line 229: how is the DOM redistributed between layers? Is it described in the paper?

3. Equations describing key phosphorus processes/fluxes and their parameterization

One major trouble to me is that the use of both uppercase and lowercase P (p) in the equations. It is extremely difficult sometimes, please consider replacing one of them with another letter. Another major issue is that I could not find information on how leaf P content is calculated, which is essential to understand some results

Line 243: judging from Eq.4, I don't think "proportional" is the right word here

Line 247: how does erosion affect weathering rate? I cannot find it in the paper

Line 254: there is a potential problem that diffusion is also considered as weathering. how uncertain is it to assume diffusion and weathering has the same temperature response? This is even a bigger problem for pH response as there is no evidence that pH affects diffusion

Line 295: I am not sure if this theory is applicable to leaf CP ratio since P is not as essential as N for photosynthesis and the role of leaf P in photosynthesis is not well understood yet. As I mentioned before, it will be interesting to conduct model experiments to test this theory. Additionally, I did not find the information on how Coup-CNP calculates leaf P content.

Line 303: The mycorrhiza module??? This sentence is confusing to me

Line 314: Eq.9 seems the only place that soluble P concentration is used except leaching, how realistic is it to take this assumption directly from N, given the fact that P concentration is much lower than N?

Line 316: "wais" => was

Line 317: the piavail is another very problematic assumption, and I cannot find any theory or evidence to support it. Since the soluble P concentration is not used to calculate the plant P uptake, I could foresee that if labile P is freely taken up by the plant, the model might end up with no P limitation and the labile P might get depleted very soon. If there is no theory or literature to support this parameter, at least it should be tested in the sensitivity analysis

Line 405: where is f(Piavail) used? which equation?

4. Description of the region used for simulation and model setup

It seems the same study regions have been tested with the previous version of Coup-Model before, and it is unclear from this section if the new Coup-CNP model is recalibrated in this study. Please state it clearly in the paper how the model is parametrized and why some parameter values differ from previous studies (I assume that is the case)

Line 420: kg N ha-1 yr-1, right?

Line 423: please cite the most recent FAO standard

Line 443-446: difficult to follow the sentence

Line 449-450: The model was spun up for 10 years, and then a clear cut is simulated??? How do you determine the initial SOM content and soil stoichiometry? How big are the effects of initial SOM status?

Line 450-452: difficult to follow. Unclear to me what are the plant components and how are they treated

Line 470: "chronicle"? I am not sure that is the right word here???

Line 477: This is a very unrealistic assumption; please see Yu et al. 2020 GMD

Line 485-487: One specific question to Table 2 is that, why the decomposition and uptake rates for different latitudes are different, given that the temperature response function already accounts for the difference in temperature? If they are calibrated separately, what is the meaning for choosing a climatic gradient???

Table 3: I would recommend running a full sensitivity test with parameters in this table

Line 519-521: difficult to follow the sentence

5. Results

Line 561: confusing, please rephrase

Line 564: why the new Coup-CNP C sequestration rates are so different from previous studies of the same regions?

Line 573-575: I only see that the P leaching is very small, which may infer that it has a small effect. But the fact that P leaching accounts for 30% of P deposition does not lead to the conclusion that "a small effect on the system". I guess the key point here is that both P deposition and P loss are very small compared to other fluxes, e.g. plant P uptake

Section 5.2: the rotation period, 10 years to 10 years after the final felling, makes it a bit difficult to understand the results in figure 4, particularly the plant growth and change in plant in panel A. For me it is very difficult to judge how much of the changes in plant and soil pools are due to the very short spin-up time (10 years)? Is it possible to run a real spin-up to ensure a more stable state of the soil pool? Also, I did not fully understand why the pool size of 10-year-old trees differ so much in N and P size, to me it seems to be the effect of model initialization and spin-up.

6. Discussion

Section 6.1: all the studies that are compared to in the section are modeling studies, which should be made very clear.

Section 6.2:

In general, the discussion is interesting and the findings are encouraging. However, I do have an understanding problem regarding the soil N/P ratio. From the description in the method part, the soil N/P ratio seems to be a parameter in the sensitivity analysis. But its value is not reported in Table 3, and it seems that it is also not a constant value from Figure 3d.

A more methodological problem is, only three parameters were tested in the sensitivity analysis, and the result for one parameter was presented. How could one conclude that this one is the most important parameter for the ecosystem? As I mentioned before, if this is the first study of the Coup-CNP, a better-designed sensitivity test should be performed. I am very convinced by the authors that soil N/P is an important indicator of Swedish forests, but I am convinced by the way it was accidentally chosen in this study.

Line 676: where does this conclusion come from? increasingly P limited with time or latitude, or another gradient?

Line 682: have you checked if the threshold is the same for pine and spruce? if not, please be specific about tree species
* * *

---

## Referee Comment (RC2) · Anonymous Referee #2 · 4 Aug 2020

The paper of He et al., brings us a model that couples P into an existing CN model. It is an interesting study with special focus on mycorrhizal fungi, which is important in P dynamics, but has yet to be adequately represented in current literature. My major concern though, is that the model is heavily parameterised with great details and many parameters, but the model performance is systematically biased. Figure 2 and Table 4 evaluated modelled tree biomass, leaf C:N, leaf C:P, leaf N:P and P leaching against measurements. First of all, for a model that covers many aspects of C, N, P dynamics, variables evaluated here are not adequate to show the model performance. Secondly,

the model systematically overestimate leaf C:P (all sites) and leaf C:N (3 out of 4 sites), and underestimate P leaching (all sites). I am not convinced that the model does a good job in capturing the system. Additional work and data are needed to understand the model dynamics and thoroughly assess the model performance.

In addition, I feel it is quite difficult to follow the model description. Sometimes there are logical issues related to terminology and the separation among system compartments (please see detailed comments below). Sometimes it is due to lack of critical information in P cycling in the main text, for example, P dynamics in vegetation (allocation, resorption etc.), through mineralization etc. It might be better to put part of the information in the appendix into the main text, or at least have some overall description of these processes in the main text and point to the appendix for detailed information. The goal is to give the reader a complete picture of P cycling the model tracks.

The novel part of this model, from my perspective, is related to symbiotic mycorrhizal fungi. I did not find any observations to initialise, evaluate model performance or constrain model parameters related to this part. It is also not clear what is the advantage of incorporating detailed symbiotic mycorrhizal fungi, how it affects system dynamics, what are the novel model behaviours due to this part? I feel these questions are worth answering to persuade the reader that the model is advantageous and worth the great details.

Detailed comments: Before Line 65-70, CMIP6 model results are openly available now. One model (probably the only one) that has land P component is from CSIRO, Australia. The name of the earth system model is ACCESS and land component is CABLE-CNP.

Lines 70-75, whether CNP models from Goll et al., 2012; Wang et al., 2010; Yang et al., 2014 are simplified are context dependent. As far as I know, these models incorporated key processes in C,N,P, water and energy dynamics and take into account coupling and interactions across spatial-temporal scales. They are not necessarily simpler than the

model presented here.

Line 75-80. Models in Medlyn et al 2016 are not earth system models per se. They are process-based vegetation models. ESMs have coupled land, atmosphere, ocean etc. Some models might be used as the land component of some ESMs. Some models may not be directly coupled.

Line 80-85. Low eco2 response do not imply "In other words, the vegetation is rather inflexible to increase P uptake". There are many factors come into play. Without CNP, the models have difficulties in capturing nutrient limitation on CO2 response. In nutrient limited locations, nutrient limitation is likely to reduce eco2 responses. And it is not only about the uptake capability. It is also related to nutrient availability.

Line 140-150, "The main model structure is a one-dimensional, vertical layered soil profile including plants." This sentence is confusing. How vertical soil profile could include plants ?

Line 150-155, the concept of "big leaf" model assumes canopy carbon fluxes have the same relative responses to the environment as any single unshaded leaf in the upper canopy. You have two layers, trees and understory. Normally when people talk about "big leaf" model, it does not simulate light competition between up- vs. understory plants.

Line 170-171, the naming convention is quite confusing. By common definition, inorganic P is part of soil mineral P.

Line 180. The description of different P pools is rather confusing. If "soil mineral P is the total soil P without organic Po and labile P", how could you estimate it with total P content and bulk density. When we measure bulk density, we do not exclude the contribution from the organic matter.

Line 180-185. What do you mean by "fresh plant residues"? If plant residue that stays above soil, but it is not fresh (e.g., it is from the last year), do you exclude it from litter?

Line 180-185, "In CoupModel, soil litter could be further divided into two litter pools: one which contains readily decomposing materials (e.g., plant leaves and fine roots) and another for decomposition-resistant litter (e.g.,stems and coarse roots)". If you do not represent these in your model, please skip these texts to reduce confusion.

Line 190-195. Do you take into account the hysteresis in P adsorption/desorption?

Line 170-205, you talked about litter pool, how do you treat soil organic matter/P pool? Do you only have humus pool? If so, non-symbiotic soil microbes are classified as litter in your model?

Lines 210-215, "During certain seasons, plants can also capture mobile P (as well as mobile N) to prepare for rapid growth in the spring". What do you mean here? You mean plants take up more P in other seasons other than Spring, store it and use it in Spring? How does it occur? What do you mean by mobile P(N)?

Lines 220-225, I don't understand what do you mean by"In Coup-CNP, biochemical mineralization is defined as organic uptake". Biochemical mineralization and organic uptake are different processes.

Line 316, "wais" to "was"

Line 535 – 540 and Figure 2. From Figure 2, the model systematically over-estimate Leaf C/P and leaf C/N ratio (except one site). Is it because an over-estimation of the leaf biomass? If there are coherent bias for all or most sites, it is not a neglectable issue.

Figure 4. Why do you plot plant growth in C flux but change in plant for P flux, please be coherent and consistent.

Table 6, systematically underestimation of P leaching

---

## Author Comment (AC1) · 30 Sep 2020

**Final response to interactive discussion**

Dear Referees, Dear Editor,

We would like to thank you very much for your positive comments and constructive suggestions to our manuscript "CoupModel (v6.0): an ecosystem model for coupled phosphorus, nitrogen and carbon dynamics – evaluated against empirical data from a climatic and fertility gradient in Sweden"

In this document, we provide responses to each comment of the two referees; the referee comments are given in normal font, and our responses in italics.

One important change in the revised version is that we will additionally evaluate the sensitivities of all the newly introduced Coup-CNP parameters by a Monte Carlo sensitivity analysis method. The covariance between parameters and equifinality for the selected model outputs will be presented (compared to three key parameters sensitivity test in current version). The sensitivity analysis enables us to address the concerns raised by referee#1 about the global sensitivity and assumptions, and also the model behavior issue raised by referee#2.

Further, we will improve the introduction and the presentation of model description by merging the process description of Section 2.2 with the equation number in Section 3 and linking to equations numbers in Appendix A as suggested by referees.

Below, we state how we will address each specific comment when revising our manuscript. After revising, we will send the manuscript for language checking by professional English language proofing company. Hence, some formulations below may change.

We hope that our response together with the revision of the manuscript sufficiently addresses the referees' concerns.

Sincerely,

Hongxing He (on behalf of the author team)

**Reply to Anonymous Referee #1**

We thank Anonymous Referee #1 for your positive comments and constructive suggestions to our manuscript. Here are our responses to the comments. The referee comments are given in normal font and our response in italics.

The paper by He et al. presented the integration of the phosphorus (P) cycle into the CoupModel and evaluation of the new model, Coup-CNP, against four regions in Sweden that differ in climate and fertility. It is overall a very interesting paper, particularly with the novel setup of both NP cycles and mycorrhiza. The results are well presented, and the discussion is clear and well organized; the authors have put much effort into compiling the information of the model development both in the paper and appendix. Overall I think the paper is in good shape and contributes to advances in the modeling community.

- *We are glad the referee appreciates the science and novelty value of this new model development. We appraise that our P cycle and mycorrhiza model has a good potential to contribute to improved understanding of and insights into current ongoing discussion of nutrients impacts on C cycle. We also pleased that the referee finds the presentation of the results and discussions are in a good shape.*

However, the current quality of the paper needs to be improved before final acceptance. First of all, I found the quality of English writing an obstacle for me to keep focusing on the scientific content of the paper. I would recommend the authors to go for a professional editing service with the paper. I listed some obvious mistakes in the detailed comments, mostly before the results section, since I stopped to do that for the rest of the paper simply due to the heavy load of scientific information.

- *This time we will send the manuscript for language checking by professional English language proofing company after revision is completed. The language of the manuscript had previously been edited by professional editing service, but that was not the final, submitted version.*

Secondly, this study lacks a proper sensitivity analysis. The authors did a simple sensitivity test on the fungal organic uptake rates of N and P and presented the result in the appendix. As far as I see from the description in this paper, Coup-CNP is a heavily parameterized model with a huge number of parameters. It is extremely important to run a proper sensitivity analysis with multiple parameters, not only to see the effects of parameterization on model outputs but also to test the stability and robustness of the model.

- *We agree a more comprehensive sensitivity test including all the P parameters and few N related parameter could be a good addition. We will conduct a more thoroughly sensitivity test of all the new introduced parameters by a Monte Carlo based sensitivity analysis of all the newly introduced P parameters in the revised paper.*

Thirdly, apart from a sensitivity test, I would also recommend the authors to conduct a few model experiments to see the model responses to alternative model assumptions or changing climatic/environmental conditions. For example, the authors introduced the plant growth

response to P stress based on leaf C/P ratio (Eq.9), which is novel and interesting but at the same time debatable. I personally would really like to see the effect of this mechanism on the predicted GPP/NPP and biomass. Another example is the role of mycorrhiza uptake and the so-called organic uptake of N and P. I found that the authors made some very strong assumptions regarding the uptake competition (the sequence of uptake) between plant, fungal and adsorption, and it would be interesting (and fundamental) to see the effect of these strong assumptions.

- *We agree. These are interesting implications of our new model. Our aim is to be able to test state-of-art theory on the importance of fungal and plant uptake with our model. In the revision, we will better backup assumptions with literature and carry out a sensitivity analysis including parameters relevant for the assumptions, e.g., C/P optimal (leaf) $p_{cp,opt}$, C/P threshold(leaf) $p_{cp,th}$ (Table 3) and fungi uptake P competition parameters, e.g. potential unit fungal mycelia uptake rate PO4, $p_{i,rate}$ (Table 3). We will further discuss how these parameters link to the modelled outcome, e.g. plant biomass, predicated GPP/NPP.*
*Please note the changing climatic/environmental model experiments were already conducted by applying the new Coup-CNP model on four regions with varying climate (annual temperature between 0.7-7.1 C) and fertility (with the soil C/N and C/P ratios between 19.8-31.5 and 425-633, respectively) along a gradient from South to North Sweden.*

Detailed comments

1.  Abstract and Introduction

Line18: make "which explicitly consider mycorrhizal interactions" a relative clause after "The extended Coup-CNP"

- *We will revise this accordingly.*

Line 26: what is "a steady state in P availability"? I don't find "P availability" from the P budget

- *We will revise the texts to make it clear. A steady state in P availability means the outflow of P fluxes including plant uptake, leaching losses, harvest etc is more or less balanced by the inflow of P fluxes including weathering, deposition and mineralization.*

Line 40: "nutrient cycling" is not a biochemical reaction

- *We agree that nutrient cycling is more than a biochemical reaction. We will rephrase the sentence by replacing 'biochemical reactions' with 'processes' and 'nutrient cycling' with 'decomposition of soil organic matter and nutrient uptake'.*

Line 50: it is true that N inputs to the atmosphere increased due to human activity, but for terrestrial ecosystems, the important process is the N deposited from the atmosphere

- *Agreed. We will revise to "N deposition from the atmosphere"*

Line 52: mechanisms can not be amplified, right? Second paragraph: I think it is a brilliant idea to review the literature of the P cycle in current models, but the organization

of information needs to be much improved in this paragraph. I also have some disagreements with the authors about the interpretations of some cited publications, and would like to discuss with the authors about them.

- *Agreed. We will remove "mechanisms" in the revision. We will shorten the literature P model review to make it more focused and thoroughly recheck the cited publications to ensure they are cited correctly. If the reviewer has one or some particular in mind besides the ones mentioned below, we would appreciate to know which.*

Line 56-65: I think this part is irrelevant to the overall discussion and conclusion of this study. I would recommend to remove or to shorten it.

- *Agreed. We will shorten this part.*

Line 72: there are some more P-enabled ESMs, e.g. Zhu et al. 2016 Biogeosciences, Goll et al. 2017 GMD, Thum et al. 2019 GMD.

- *Thanks for the information. We will add these into the revised paper*

Line 75: Zaehle et al. 2014 does not support your statement here

- *Agreed. Zaehle et al 2014 discussed mainly N not P.*

Line 76-92: The interpretation of these studies is a bit imprecise and vague. I found it difficult to jump from one study to another one; maybe it is better to reorganize all the studies with some intrinsic links, such as common problems or findings. What I will recommend is to focus on the role and effect of plant P uptake in different model studies. Yu et al. only included the P cycle into the ForSAFE model. I would not phrase it as "developed the model", which causes confusion

- *Thanks for the great suggestion. We will update the organization of the literature P model and compare the COUP-CNP approach with those of existing models. We will also reword the ForSAFE model statement.*

Line 99: whose interaction with soil mycorrhizal fungi?

- *Interaction between plants and mycorrhizal fungi, we will rephrase this sentence to make it clear*

Line 100: I don't fully agree with the interpretation of the references here. These data driven meta-analyses do not really explain "how P availability affects plant growth", and if this mechanism is influenced by mycorrhizae-plant interactions. They are more of "a proof" than "an explanation" to me

- *Agreed. We use this evidence here to highlight the importance of fungi for P availability, thus motivate the fungi model development. We will rephrase to make it more clear*

Line 109-111: please restructure the sentence

- *We will restructure the sentence*

Line 132: soil organic matter is a more commonly used term than "soil organics"

- *Agreed.*

Line 134: there is little evidence for organic P uptake of plants and microbes, as far as I know

- *First, we would like to point out that the organic uptake defined in Coup-CNP, differs that of direct uptake of organic P molecules, see the definition of "organic uptake" Line 220-225. The model assumes plant roots and symbiotic fungi bypass the labile $P_{ilab}$ pool, and obtain mineralized $P_i$ directly from the organic matter $P_{olit}$ and $P_{ohum}$ pools. Essentially, the organic uptake in the model is a short cut for plant uptake that bypass the microbe mineralization process.*
  *However, we agree with the referee that the evidence for direct uptake of organic P molecule by plants and microbes are few, the most uptake forms remains to be $P_i$. There are some empirical studies (e.g. Jayachandran et al. 1992; Fox et al. 2010) show mycorrhizal fungi may be able to acquire P from organic sources that are not available directly to the plant (e.g. phytic acid and nucleic acids). Also see, Lindahl et al. (2002) and Johnson and Gehring (2007). Such potential uptake is also included in our organic uptake concepts.*
  *Our model application also showed that organic P uptake of plants are needed to sustain the soil C/P ratios and plant growth demand, particularly for P poor region see Table 4. The modeling study by Orwin et al 2011 conclude the same.*

2. Model structure and description of processes linked to the phosphorus pool

Please rename the title, maybe "Model structure and phosphorus process description"? Another piece of advice is linking the process description of Section 2.2 with the equation number in Section 3 and Appendix A. It is much easier for the readers to track information in this way.

- *We agree with the referee. We will merge part of the section 2 and 3 in the revision to make it easier to follow.*

Line 142: what does "flexible" model mean?

- *CoupModel has a number of modules that can be activated by choice of the model user. We will remove "flexible" to avoid misunderstanding*

Line 145: please check the grammar

- *We will reword the sentence.*

Line147: maybe already mention the normal time step and the smaller time step here?

- *Agreed.*

Line 149: "crucial"??? what and why?

- *Here is a general description of the model, when calculate e.g. event like snow melting peaks which is important to have a short time resolution to water flow*

*estimation to avoid numerical error and water imbalance. We will remove this to avoid confusion.*

Line 151: why the radiation forcing has to be "global"???

- *Global Radiation is used according to accepted meaning as the sum of both diffuse and direct incoming shortwave radiation. We may reword this to "short wave incoming radiation". This is a CoupModel convention, that the global radiation includes both direct and indirect radiation.*

Line 153: compete for light??? Not "light interception"?

- *Yes, we will revise it in the revision.*

Line 161: strange sentence structure, please consider adjusting it

- *We will rephrase the sentence.*

Line 164: "can differ" => differs, or do you mean that there are two options for time step???

- *In CoupModel, it is possible to choose one-time step for water & heat, but another for C, N &P. We will delete this sentence to avoid confusion since the same time step was used in this study.*

Line 166: difficult to understand the sentence

- *We will reword to make it more clear*

Line 171: there is not a common definition of "mineral P", please distinguish it from other inorganic P forms

- *We realize the definition of mineral might be unclear. We therefore will re-describe this state variable and rename it to distinguish to the other inorganic forms. In the model, soil inorganic phosphorus can be divided into labile inorganic P (Pi, phosphate ions, e.g., H3PO4, H2PO4-, HPO42-, PO43-) and **soil solid mineral**, (Pm). Soluble (Pisol) are a part of labile (Pilab) that are dissolved and not adsorbed. **Soil solid mineral Pm** is a lumped state variable containing primary and secondary solid P compounds.*

Line 174: "inorganic-phosphorus", why a hyphen here?

- *We will remove the hyphen in the revision*

Line 176-180: I don't see the connection between the model definition and Hedley fractionation. Please elaborate.

- *The defined inorganic phosphorus forms in the model was partly adopted from the Hedley fraction, e.g. by using different extraction method for soluble and labile P. We will delete the Hedley fraction sentence in the revised paper to avoid confusion.*

Line 183: "which contains"=> "for"

- *Agreed.*

Line 185: which decomposition rate is used for the combined litter pool?

- *The litter decomposition rate coefficient that integrated the readily and more resistant litter was used. The same was used as in Svensson et al. 2008 and He et al. 2018 calibration paper and all CoupModel publications before 2000, when only one litter state variable existed. The litter decomposition coefficient was obtained through the litter bag incubation studies at those or nearby Swedish forests, more details see Table 5, Svensson et al. 2008. We will add this information in the revised paper.*

Line 185-200: If I understand correctly, Coup-CNP applied a three-pool structure for soil inorganic P, which is different from most other P models. One thing that is particularly different in this study is that the role of adsorption/desorption is greatly neglected by most biochemical processes since Pisol is only relevant to transport and Pilab is relevant for other processes, such as deposition, weathering, plant/fungal uptake and etc.. It is a very interesting setup, but I think it needs to be better explained. Particularly, the statement that "These Pi ions are normally loosely adsorbed to surfaces and can thus easily re-enter the Pilab pool through the desorption process (McGechan and Lewis, 2002)." is wrong. There is plenty of evidence for the strong adsorption of phosphate, which is also the main reason for the extremely low soluble inorganic P concentration in the soil water. The main reason that plant and microbe can take up enough P in such a low P concentration is probably the fast replenishing of soluble P in soil water, which are the consequences of desorption/diffusion and biological mobilization (mineralization). Please see Buenemann et al. 2016, SBB and Pistocchi et al. 2018, SBB, and the references therein for more information.

- *We thank the referee for pointing out the potential importance of strong adsorption of phosphate in regulating the Pi availability. However, we have already assumed a strong and instantaneous response by the split between labile and soluble form of mineral P. Our current model does not allow formation of secondary solid minerals. We understand that Coup-CNP use fewer inorganic pools compared to some of the other models (e.g. Wang et al. 2010). Such models have separated primary, secondary mineral P (i.e. sorbed and strongly sorbed P pool in Wang et al paper), also occluded P, thus further introduce fluxes exchange between these pools. Theoretically these might represent a more physical realistic picture of the inorganic P dynamics, and we are aware this could lead to some over-simplifications (see our response to the weathering calculation)*
- *However, in reality very few data exist for the state variable size and the flux rates at the ecosystem level and hence in Coup-CNP, ' soil solid mineral Pm is a lumped pool containing primary and secondary mineral P (and occluded P) ( line 173-174),'and one net weathering flux were used. In addition, we would also like to point out that the fast replenishing of soluble P was the case in current model structure since we assume the instant equilibrium of the total soluble P and the labile P in the soil water.*

Line 214: what is mobile P and N? this is a very strong assumption that plants can capture nutrients from litterfall, and I wonder how sensitive are the model outputs to this assumption.

- *The plant mobile N and P are state variables designed to mimic the nutrient reallocation or retranslocation process, a process by which nutrients (here N and P) were mobilized from senescing structure to developing tissues e.g. before litterfall, thus acting as an important mechanism to reduce dependence on nutrient uptake and increase nutrient recycle. Aerts, (1996) showed mean nutrient resorption efficiency for perennial plant was c.a. 50% for N and slightly higher, 52% for P. A more recent global synthesis of most measured data for woody plants showed similar results that mean N resorption efficiency of 48.4% and mean P resorption efficiency was 53.3% (Yan et al. 2018). Nieminen and Helmisaari (1996) show a 67-88% mobile nutrient N and P decreasing during needle senescence for Scots pine in Finland. These evidence serves the rational of including the nutrient resorption process in the model. In Coup-CNP, P was a similar concept to that of N, this process simulates the resorption of nutrients before litterfall (N and P were assumed to be stored internally in the mobile state variable and these can then be used next year to develop new shoots/leaves). We will describe this more in detail and add this into sensitivity test to evaluate how sensitive the model outs to this.*

Line 221: what are the enzymatic processes? Please be specific. Btw, phosphatase is not a process

- *For example, phosphatase released by the root exudates. We will reword and make it more specific*

Line 222-225: well, this is another astonishing assumption, which needs to be properly tested. And the hidden hypothesis that it only occurs after inorganic P uptake when plant P demand is not fully met is also quite strong from my personal feeling. It basically means that there are no interactions (feedback/competition) between soil organic and inorganic P cycling processes, all the feedback mechanisms have to go through the plant growth & litterfall pathway. I wonder how the model will perform in an extremely P limited ecosystem.

- *We thank the referee for the comments on P uptake. We realize the description may be unclear thus we will revise this to make it clearer. The soil organic and inorganic P cycling processes are highly interacted in current model structure, besides through the plant growth & litterfall pathway, other the feedback mechanisms include , e.g. the decomposition of soil organic pools resulting in phosphate formation (L815-835 equ A2), the soil microbes regulates the mobilization or immobilization of the inorganic P from organic P (L815-835, equ A2); The growth of microbes and the plant organic uptake also compete for inorganic P;  the availability and adsorption also regulates the competition for plant P uptake (L265-277, equ5).*

  *In our model, the P uptake is driven by the demand of the plants but regulated by the availability in the soil. The uptake priority is the mineral $P_i$ then the "organic P". For the mineral $P_i$, the availability is regulated by the availability fraction parameter, $p_{iavail}$. The availability of organic uptake is similarly regulated by a coefficient called*

*nutrient shortcut uptake rate (equ A4, Table 2, called fungi organic uptake coefficient in current version). The organic uptake concept was initially suggested by Beier & Eckersten,1998; Gärdenäs et al. 2003, (which was refined in) Svensson et al. 2008. We do not account for any organic uptake providing that the demand was fully met by the mineral $P_i$ uptake. However, we can reduce the efficiency of uptake by reducing the parameter value for $p_{iavail}$. Our results also indicate the nutrient shortcut uptake coefficient needs to be higher in the P limited ecosystem. These are also why the nutrient shortcut uptake rates are selected as sensitivity analysis. The sensitivity results shown in Fig C1 and table B1, B2, B3 shows the organic uptake clearly impact the plant growth, the P leaching, thus shows that the inorganic uptake is strongly linked to the organic uptake.  But there are no direct interaction from organic uptake mechanism to the mineral uptake in Coup-CNP.*

*The north 61 ᵒN region was a P limited forest ecosystem, the model showed a reasonable agreement with tree biomass although slightly underestimation of growth (probably caused by the overestimation of the leaf C/P, C/N ratios), but one of the noticeable results are the need of higher organic uptake to fulfill the plant demand when P was highly limited (e.g. Table 4).*

Line 229: how is the DOM redistributed between layers? Is it described in the paper?

- *The redistribution is done following that of water flow, as the DOM is assumed to have full mobility with water. The formation of DOM is from litter and humus, but DOM can also be fixation back to humus, see equations (A.7), parameterization following Svensson et al., 2008. We will add description and link to the equation (A.7) into the revision.*

   3.  Equations describing key phosphorus processes/fluxes and their parameterization

One major trouble to me is that the use of both uppercase and lowercase P (p) in the equations. It is extremely difficult sometimes, please consider replacing one of them with another letter. Another major issue is that I could not find information on how leaf P content is calculated, which is essential to understand some results

- *We plan to revise the symbols of the equations where could lead to possible confusion and double check consistency,. wW will also add explanation texts in the main texts to explain the rule of the symbol to make it easier to follow. In general, capital P represents a pool or amount (state variables), small p represents a parameter.*
- *The leaf P content (mostly using C/P ratio as its indicator in the manuscript), leaf C/P ratio were calculated as the ratio between the leaf C (g C m-2) and leaf P (g P m-2), where leaf C and P were calculated separately, and updated for each time step. For each time step, the model calculate the leaf C include the C influx: photosynthesis allocation to leaf, C outflux: leaf respiration, leaf litterfall, etc. For leaf P, the model calculates the P influx: total P uptake allocate to leaf, P outflux: P litterfall, P to internal mobile P, etc. The model updates the leaf C:P ratio at each time step and used to estimate the photosynthesis for next time step. Note that the uptake of P is driven first by the demand of P and second by the availability of P. The demand of P*

*is driven by the C but the availability of P is independent of C. We will add and/or clarify the description in the revised paper.*

Line 243: judging from Eq.4, I don't think "proportional" is the right word here

- *We mean here the flux is proportional to the pH response where the response itself is another equation. We will reword this in the revision.*

Line 247: how does erosion affect weathering rate? I cannot find it in the paper

- *The erosion cause transport of particles that contains P. However, we only simulate a loss of particles (See appendix A12, A13, A14). We will improve the references to appendix by adding equations number in the revised paper to make it clearer.*

Line 254: there is a potential problem that diffusion is also considered as weathering. how uncertain is it to assume diffusion and weathering has the same temperature response? This is even a bigger problem for pH response as there is no evidence that pH affects diffusion

- *First, the weathering in Coup-CNP is independent of the mobile part of P in our model structure. The diffusion/desorption flux was implicitly included into the weathering flux (L196-198). Thus, how the diffusion/desorption flux response to temperature and pH is not explicit considered. However, again our aim was to build a simple yet realistic P net weathering flux. We compared to the current net weathering flux to a more detailed and rigorous geochemical model PROFILE, but not a dynamical model; that is more widely used for weathering estimates and current P flux estimates were rather similar (L635-645). We will revise the weathering part in our model conceptual presentation and add remarks in the equations description to make it clear about our assumptions.*

Line 295: I am not sure if this theory is applicable to leaf CP ratio since P is not as essential as N for photosynthesis and the role of leaf P in photosynthesis is not well understood yet. As I mentioned before, it will be interesting to conduct model experiments to test this theory. Additionally, I did not find the information on how CoupCNP calculates leaf P content.

- *We will include the leaf CP ratio parameters in the Monte Carlo sensitivity analysis to evaluate the C/P ratio impact on the growth of the plants, See response above for the leaf P content calculation.*

Line 303: The mycorrhiza module??? This sentence is confusing to me

- *We remove the sentence "P fungi processes analog to N processes (He et al., 2018) are found in appendix A" as the fungi P processes are described in main text. Hopes that answer the reviewer question as we are not completely sure we got it right.*

Line314: Eq.9 seems the only place that soluble P concentration is used except leaching, how realistic is it to take this assumption directly from N, given the fact that P concentration is much lower than N?

- *We will revise the wording here to clarify, it should be analogue instead of the same. Equ 9 is a response function that account the decreasing of plant C allocation to fungi when inorganic P are high. Mathematically it ranges from 1 to a threshold value. What we mean here is an analogue equation formula is used to describe P as for N but with different coefficients and drivers. The P dependency is defined by the soluble P concentration and the reduction parameter, $p_{avai}$ in equ (9).*
*A number of studies show higher fungal production under more P-limiting conditions, e.g. under future $eCO_2$ (Bahr et al., 2015; Ekblad et al., 1995; Nylund and Wallander, 1992), Increasing soil $P_i$ concentrations is also shown to reduce the plant carbon allocation to fungi (Bahr et al., 2015; Gower and Vitousek, 1989). This is the rationale behind this reduction function.*
- *In addition, we will include this parameter for the sensitivity analysis*

Line 316: "wais" => was

- *Agreed.*

Line 317: the piavail is another very problematic assumption, and I cannot find any theory or evidence to support it. Since the soluble P concentration is not used to calculate the plant P uptake, I could foresee that if labile P is freely taken up by the plant, the model might end up with no P limitation and the labile P might get depleted very soon. If there is no theory or literature to support this parameter, at least it should be tested in the sensitivity analysis

- *We will include this parameter in the sensitivity analysis. The conceptual meaning of $p_{iavail}$ is that only a fraction defined by this parameter that could be available for plant uptake at the time step of calculation. Please also see our response above on the plant uptake concepts and rationale.*

Line 405: where is f(Piavail) used? which equation?

- *see Equ (A8), we will add this in the revised paper*

3. Description of the region used for simulation and model setup

It seems the same study regions have been tested with the previous version of CoupModel before, and it is unclear from this section if the new Coup-CNP model is recalibrated in this study. Please state it clearly in the paper how the model is parametrized and why some parameter values differ from previous studies (I assume that is the case)

- *These regions were previous tested and used in a number of CoupModel studies (Svensson et al 2008; He et al 2018), In Svensson et al. 2008 study parameters were subjectively calibrated with the Coup-CN only model to the regional biomass data. He et al. 2018 employed a formal Bayesian calibration to the four regions with Coup-CN but with newly developed fungi model. Thus most C-N related parameters were previously calibrated, the newly introduced P parameters mostly were derived from literature if not then a subjective calibration were made to fit the observed data, in the revision we will make a Monte Carlo sensitivity analysis for all the newly introduced P parameters Few parameters values were different from previous studies, mainly due to an updated simulation design (L438-446). e.g. The rotational period in previous*

*studies were 100 years for all the region, in current setup, a different rotational period was designed. We revised this since we consider this was closer to the real forest management practice in Sweden.*

- *In addition, by adding the P cycle also some bugs were revealed, for example how interception of light by trees affect light interception by understorey, resulting into adjustments of parameter settings.*

Line 420: kg N ha-1 yr-1, right?

- *Agreed.*

Line 423: please cite the most recent FAO standard

- *We will revise to the more recent FAO 2006 guideline*

Line 443-446: difficult to follow the sentence

- *We will rephrase the description text to make it clear and easy to follow*

Line 449-450: The model was spun up for 10 years, and then a clear cut is simulated??? How do you determine the initial SOM content and soil stoichiometry? How big are the effects of initial SOM status?

- *Concerning spin up (L 432-434), the first 10 years after 1st clear-cut were used as spin-up period for the plants. From year 10 simulation results were saved until 10 years after 2st clear-cut. This was done to ensure to cover the potential nutrient leaching during the regeneration phase as in Gärdenäs et al. (2003).*
- *The initial SOM content and soil stoichiometry was reported in the Table 1 and also tested in the sensitivity analysis of varying soil N/P ratio (Fig 5 and Fig. C1). The effects of the initial SOM status were determined by the inventory data. The effects of the initial SOM status were important for the results (Fig. 5), as seen in our sensitivity test results. We reported these effects in section 5.3 and discussed thoroughly in section 6.2. However, we will also further discuss the additional effects (if any) of the new sensitivity analysis in the revised paper. Here we will add the additional soil N and P data of the Swedish Forestry Agency. Please also see the response to the initial and boundary condition comment of the topical editor, also see response concerning spin up below.*

Line 450-452: difficult to follow. Unclear to me what are the plant components and how are they treated

- *We will reword the text to make it clearer.*

Line 470: "chronicle"? I am not sure that is the right word here???

- *Agreed. We will reword in the revision.*

Line 477: This is a very unrealistic assumption; please see Yu et al. 2020 GMD

- *Thanks for pointing out this unclear formulation. We assumed for the total soil organic matter (SOM) that 5% decomposed with litter decomposition rate and 95% according to humus decomposition rate. Total amount of SOM decreased exponential*

*with soil depth. The litter were assumed to be distributed down to 0,5 m but the humus down to 1 m depth. This follows Svensson et al 2008 and He et al. 2018.*

- *We agree this is quite important as also shown in our results of soil N/P ratio sensitivity (Fig. 5), however due to the measured data only covers the O horizon, and no further data available in the soil N/P ratio in the deeper layer. However, we have conducted a survey in the literature value of N/P ratio data in Swedish forests and used the range to test the sensitivity, as the results presented in Fig 5. We will further discuss this with the new sensitivity analysis. Note we also get hold of some additional soil N and P content at the organic layer where leaf N and P content were measured and we will add that in the revision.*

Line 485-487: One specific question to Table 2 is that, why the decomposition and uptake rates for different latitudes are different, given that the temperature response function already accounts for the difference in temperature? If they are calibrated separately, what is the meaning for choosing a climatic gradient??? Table 3: I would recommend running a full sensitivity test with parameters in this table

- *We agree the referee with the full sensitivity test and a full Monte Carlo based sensitivity test will be added. We are glad the referee pointed out the different decomposition coefficient, which we had discussed these interesting different coefficients phenomena in different latitudes in a series of previous papers, e.g. Svensson et al., 2008  (Table 8, Fig 5 and Fig 6), later a formal Bayesian calibration was also conducted to constrain these decomposition parameters to the measured growth rate in He et al 2018 (e.g. Fig. 7), both results show a different coefficients were needed to obtain the measured growth data.*
*Mechanically, the different decomposition rate of humus was reflecting implicitly other drivers such as different microbial functional groups and/or the quality of the soil organic matter. Microbial responses to temperature increase in northern Sweden are known to be faster where growing season is shorter. In the previous study the impact of P and other elements on SOM quality was not explicitly taken into account. This is another reason why temperature response may differ. (Table 1), Note this issue has been discussed in great detail in Svensson et al. 2008, we will add those references in the revised manuscript.*

Line 519-521: difficult to follow the sentence

- *We will revise these sentences to make it easier to follow*

4. Results

Line 561: confusing, please rephrase

- *We will revise and make it more clear*

Line 564: why the new Coup-CNP C sequestration rates are so different from previous studies of the same regions?

- *We thank referee's comment on soil sequestration rate but we respectively disagree with the interpretation of these C sequestration rates. First, Previous model results*

*did not consider P, and we had discussed our newly introduced P cycle has clear impacts on C sequestration rates (Fig 5b). Second, we also had different set up with previous settings (see response above), thus a different C sequestration rate could be expected, given the soil sequestration rate were a net result of a number of C fluxes. However, the general trends were the same where an increasing soil C sequestration rate moving towards the south and we consider these results were rather in according with previous results not differs.*

Line 573-575: I only see that the P leaching is very small, which may infer that it has a small effect. But the fact that P leaching accounts for 30% of P deposition does not lead to the conclusion that "a small effect on the system". I guess the key point here is that both P deposition and P loss are very small compared to other fluxes, e.g. plant P uptake

- *We agree and the intention was to compare to the outflow flux to the internal flow, we will revise this in the revised paper.*

Section 5.2: the rotation period,10 years to10 years after the final felling, makes it a bit difficult to understand the results in figure 4, particularly the plant growth and change in plant in panel A. For me it is very difficult to judge how much of the changes in plant and soil pools are due to the very short spin-up time (10 years)? Is it possible to run a real spin-up to ensure a more stable state of the soil pool? Also, I did not fully understand why the pool size of 10-year-old trees differ so much in N and P size, to me it seems to be the effect of model initialization and spin-up.

- *The rotational period set up were intended to capture the high leaching period after the final felling, for the rationale and detail results see Gärdenäs et al 2003. The initial conditions for the plants were generally known and were set according the data for the plant seedlings thus were unlikely to influence our results. The soil pools (e.g. litter, humus) were previous calibrated and used for these four regions before, e.g. the soil pools were evaluated to reach a steady state also a steady soil C/N ratio over 100-year period in Svensson et al 2008. Therefore, we believe our results were not likely be much influenced by the spin-up of the plant and soil, however, we will report the initial and boundary conditions more in detail in our revision.*
*Besides, the use of a spin up to find a stable equilibrium for the soil initial values are avoided by purpose. The main reason is that we have no evidence for a long-term equilibrium of the organic pools in Swedish Forest Soils. The Swedish forest is strongly dependent of the recent history and we have substantial ongoing transitions in the organic pools. We would also like to note that our soil state variables and our simulated soil C/N ratio and N/P ratio was shown to the in a steady state (Fig 3), previous studies by Svensson et al. 2008 show a that the soil C/N ratio may be a possible indicator of the state of forest soils also when they are in a long term transition process*
- *The difference of plant pool of N and P could be a number of reasons, 1) a higher nutrient availability after clear felling 2) the clear felling kept 5% of trees thus a higher biomass and thus N and P were expected.*

5. Discussion

Section 6.1: all the studies that are compared to in the section are modeling studies, which should be made very clear.

- *We will revise the title of the section to make clear that are modelling studies compared.*

Section 6.2: In general, the discussion is interesting and the findings are encouraging. However, I do have an understanding problem regarding the soil N/P ratio. From the description in the method part, the soil N/P ratio seems to be a parameter in the sensitivity analysis. But its value is not reported in Table 3, and it seems that it is also not a constant value from Figure 3d. A more methodological problem is, only three parameters were tested in the sensitivity analysis, and the result for one parameter was presented. How could one conclude that this one is the most important parameter for the ecosystem? As I mentioned before, if this is the first study of the Coup-CNP, a better-designed sensitivity test should be performed. I am very convinced by the authors that soil N/P is an important indicator of Swedish forests, but I am convinced by the way it was accidentally chosen in this study.

- *We thank the referee for found the discussion interesting and convincing. We will revise the section with N/P ratio confusion and made it clearer, the initial soil N and P data were reported at Table 1. The varying soil N/P ratio were varied from 10 to 25 (L476-480)*
  *The results for the other parameters of the sensitivity analysis will be added in the revision, please also see response above.*

Line 676: where does this conclusion come from? increasingly P limited with time or latitude, or another gradient?

- *With decreasing latitude, we will revise this to make it more clear*

Line 682: have you checked if the threshold is the same for pine and spruce? if not, please be specific about tree species

- *We will check the data again from previous publications of Swedish forests and be specific about the tree species in the revision.*

**Reply to Anonymous Referee #2**

We thank Anonymous Referee #2 for your positive comments and constructive suggestions to our manuscript. Beloware our responses to the comments; The referee comments in normal font and our response in italics.

The paper of He et al., brings us a model that couples P into an existing CN model. It is an interesting study with special focus on mycorrhizal fungi, which is important in P dynamics, but has yet to be adequately represented in current literature. My major concern though, is that the model is heavily parameterised with great details and many parameters, but the model performance is systematically biased. Figure 2 and Table 4 evaluated modelled tree biomass, leaf C:N, leaf C:P, leaf N:P and P leaching against measurements. First of all, for a model that covers many aspects of C, N, P dynamics, variables evaluated here are not adequate to show the model performance. Secondly, the model systematically overestimate leaf C:P (all sites) and leaf C:N (3 out of 4 sites), and underestimate P leaching (all sites). I am not convinced that the model does a good job in capturing the system. Additional work and data are needed to understand the model dynamics and thoroughly assess the model performance.

- *We are glad the referee appreciates the value of this new model development. The referee raised about concerns about the model performance. Our aim was to demonstrate model behavior and the implication of the newly added P in the model structure. We tested the model, to identify the implications of integrating the P cycle using published parameter settings for C, N and mycorrhiza. The intention was not to make a site specific detailed model calibration.*
  *Moreover, possibilities toevaluateg the performance in more details are limited due to lack of P data currently available representing the four regions in a consistent way. The forest biomass was a direct result from the regional survey, thus represent the regional characteristic however; the leaf C/P ratio data are data from some few representative sites within the region and the measured P concentrations in the streams also include other source of P from the whole watershed. When evaluating the model performance, these should be bear in mind. So, the results are mostly to demonstrate current understanding of P cycle and interactions with N and C.*
- *However, to elucidate in more detail we suggest to add a Monte Carlo based sensitivity analysis of all the newly introduced P parameters to systematically evaluate the parameter sensitivity. This will demonstrate also co-variance between parameters. Yet, we estimate possibilities to reduce the current parameter uncertainty by the few data available in the current paper are limited but have some hopes as during summer, we got hold of some additional data of P and N content in the organic layer at the sites where N and P content of leaves where measured.*
  *We hope the sensitivity analysis further elucidate the model behavior and demonstrate the model ability of capturing the system response to the four regions.*
  *However, already the current test demonstrate that the model can capture the main features of the system behavior. First, the model P budgets have been detailed discussed in section 6.1 and compared to different previous modeling results and empirical data available, since we aims to present a first modelled P budget*

*complement to C and N budget, the modelled P budgets including the net weathering fluxes, the plant P uptake, harvest remove of P, soil C sequestration, soil C/N ratio, soil C/P ratio, etc were all thoroughly discussed (section 6.1). The conclusion from those comparison suggest the Coup-CNP model captured a reasonably P budget. Concerning the underestimation of P leaching, please bear in mind that the model simulated P leaching from the mineral soil. The measured stream P concentration contains also sources of P from the whole watershed but our model contains only the upstream.*

In addition, I feel it is quite difficult to follow the model description. Sometimes there are logical issues related to terminology and the separation among system compartments (please see detailed comments below). Sometimes it is due to lack of critical information in P cycling in the main text, for example, P dynamics in vegetation (allocation, resorption etc.), through mineralization etc. It might be better to put part of the information in the appendix into the main text, or at least have some overall description of these processes in the main text and point to the appendix for detailed information. The goal is to give the reader a complete picture of P cycling the model tracks.

- *We will thoroughly revise the model descriptions (sections 2.2 and 3) and add key information of the vegetation representation and mineralization in the main text. We will reorganize the model description so that the linkage of P processes to C and N processes as well as link between equations to concepts, in main text and appendix are easy for a reader to follow*

The novel part of this model, from my perspective, is related to symbiotic mycorrhizal fungi. I did not find any observations to initialise, evaluate model performance or constrain model parameters related to this part. It is also not clear what is the advantage of incorporating detailed symbiotic mycorrhizal fungi, how it affects system dynamics, what are the novel model behaviours due to this part? I feel these questions are worth answering to persuade the reader that the model is advantageous and worth the great details..

- *We thank the referee for the commenting on the importance of the fungi module. This paper follows a series of publications of the importance of considering the various uptake pathways for nutrients, Näsholm et al. (1998, 2009) experimentally show boreal forests take up organic nitrogen. For modelling, previously Beier and Eckersten (1998) had shown the need of organic N uptake to sustain the forest and soil N status in Swedish forests. These was also later shown in the calibration study by He et al. (2018) that the include of mycorrhizal fungi will make an important impact on N dynamics including the plant N uptake and N leaching (Fig. 3, He et al. (2018)), distribution of soil organic and mineral N (Fig. 4, He et al. (2018)), plant GPP, NEE, soil C soil C sequestration, etc (Fig. 5, He et al. (2018)). All these previous researches show the importance of the inclusion of mycorrhizal, thus motivates our current work, our current model description paper is to present the new P model and how the new model compared to the previous models were discussed, shown by our model, at least we can conclude that considering the P cycle and mycorrhiza fungi explicitly has aclear impact on the forest growth, and soil C sequestration (e.g. Fig 5). Our previous model estimates of the soil C sequestration in Swedish forests thus might be biased due to ignore the impact of P. When compare to*

*our more detailed fungi model the litterfall is higher, and also the plant P uptake was higher than the previous model estimates (Table 4).*

- *The previous model calibration study (He et al 2018) also compare the non-organic uptake approach to, the explicit and implicit approach of representing fungi. The data clearly reject the non-organic uptake- approach and show the importance of organic N uptake (Fig 2, He et al 2018). Those rejections are based on the fact that conventional assumptions of mineral N uptake to plant roots cannot satisfy the demands from the trees. Our new model provides one complimentary description of regional uptake patterns of nutrient consistent with available biomass data. However, a detailed evaluation and compare to the other non-explicit approaches is out of the scope of current work, with the overall aim to present the new P model. Instead, we strongly recommend including such efforts in new research.*

Detailed comments:

BeforeLine65-70, CMIP6 model results are openly available now. One model (probably the only one) that has land P component is from CSIRO, Australia. The name of the earth system model is ACCESS and land component is CABLECNP.

- *Thanks for the information, we will update CMIP6 in the revised paper*

Lines70-75, whether CNP models from Goll et al., 2012; Wang et al., 2010; Yang et al., 2014 are simplified are context dependent. As far as I know, these models incorporated key processes in C, N, P, water and energy dynamics and take into account coupling and interactions across spatial-temporal scales. They are not necessarily simpler than the model presented here.

- *Agreed. "simple" here refer to the P uptake pathways, i.e. none of the global models explicitly consider the fungi. We will revise the texts to make it clear.*

Line 75-80. Models in Medlyn et al 2016 are not earth system models per se. They are process-based vegetation models. ESMs have coupled land, atmosphere, ocean etc. Some models might be used as the land component of some ESMs. Some models may not be directly coupled.

- *Agreed, we will clarify this in the revision.*

Line 80-85. Low eco2 response do not imply "In other words, the vegetation is rather inflexible to increase P uptake". There are many factors come into play. Without CNP, the models have difficulties in capturing nutrient limitation on CO2 response. In nutrient limited locations, nutrient limitation is likely to reduce eco2 responses. And it is not only about the uptake capability. It is also related to nutrient availability.

- *We agree there are several explanations for the low eCO2 response, but here we would like to highlight the potential role of P in regulating the NPP in the models. In other words, the vegetation is rather inflexible to increase P uptake is refer to "The P cycle is assumed to be relatively closed". We believe the missing the linkage between plants-mycorrhiza fungi reduce the uptake flexibility. We will reformulate this paragraph to make this clear.*

Line 140-150, "The main model structure is a one-dimensional, vertical layered soil profile including plants." This sentence is confusing. How vertical soil profile could include plants ?

- *We will rephrase this sentence in the revised paper.*

Line 150-155, the concept of "big leaf" model assumes canopy carbon fluxes have the same relative responses to the environment as any single unshaded leaf in the upper canopy. You have two layers, trees and understory. Normally when people talk about "big leaf" model, it does not simulate light competition between up- vs. understory plants.

- *Multi-big leaves model concept was used. We will reword this.*

Line 170-171, the naming convention is quite confusing. By common definition, inorganic P is part of soil mineral P.

- *Agreed. We will revise the naming in our revision thoroughly. We will revise the definition as "Soil mineral phosphorus can be divided into labile inorganic P (Pi, phosphate ions, e.g., H3PO4, H2PO4-, HPO42-, PO43-) and **soil solid mineral**, (Pm)". We will clarify that we define soil mineral P as solid P containing primary and secondary mineral P (and occluded P) see L173-174.*

Line 180. The description of different P pools is rather confusing. If "soil mineral P is the total soil P without organic Po and labile P", how could you estimate it with total P content and bulk density. When we measure bulk density, we do not exclude the contribution from the organic matter.

- *We thank the referee to comments, the soil solid mineral P is a conceptual pool, the total P content and dry bulk density gives the total P in the soil layer, then subtract organic P and labile P (including soluble Pisol, that is phosphates)gives the soil solid mineral P, a lumped pool of solid primary and secondary mineral P like. We will revise the naming in our revision to clarify that we mean solid P with mineral P.*

Line 180-185. What do you mean by "fresh plant residues"? If plant residue that stays above soil, but it is not fresh (e.g., it is from the last year), do you exclude it from litter?

- *No, conceptually the pool can include litter from this year and before. We will revise the definition of litter and make it clearer, thanks for point this out.*

Line 180-185, "In CoupModel, soil litter could be further divided into two litter pools: one which contains readily decomposing materials (e.g., plant leaves and fine roots) and another for decomposition-resistant litter (e.g., stems and coarse roots)". If you do not represent these in your model, please skip these texts to reduce confusion.

- *Agreed. We will remove this to avoid confusion.*

Line 190-195. Do you take into account the hysteresis in P adsorption/desorption?

- *No, we assume instant equilibrium between the labile and soluble P. The adsorption/desorption of soluble P within one-time step (daily) was assumed excluding eventual hysteresis of the soluble P and labile P in the soil water. Similar set up as other models, e.g. Wang et al 2010.*

Line 170-205, you talked about litter pool, how do you treat soil organic matter/P pool? Do you only have humus pool? If so, non-symbiotic soil microbes are classified as litter in your model?

- *The soil organic matter is conceptually divided into litter and humus pools, non-symbiotic soil microbes are implicitly classified as litter in current model, this was made to follow Svensson et al. (2008) and He et al. (2018). We will add this information of the soil organic matter pools in the revised paper.*

Lines 210-215, "During certain seasons, plants can also capture mobile P (as well as mobile N) to prepare for rapid growth in the spring". What do you mean here? You mean plants take up more P in other seasons other than Spring, store it and use it in Spring? How does it occur? What do you mean by mobile P(N)?

- *We realize this description is unclear for both referees asked about the mobile pools and thus we will carefully reword this in the revise paper to make it clear.*
*The plant mobile N and P are state variables designed to mimic the nutrient reallocation or retranslocation process, a process by which nutrients (here N and P) were mobilized from senescing structure to developing tissues e.g. before litterfall, thus acting as an important mechanism to reduce dependence on nutrient uptake and increase nutrient recycle. Aerts, (1996) showed mean nutrient resorption efficiency for perennial plant was c.a. 50% for N and slightly higher, 52% for P. A more recent global synthesis of most measured data for woody plants showed similar results that mean N resorption efficiency of 48.4% and mean P resorption efficiency was 53.3% (Yan et al. 2018). Nieminen and Helmisaari (1996) show a 67-88% mobile nutrient N and P decreasing during needle senescence for Scots pine in Finland. These evidence serves the rational of including the nutrient resorption process in the model. In Coup-CNP, P was a similar concept to that of N, this process simulates the resorption of nutrients before litterfall (N and P were assumed to be stored internally in the mobile state variable and these can then be used next year to develop new shoots/leaves). We will describe this more in detail and add this into sensitivity test to evaluate how sensitive the model outs to this.*

Lines 220-225, I don't understand what do you mean by "In Coup-CNP, biochemical mineralization is defined as organic uptake". Biochemical mineralization and organic uptake are different processes.

- *We agree with the referee that theoretically these processes are different and will rephrase it in the revision.*

Line 316, "wais" to "was"

- *Agreed.*

Line 535 – 540 and Figure 2. From Figure 2, the model systematically over-estimate Leaf C/P and leaf C/N ratio (except one site). Is it because an over-estimation of the leaf biomass? If there are coherent bias for all or most sites, it is not a neglectable issue. Figure 4. Why do you plot plant growth in C flux but change in plant for P flux, please be coherent and consistent. Table 6, systematically underestimation of P leaching

- *We thank for the comments regarding the model performance. There is generally a lack of P data currently, the model configuration was designed and benchmarked to a regional representation, where the forest biomass was a direct result from the regional survey, thus represent the regional characteristic however; the leaf C/P ratio data are data from some few representative sites within the region (two sites for each region, max three sites for the southmost region) and the measured P concentrations in the streams also include  other P source from the whole watershed thus discrepancy could also be attribute to these. When evaluating the model performance, these should be bear in mind. We will revise the descriptions to make this clear.*
- *However, to further investigate the model performance, we had employed the Monte Carlo based sensitivity analysis   to further analyze the model with possibly investigate how the model behavior vary with varying parameters. The plant growth in Fig. 4 represents the net ecosystem production, which is an additional flux from the atmosphere compared to N and P. Change  in plant is the difference between the pool at start of simulation (when forest was 10 years old) and at end of simulation in the next generation forest at age 10 years.  We will describe this more clearly in the revision, please also see response above for model performance.*

**Reply to editor comments**

We thank editor for your positive comments and constructive suggestions to our manuscript. Here are our responses to the comments; The editor comments are in normal font and our response in italics.

Thanks for preparing a revised version of the manuscript addressing my previous comments. I will accept now the manuscript for publication in the discussion forum and formally start the peer review process. However, your answer to my question on the type of dynamic update, with your respective answer about coupled partial differential equations, suggests that your presentation of equations in the text is not adequate, and that you would have to rewrite many of the equations to make explicit the use of partial differential equations. You also would have to state more explicitly the boundary conditions and the initial conditions since these are factors that strongly influence the solution of the system of equations.
I accept the current version for the review process, but keep this comment in mind when preparing a revised version addressing reviewers' comments.

- *Thanks for the comments. We will rewrite our equations to differential forms accordingly. We will also make the initial conditions and boundary conditions more explicit in the text. The water and heat boundary and initial conditions were kept the same with the previous publications, i.e. Svensson et al 2008 and He et al 2018. We will explicitly descript the initial conditions of P in addition to current Section 4.2 and also Table 1.*

**References**

Aerts, R., 1996, Nutrient resorption from senescing leaves of perenials: are there general patterns? Journal of Ecology, 84 (4): 597-608.

Bahr, A., Ellström, M., Bergh, J. and Wallander, H., 2015. Nitrogen leaching and ectomycorrhizal nitrogen retention capacity in a Norway spruce forest fertilized with nitrogen and phosphorus. Plant and Soil, 390(1-2): 323-335.

Beier and Eckersten, (1998), Modelling the effects of nitrogen addition on soil nitrogen status and nitrogen uptake in a Norway spruce stand in Denmark, Environmental Pollution, 102 (1): 409-414, doi.org/10.1016/S0269-7491(98)80061-4.

Ekblad, A., Wallander, H., Carlsson, R. and Huss-Danell, K., 1995. Fungal biomass in roots and extramatrical mycelium in relation to macronutrients and plant biomass of ectomycorrhizal Pinus Sylvestris and Alnus incana. New Phytologist, 131: 443-451.

Gower, S.T. and Vitousek, P.M., 1989. Effects of nutrient amendments on fine root biomass in a primary successional forest in Hawai'i. Oecologia, 81: 566-568.

Gärdenäs, A., Eckersten, H., and Lillemägi, M.: Modeling long-term effects of N fertilization and N deposition on the N balances of forest stands in Sweden, Swedish University of Agricultural Sciences1651-7210, 34, 2003

He, H., Meyer, A., Jansson, P.-E., Svensson, M., Rütting, T., and Klemedtsson, L.: Simulating ectomycorrhiza in boreal forests: implementing ectomycorrhizal fungi model MYCOFON in CoupModel (v5), Geosci. Model Dev., 11, 725–751, https://doi.org/10.5194/gmd-11-725-2018, 2018.

Jayachandran K, Schwab AP, Hetrick BAD (1992) Mineralization of organic phosphorus by vesicular-arbuscular mycorrhizal fungi. Soil Biol Biochem 24:897–903.

Lindahl, B., Taylor, A. F.S., and Finlay, R.D., (2002) Defining nutritional constraints on carbon cycling in boreal forests-towards a less "phytocentric" perspective, Plant and soil, 242: 123-135

Näsholm, T., Ekblad, A., Nordin, A. et al. Boreal forest plants take up organic nitrogen. Nature **392,** 914–916 (1998). https://doi.org/10.1038/31921.

Näsholm, T., Kielland, K., Ganeteg, U., 2009, Uptake of organic nitrogen by plants, New Phytologist, 182(1):31-48, doi.org/10.1111/j.1469-8137.2008.02751.x.

Nieminen, T., and Helmisaari, H.-S., (1996), Nutrient retranslocation in the foliage of pinus sylvestris L. growing along a heavy metal pollution gradient, Tree Physiology, 16 (10): 825-831, 10.1093/treephys/16.10.825.

Nylund, J.-E. and Wallander, H., 1992. Ergosterol Analysis as a means of quantifying mycorrhizal biomass. Methods in Microbiology, 24: 77-88.

Svensson, M., Jansson, P.-E., Kleja, D. B., (2008) Modelling soil C sequestration in spruce forest ecosystems along a Swedish transect based on current conditions, Biogeochemistry, 89: 95-119, 10.1007/sl0533-007-913.

Wang, Y. P., Law, R. M., and Pak, B. (2010), A global model of carbon, nitrogen and phosphorus cycles for the terrestrial biosphere, Biogeosciences, 7, 2261-2282, 10.5194/bg-7-2261-2010.

Yang, T., Zhu, J. and Yang, K., (2018), Leaf nitrogen and phosphorus resorption of woody species in response to climatic conditions and soil nutrients: a meta-analysis, Journal of Forestry Researcher, 29: 905-913, doi.org/10.1007/s11676-017-0519-z .

---

## Author Response (AR1)

**Point by point response to all reference comments**

Dear Editor and reviewers,

We are thankful for the valuable comments you and both reviewers made for our manuscript.
5   In the following, you can find our point-by-point reply to all the comments. Since most of the comments are already answered in our published response during the open discussions. This reply then aims to give the details of what changes we have made during the revision. Besides, we would like to highlight the following three major changes we made during the revision.

First, we further conducted global sensitivity analysis for all the newly P parameter and some
10   N relevant parameters for the Coup-CNP model. The details of the approach and results are given as a Supplement. The global sensitivity analysis results show the importance of P and N availability in determining the C, N and P cycling in forests ecosystems in Sweden. This confirms our major conclusions in the main paper but also justifies our key parameter sensitivity analysis reported in the paper. The simulation files for global sensitivity analysis are
15   archived at Zenodo (https://doi.org/10.5281/zenodo.4291963).

Second, we thoroughly rewrite most of the introduction and presentation of the model as suggested by both referees.

Third, before submission, the revised paper and newly added supplement have been language checked by professional English language proofing company. Hence, some formulation below
20   might slightly change.

We hope our revised paper now to be satisfactory for publishing.

Sincerely,

25   Hongxing He (on behalf of the author team)

**Reply to Anonymous Referee #1**

We thank Anonymous Referee #1 for your positive comments and constructive suggestions to our manuscript. Here are our changes made to address the comments. The referee comments are given in normal font and our responses in italics.

The paper by He et al. presented the integration of the phosphorus (P) cycle into the CoupModel and evaluation of the new model, Coup-CNP, against four regions in Sweden that differ in climate and fertility. It is overall a very interesting paper, particularly with the novel setup of both NP cycles and mycorrhiza. The results are well presented, and the discussion is clear and well organized; the authors have put much effort into compiling the information of the model development both in the paper and appendix. Overall I think the paper is in good shape and contributes to advances in the modeling community.

- *We are glad the referee appreciates the science and novelty value of this new model development. We appraise that our P cycle and mycorrhiza model has a good potential to contribute to improved understanding of and insights into current ongoing discussion of nutrients impacts on C cycle. We also pleased that the referee finds the presentation of the results and discussions are in a good shape.*

However, the current quality of the paper needs to be improved before final acceptance. First of all, I found the quality of English writing an obstacle for me to keep focusing on the scientific content of the paper. I would recommend the authors to go for a professional editing service with the paper. I listed some obvious mistakes in the detailed comments, mostly before the results section, since I stopped to do that for the rest of the paper simply due to the heavy load of scientific information.

- *We have thoroughly rechecked the texts of the paper, also rewritten the whole model description (now section 2) and description of the region (now section 3). The language of the revised manuscript and the newly added supplement had been edited by professional editing service in UK. We believe the language has been much improved now.*

Secondly, this study lacks a proper sensitivity analysis. The authors did a simple sensitivity test on the fungal organic uptake rates of N and P and presented the result in the appendix. As far as I see from the description in this paper, Coup-CNP is a heavily parameterized model with a huge number of parameters. It is extremely important to run a proper sensitivity analysis with multiple parameters, not only to see the effects of parameterization on model outputs but also to test the stability and robustness of the model.

- *We have now conducted global sensitivity tests including all the P parameters and few N related parameter (n=34). The tested parameters, model approach and sensitivity analysis results, including parameter sensitivity for different selected model variables, are given in the newly added supplement (details see section 1-2 in supplement).*
- *Briefly, the global sensitivity results (i.e., Table S.2; Table S.3; Table S.4) show that parameter importance/sensitivity differs in terms of selected out variables, also in terms*

*of studied region. Overall, the most crucial parameters (ranked by the correlation coefficient, r between parameter and model output) in controlling the C, N and P outputs is the initial soil organic P in humus, then followed by the short-cut N/P uptake rates (renamed to replace 'organic uptake' in previous version) and non-symbiosis microbe C/P ratio both of which directly regulating the availability of N and P; then followed by the plant growth and photosynthesis related parameters (e.g. leaf optimal/threshold C/P ratios for photosynthesis); then followed by the fungi related parameters, and the least importance is the weathering and partitioning of inorganic P parameters. All parameters show importance, in determining the output (note only, r ≥0.2 or ≤ -0.2 reported in Table S.2, S.3, S.4). Our global sensitivity analysis thus confirms the stability and robustness of the model (see section 3 in supplement).*

- *More importantly, the global sensitivity analysis results also show the importance of P and N availability in determining the C, N and P cycling in forests ecosystems in Sweden. This confirms our major conclusions in the main paper but also justifies our key parameter sensitivity analysis (initial soil organic P, short-cut N/P uptake coefficients) reported in the paper.*

- *We have now added a separated section in the main paper, "Section 3.5 sensitivity analysis" to briefly describe the global sensitivity analysis and links to the key parameter sensitivity analysis reported in the paper as following: "We conducted a global sensitivity analysis of the new Coup-CNP model to its parameterization (n=34) using a Monte-Carlo based sensitivity analysis method to assess the stability and robustness of the model with respect to its parameter values. The sampled parameters and their ranges (Table S.1), model design and global sensitivity results (Table S.2, S.3, S.4) were reported in detail in the supplement. Based on these simulations and parameter sensitivity rankings, we select three most important parameters (n=3), which has a strong effect on the model outcome to further form a new set of model runs, which are used for the model sensitivity analysis presented in this paper. The selected three parameters are initial soil humus P, short-cut N uptake rate and short-cut P uptake rate (Table S.2, S.3), all strongly regulates the soil N and P availability."*

Thirdly, apart from a sensitivity test, I would also recommend the authors to conduct a few model experiments to see the model responses to alternative model assumptions or changing climatic/environmental conditions. For example, the authors introduced the plant growth response to P stress based on leaf C/P ratio (Eq.9), which is novel and interesting but at the same time debatable. I personally would really like to see the effect of this mechanism on the predicted GPP/NPP and biomass. Another example is the role of mycorrhiza uptake and the so-called organic uptake of N and P. I found that the authors made some very strong assumptions regarding the uptake competition (the sequence of uptake) between plant, fungal and adsorption, and it would be interesting (and fundamental) to see the effect of these strong assumptions.

- *We have conducted global sensitivity analysis for two regions with different environmental conditions: one for the northernmost 64ºN region with N limitation (Table S.2; Fig.S.1; Fig.S.2) and the southernmost 56ºN region with P limitation (Table S.3; Fig.S.3; Fig.S.4). We further conducted an additional global sensitivity analysis in 64ºN region by removing the identified dominant impacts of the three parameters (i.e. initial soil organic P in humus pool, and short-cut N/P uptake*

*coefficients) to better show the sensitivity of the other parameters (Table S.4). Thus, overall three extra model experiments have been conducted (details see section 1-3 in supplement).*

- *The parameters show different sensitivity to the different region. For the northernmost 64ºN region, C and N change in plant and soil is mostly sensitive to the short-cut N uptake rate ($o_{uptNhumus}$) (Table S.2). This is expected as the region is identified as being N limited. For the southernmost 56ºN region, C and N change in plant and soil is more sensitive to the initial soil organic P in humus (Table S.3). This is also expected, as the region is P limited. Other global sensitivity analysis results see supplement section 3 and response above.*

- *Our global sensitivity analysis show high sensitivity of the P response to GPP to the leaf C/P optimal parameter, $p_{cp,opt}$. Similarly, N response to GPP is sensitive to leaf C/N parameters (Table S.2, S.3, S.4). However, leaf C/P optimal parameter does not shown high sensitivity to the simulated plant C change, likely due to dominant of other parameters that regulates the N and P availability (short-cut uptake rates, e.g. $o_{uptNhumus}$, $o_{uptPhumus}$), plant allocation, $p_{fopt}$ etc. (Table S.2, S.3). The additional sensitivity analysis for the N limited 64ºN region in Table S.4 show the importance of leaf C/N threshold parameter, $n_{cn,th}$ in determining plant C change and total C harvest.*

- *Our global sensitivity analysis show parameters relevant to plant P demand, short-cut uptake rate, fungi related parameters, and partitioning in the soil are all sensitive in regulating the plant and soil P cycle (see supplement section 3). Within our tested parameter ranges, the parameter sensitivity rankings show, the most crucial parameter is the soil organic P, followed by the short-cut uptake rates for N and P, then the fungi parameters then lastly is the partitioning (i.e. adsorption).*

- *Please note the changing climatic/environmental model experiments were already conducted by applying the new Coup-CNP model on four regions with varying climate (annual temperature between 0.7-7.1 C) and fertility (with the soil C/N and C/P ratios between 19.8-31.5 and 425-633, respectively) along a gradient from South to North Sweden.*

Detailed comments

1. Abstract and Introduction

Line18: make "which explicitly consider mycorrhizal interactions" a relative clause after "The extended Coup-CNP"

- *Changed accordingly.*

Line 26: what is "a steady state in P availability"? I don't find "P availability" from the P budget

- *We have changed to "The simulated P budgets revealed that southern forests are losing P, while northern forests have their P budget in balance"*

Line 40: "nutrient cycling" is not a biochemical reaction

- *We have changed to "Phosphorus (P) is an essential element for all photosynthesizing plants in terrestrial ecosystems, with the P cycle coupled to Carbon (C) and Nitrogen*

*(N) fluxes especially through processes such as decomposition of soil organic matter and nutrient uptake"*

Line 50: it is true that N inputs to the atmosphere increased due to human activity, but for terrestrial ecosystems, the important process is the N deposited from the atmosphere

- *Revised to "atmospheric N deposition"*

Line 52: mechanisms can not be amplified, right? Second paragraph: I think it is a brilliant idea to review the literature of the P cycle in current models, but the organization of information needs to be much improved in this paragraph. I also have some disagreements with the authors about the interpretations of some cited publications, and would like to discuss with the authors about them.

- *We have removed "mechanisms". We shorten the literature P model review and thoroughly recheck the cited publications to ensure they are cited correctly. The reference to the reviewed P models is revised by referring to the more recent model description paper (e.g. Groenendijk et al., 2005). Papers which are published during open discussions are also added (e.g. Du et al. 2020).*

Line 56-65: I think this part is irrelevant to the overall discussion and conclusion of this study. I would recommend to remove or to shorten it.

- *We shortened this model review part in to "Nevertheless, the P cycle is seldom incorporated into ecosystem model structures. Incorporating the P cycle is essential in improving global models as a tool for assessment of climate-C cycling interactions (Reed et al., 2015). Most of the process-based models that can simulate P cycling were specifically developed for agricultural systems and focus on the soil processes, e.g., EPIC (Jones et al., 1984, Gassman et al., 2005), ANIMO (Groenendijk et al., 2005), and GLEAMS (Knisel and Turtola, 2000). A few catchment-scale models focus on surface water quality, e.g. SWAT (Arnold et al., 2012), HYPE (Arheimer et al., 2012), and INCA-P (Jackson-Blake et al., 2016)."*

Line 72: there are some more P-enabled ESMs, e.g. Zhu et al. 2016 Biogeosciences, Goll et al. 2017 GMD, Thum et al. 2019 GMD.

- *Added accordingly*

Line 75: Zaehle et al. 2014 does not support your statement here

- *Deleted.*

Line 76-92: The interpretation of these studies is a bit imprecise and vague. I found it difficult to jump from one study to another one; maybe it is better to reorganize all the studies with some intrinsic links, such as common problems or findings. What I will recommend is to focus on the role and effect of plant P uptake in different model studies. Yu et al. only included the P cycle into the ForSAFE model. I would not phrase it as "developed the model", which causes confusion

- *We reorganized and revised the texts for the reviewed modelling studies*

- *We have changed "developed the model" to "included the P cycle into" for ForSAFE statement.*
- *We have now added the "implicitly or explicitly of symbiotic mycorrhiza and other soil microbes" after "these P enabled models differ in how they described soil P dynamics" to be more specific.*

Line 99: whose interaction with soil mycorrhizal fungi?

- *Revised to "interaction between plants and mycorrhizal fungi"*

Line 100: I don't fully agree with the interpretation of the references here. These data driven meta-analyses do not really explain "how P availability affects plant growth", and if this mechanism is influenced by mycorrhizae-plant interactions. They are more of "a proof" than "an explanation" to me

- *We have changed to "Global meta-analysis studies highlighted that symbiosis between plants and soil mycorrhizal fungi strongly influences plant P availability that further affects plant growth" to avoid confusion.*

Line 109-111: please restructure the sentence

- *We have changed sentence into "To the best of our knowledge, only Orwin et al. (2011) have presented an ecosystem model that consider C, N and P together with symbiotic fungi. They found that considering organic nutrient uptake by symbiotic fungi in an ecosystem model can significantly increase soil C storage, with this effect more pronounced under nutrient-limited conditions. The organic nutrient uptake in their model was to mimic the additional pathway that plant can utilize organic nutrients by biochemical mineralization either in symbiosis with mycorrhizal fungi or root exudates"*

Line 132: soil organic matter is a more commonly used term than "soil organics"

- *Changed accordingly.*

Line 134: there is little evidence for organic P uptake of plants and microbes, as far as I know

- *We have added the following in the introduction section to explain the organic nutrient uptake concept better in Coup-CNP: "The organic nutrient uptake in their model was to mimic the additional pathway that plant can utilize organic nutrients by biochemical mineralization either in symbiosis with mycorrhizal fungi or root exudates (e.g. Schachtman et al., 1998, Gärdenäs et al. 2011, Richardson et al. 2005). However, in Orwin's model, plant growth was static; thus plant-soil or plant-environment interactions were largely ignored. Our model (Eckersten and Beier, 1998; He et al., 2018) also includes a nutrient-short-cut uptake as a process in the rhizosphere. The assumption is that nutrients released by biochemical mineralization are instantly taken up by the symbiotic microbes and/ or the plants, thereby by-passing the soil matrix solution."*
- *The section 2.2.3 further describe the concepts and calculations as "Biochemical mineralization, on the other hand, describes the release of $P_i$ through extracellular enzymatic releases (e.g., phosphatases by root exudates), which are driven by plant*

*demand for nutrients (Richardson and Simpson, 2011). In Coup-CNP, biochemical
mineralization is conceptually included in the nutrient short-cut uptake (called
organic uptake in earlier CoupModel publications) and assumed to be driven by the
unfulfilled plant P demand after $P_{ilab}$ root uptake (equ A.8) but regulated by the
availability (i.e. short-cut uptake coefficients in equ A.4)."*

- *Further, to avoid confusion with the uptake of organic molecules, we rename the
organic P/N uptake into "short-cut P/N uptake" in the entire paper.*

2.  Model structure and description of processes linked to the phosphorus pool

Please rename the title, maybe "Model structure and phosphorus process description"?
Another piece of advice is linking the process description of Section 2.2 with the equation
number in Section 3 and Appendix A. It is much easier for the readers to track
information in this way.

- *We retitled this section to "Description of model structure and phosphorus model".*
- *We have merged previous section 2 and 3 and substantially rewritten the entire
section. We have changed the model description organization by starting with
describing the new P model concepts and defining its pools and assumptions in
section 2.2. Then starting the detailed process and equation description in sub-
sections 2.2.1-2.2.6. "Soil inorganic phosphorus dynamics and nutrient short-cut
uptake" has now been moved to section 2.2.3. We have briefly described the
processes that detailed described in appendix, and added linkages (i.e. equation
numbers, sections in appendix) to the appendix when possible to make it easier for
readers to follow.*

Line 142: what does "flexible" model mean?

- *Removed.*

Line 145: please check the grammar

- *We have changed to "The main model structure is a one-dimensional, vertical layered
soil profile. The core of the model consists of five sets of coupled partial differential
equations, one for water, heat, C, N, and P cycles (the later one in v6.0),
respectively"*

Line147: maybe already mention the normal time step and the smaller time step here?

- *We have added the time step "daily" here as suggested.*

Line 149: "crucial"??? what and why?

- *Removed. We have further described what we mean "crucial" here by "In this
application, we used a daily time step for all five, but a smaller time step was applied
for the water and heat calculations during specific events with peaks in water and/or
heat flow such as during snow melting to ensure the numerical stability and
accuracy".*

Line 151: why the radiation forcing has to be "global"???

- *Reworded this to "global, i.e. sum of direct and diffuse shortwave incoming radiation".*

Line 153: compete for light??? Not "light interception"?

- *Corrected.*

Line 161: strange sentence structure, please consider adjusting it

- *Reworded into "We developed the P model in a way that 1) focus on the key P processes for biogeochemical cycling, e.g., dynamic plant growth and P leaching, and 2) follows the conceptual structure of CoupModel as closely as possible.".*

Line 164: "can differ" => differs, or do you mean that there are two options for time step???

- *Deleted this sentence since the same time step was used in this study.*

Line 166: difficult to understand the sentence

- *Revised to "For simplicity, the equations are given in a form that reflects one time step and one of the layers that represent the entire soil profile. The symbols in this paper were designed to conform the CoupModel nomenclature in the following way: uppercase P means state variables, lowercase p means parameters related in P processes"*

Line 171: there is not a common definition of "mineral P", please distinguish it from other inorganic P forms

- *We have re-described the pools as "The soil inorganic P has new and renewed state variables. New is the soil solid inorganic $P_{solid}$, a lumped pool containing primary and secondary mineral compounds containing P such as apatite (and occluded P) (Smeck, 1985; Wang et al., 2007). $P_{ilab}$ is the sum of phosphate ions absorbed and those in soil solutions, analog to the mineral pool in salt tracer representation in CoupModel (Gärdenäs et al., 2006). Instantaneous equilibrium between adsorbed and soil solution are assumed. Plant and microbes take up phosphate ions from the $P_{ilab}$. $P_{isol}$ can be compared with the sum of N state variables $NH_4^+$ and $NO_3^-$ while being an intrinsic part of $P_{ilab}$ (Fig. 1)."*
- *We have renamed the previous mineral P pool as "soil solid inorganic $P_{solid}$" in the entire paper to avoid confusion*

Line 174: "inorganic-phosphorus", why a hyphen here?

- *Removed*

Line 176-180: I don't see the connection between the model definition and Hedley fractionation. Please elaborate.

- *Removed*

Line 183: "which contains"=> "for"

- *We have deleted this sentence, since only one litter pool was used in this study.*

310      Line 185: which decomposition rate is used for the combined litter pool?

- *We have deleted this sentence, since only one litter pool was used in this study.*

Line 185-200: If I understand correctly, Coup-CNP applied a three-pool structure for soil inorganic P, which is different from most other P models. One thing that is particularly different in this study is that the role of adsorption/desorption is greatly neglected by most

315      biochemical processes since Pisol is only relevant to transport and Pilab is relevant for other processes, such as deposition, weathering, plant/fungal uptake and etc.. It is a very interesting setup, but I think it needs to be better explained. Particularly, the statement that "These Pi ions are normally loosely adsorbed to surfaces and can thus easily re-enter the Pilab pool through the desorption process (McGechan and Lewis, 2002)." is wrong.

320      There is plenty of evidence for the strong adsorption of phosphate, which is also the main reason for the extremely low soluble inorganic P concentration in the soil water. The main reason that plant and microbe can take up enough P in such a low P concentration is probably the fast replenishing of soluble P in soil water, which are the consequences of desorption/diffusion and biological mobilization (mineralization). Please see Buenemann

325      et al. 2016, SBB and Pistocchi et al. 2018, SBB, and the references therein for more information.

- *We have redescribed and defined the soil P pools more clearly in section 2.2: "The soil inorganic P has new and renewed state variables. New is the soil solid inorganic $P_{solid}$, a lumped pool containing primary and secondary mineral compounds containing P such as apatite (and occluded P) (Smeck, 1985; Wang et al., 2007). $P_{ilab}$*

330      *is the sum of phosphate ions absorbed and those in soil solutions, analog to the mineral pool in salt tracer representation in CoupModel (Gärdenäs et al., 2006). Instantaneous equilibrium between adsorbed and soil solution are assumed."*
- *We also make our assumptions more clearly. The assumption between the soluble P and the labile P is described in section 2.2.2: "the soluble ($P_{isol}$) and adsorbed P*

335      *reach equilibrium in less than 1 hour (Cole et al., 1977; Olander and Vitousek, 2005). We assume that the $P_{isol}$ and adsorbed part of $P_{ilab}$ are always in equilibrium as daily timestep is used (equ 5). The modified Langmuir isotherm (Barrow, 1979) was used to model the fast and reversible sorption process within $P_{ilab}$"*

340      - *We have deleted the "These Pi ions are normally loosely adsorbed to surfaces and can thus easily re-enter the Pilab pool through the desorption process (McGechan and Lewis, 2002)." to avoid confusion.*

Line 214: what is mobile P and N? this is a very strong assumption that plants can capture nutrients from litterfall, and I wonder how sensitive are the model outputs to this

345      assumption.

- *The following section was added in A4 plant litterfall to describe the pool and assumption more clearly "During litterfall seasons, plants can reallocate P and N from leaves to an internal, mobile storage to prepare for rapid growth in the spring, a known mechanism to increase efficient use of nutrients (e.g. Aerts, 1996; Niemien and*

350      *Helmisaari, 1996) (also see $m_{retain}$ in Table S.1 in supplementary)".*
- *Note the $m_{retain}$ is also included in the global sensitivity analysis and showed the modeled ecosystem C change is sensitive to this parameter in Table S.4*

Line 221: what are the enzymatic processes? Please be specific. Btw, phosphatase is not a process

355     -  *Reworded to "phosphatase released by the root exudates"*

Line 222-225: well, this is another astonishing assumption, which needs to be properly tested. And the hidden hypothesis that it only occurs after inorganic P uptake when plant P demand is not fully met is also quite strong from my personal feeling. It basically means that there are no interactions (feedback/competition) between soil organic and

360  inorganic P cycling processes, all the feedback mechanisms have to go through the plant growth & litterfall pathway. I wonder how the model will perform in an extremely P limited ecosystem.

    -  *We realize the previous description of P uptake may be unclear thus we have revised this to make it clearer and described our assumptions more clear. We first added the*

365     *following in introduction to motivate and describe the background of the short-cut uptake concepts: "The organic nutrient uptake in their model was to mimic the additional pathway that plant can utilize organic nutrients by biochemical mineralization either in symbiosis with mycorrhizal fungi or root exudates (e.g. Schachtman et al., 1998, Gärdenäs et al. 2011, Richardson et al. 2005). However, in*

370     *Orwin's model, plant growth was static; thus plant-soil or plant-environment interactions were largely ignored. Our model (Eckersten and Beier, 1998; He et al., 2018) also includes a nutrient-short-cut uptake as a process in the rhizosphere. The assumption is that nutrients released by biochemical mineralization are instantly taken up by the symbiotic microbes and/ or the plants, thereby by-passing the soil*

375     *matrix solution."*
    -  *We have put "soil inorganic phosphorus dynamics and nutrient short-cut uptake" as a separate section 2.2.3. This is to better describe the soil inorganic and organic processes, also to highlight the interaction of soil organic and inorganic P cycling processes.*

380 Line 229: how is the DOM redistributed between layers? Is it described in the paper?

    -  *We have added the following into the A2. Section to describe the DOM redistribution more in detail: "The redistribution is done following that of water flow, as the DOM is assumed to have full mobility with water. The formation of DOM is from litter and humus. The dissolved organic matter can be fixed by humus via adsorption,*

385     *precipitation, etc. A fixation coefficient, $d_{DOD}$, which varies between layers, was introduced (Kalbitz et al., 2000; Kaiser and Kalbitz, 2012). Parameterization from Svensson et al. (2008) were used in this study."*

    3.  Equations describing key phosphorus processes/fluxes and their parameterization

390 One major trouble to me is that the use of both uppercase and lowercase P (p) in the equations. It is extremely difficult sometimes, please consider replacing one of them with another letter. Another major issue is that I could not find information on how leaf P content is calculated, which is essential to understand some results

- *We have revised the symbols of the equations where could lead to possible confusion and double-check consistency in the entire paper. For instance, we have changed the "soil mineral $P_m$" into "soil solid inorganic $P_{solid}$". We have changed the symbols for three soil organic pools, soil litter, soil humus, dissolved organic P into $P_{Litter}$ and $P_{Hmus}$ and $P_{DOP}$ for easier to understand, and so on.*
- *To avoid confusion, we have added the following explanation texts in the main texts (section 2.2) just before the equation sections to explain the rule of the symbol to make it easier to follow: "For simplicity, the equations are given in a form that reflects one time step and one of the layers that represent the entire soil profile. The symbols in this paper were designed to conform the CoupModel nomenclature in the following way: uppercase P means state variables, lowercase p means parameters related in P processes."*
- *We have updated the leaf C/P ratio description (under section 2.2.4) with the following to make it clear the leaf C/P is a variable that calculated at each time step: "The leaf C/P ratio is calculated at each time step with the leaf state variables C and P."*

Line 243: judging from Eq.4, I don't think "proportional" is the right word here

- *We have changed this sentence into "The weathering rate depends on soil pH and temperature (Guidry and Machenzie, 2000) and is calculated as".*

Line 247: how does erosion affect weathering rate? I cannot find it in the paper

- *We have added "the erosion affect the weathering rate by reducing the pool size of $P_{solid}$, (equ. A. 14)" in 2.2.1 weathering description.*

Line 254: there is a potential problem that diffusion is also considered as weathering. how uncertain is it to assume diffusion and weathering has the same temperature response? This is even a bigger problem for pH response as there is no evidence that pH affects diffusion
- *First, the weathering in Coup-CNP is independent of the mobile part of P in our model structure. We have revised our model concept description in section 2.2 and now renamed the previous soil mineral pool into "soil solid inorganic $P_{solid}$".*
- *However, again our aim was to build a simple yet realistic P net weathering flux. We compared to the current net weathering flux to a more detailed and rigorous geochemical model PROFILE, but not a dynamical model; that is more widely used for weathering estimates and current P flux estimates were rather similar (see discussion).*

Line 295: I am not sure if this theory is applicable to leaf CP ratio since P is not as essential as N for photosynthesis and the role of leaf P in photosynthesis is not well understood yet. As I mentioned before, it will be interesting to conduct model experiments to test this theory. Additionally, I did not find the information on how CoupCNP calculates leaf P content.

- *Please see our response above*

Line 303: The mycorrhiza module??? This sentence is confusing to me

- *We have removed the sentence "P fungi processes analog to N processes (He et al., 2018) are found in appendix A" as the fungi P processes are described in main text. Hopes that answer the reviewer question as we are not completely sure we got it right.*

440 Line314: Eq.9 seems the only place that soluble P concentration is used except leaching, how realistic is it to take this assumption directly from N, given the fact that P concentration is much lower than N?

- *We have revised to "analogue to the N response function" instead of the same.*

Line 316: "wais" => was

445 - *Corrected.*

Line 317: the piavail is another very problematic assumption, and I cannot find any theory or evidence to support it. Since the soluble P concentration is not used to calculate the plant P uptake, I could foresee that if labile P is freely taken up by the plant, the model might end up with no P limitation and the labile P might get depleted very soon. If there
450 is no theory or literature to support this parameter, at least it should be tested in the sensitivity analysis

- *We have included this parameter $p_{iavail}$ in the global sensitivity analysis. This parameter show sensitivity to plant and soil N change in Table S.4 in supplement. The conceptual meaning of $p_{iavail}$ is that only a fraction defined by this parameter that*
455 *could be available for plant uptake at the time step of calculation. Please also see our response above on the plant uptake concepts and rationale.*

Line 405: where is f(Piavail) used? which equation?

- *We have added "equ (A.8)"in the texts to point out where it was used.*

460 3. Description of the region used for simulation and model setup

It seems the same study regions have been tested with the previous version of CoupModel before, and it is unclear from this section if the new Coup-CNP model is recalibrated in this study. Please state it clearly in the paper how the model is parametrized and why some parameter values differ from previous studies (I assume that is the case)

465 - *We have re-described the regions and model setup. Major changes include added two separate sections of 3.4 Model forcing, initial and boundary conditions and section 3.5 sensitivity analysis to described these into more detail.*
- *We added the following to describe how the parameter values were determined in the reference model run: "The C and N parameters for these regions in CoupModel were*
470 *previously tested and calibrated in a number of studies (Svensson et al., 2008) and those of fungi by (He et al., 2018) (Table 2). The surface cover parameters and litterfall rates of understory vegetation were modified from Svensson et al. (2008) to achieve a more realistic understory dynamics in those regions (Table 3). Most of the default values of the newly introduced P parameters were derived from literature (Table 2, 3).*
475 *For instance, the optimal leaf C/P ratios for forest growth, C/P ratios of individual*

*plant components were obtained from empirical measurements from Swedish forests (e.g. Thelin et al., (1998; 2002)). The weathering and surface runoff parameters were defined according to laboratory empirical data (e.g. Guidry and Machenzie, (2000)). "*

480 Line 420: kg N ha-1 yr-1, right?

- *Corrected.*

Line 423: please cite the most recent FAO standard

- *We have revised into the more recent FAO 2006 guideline*

Line 443-446: difficult to follow the sentence

485 - *We have reworded to "The simulated rotational period was 120, 110, 90 and 70 years from the north to south region, respectively. Two thinnings were conducted for the two north region with three thinnings for the two south regions" to be more clear.*

Line 449-450: The model was spun up for 10 years, and then a clear cut is simulated??? How do you determine the initial SOM content and soil stoichiometry? How big are the
490 effects of initial SOM status?

- *We have described the model design and setup in the beginning of section 3.3 "The results were based on simulated forest development with daily resolution over a rotation period from stand age 10 years until 10 years after final harvesting. The 10 years after final harvest were included to cover the potential nutrient leaching during*
495 *the regeneration phase as in Gärdenäs et al. (2003). The trees in all regions were assumed to be planted in 1961, and the period 1961 to 1970 was used as a spin-up period."*
- *The initial conditions for both plants and soil have been described in more detail in the newly added section 3.4 as following: "The two vegetation layers were initialized as*
500 *bare ground with a small amount of C, N and P mass in seedling to start vegetation growth. Initial conditions for solid inorganic P content, soil organic matter content, soil stoichiometry were reported in Table 1. Initial soil organic P pools (Table 1) were partitioned between soil litter (5%) and humus pools (95%) analog to N partition in Svensson et al. (2008), and total amount of soil organic P decreased exponentially with*
505 *depth (Fransson and Bergkvist, 2000). Litter was assumed to be distributed down to 0.5 m while humus down to 1 m depth. The initial labile $P_i$ concentrations were set according to previous data from similar Swedish forest sites (Kronnäs et al., 2019; Fransson and Bergkvist, 2000). Soil pH was set according to the NSFI data and kept constant over the simulation period (Table 1). The initial value of soil organic P for the*
510 *soil profile was estimated by the available measurements of soil organic matter N/P ratios, performed at the same forest monitoring sites of the Swedish Forest Agency (Wijk, 1995; Akselsson et al., 2015) where leaf nutrient content were sampled. However, only the organic N/P ratio at the O horizon was measured at most sites. Thus, in our calculations of the total stock of soil organic P, we assumed that the mean N/P*
515 *ratio measured for the O horizon also extends to the other horizons in the default model run. Model uncertainties associated with this assumption were assessed by including a range in the soil N/P ratios, 10-25, in the sensitivity analysis (e.g. Fig. 5)."*

Line 450-452: difficult to follow. Unclear to me what are the plant components and how are they treated

-   *We have changed this into "Following general forest management guidelines, it was assumed that during thinnings 20% of the stem is removed while 5% transforms into litter (Swedish Forest Agency, 2005). For leaves and roots, it was assumed that 25% transforms into litter. For all of the regions, one clearance – during which 60% of the stands is removed - was applied at the end of the stand age year 10 after spin-up. During final felling, 5% of trees are remained intact, and it is assumed that 90% of the stem is harvested and 5% becomes litter, while all leaves and roots become litter. " to be more clear.*

Line 470: "chronicle"? I am not sure that is the right word here???

-   *Changed to "describe".*

Line 477: This is a very unrealistic assumption; please see Yu et al. 2020 GMD

-   *We have now added a new section 3.4 to describe the initial conditions for both plants and soil.*
-   *We also added some few additional soil N and P content at the humus layer where leaf N and P content were measured into Table 1. However, still lacks soil P content of the layers below.*
-   *We further clearly stated what assumptions we made, and how we made an sensitivity analysis to address the uncertainty raised by this assumption as following: "The initial value of soil organic P for the soil profile was estimated by the available measurements of soil organic matter N/P ratios, performed at the same forest monitoring sites of the Swedish Forest Agency (Wijk, 1995; Akselsson et al., 2015) where leaf nutrient content were sampled. However, only the organic N/P ratio at the O horizon was measured at most sites. Thus, in our calculations of the total stock of soil organic P, we assumed that the mean N/P ratio measured for the O horizon also extends to the other horizons in the default model run. Model uncertainties associated with this assumption were assessed by including a range in the soil N/P ratios, 10-25, in the sensitivity analysis (e.g. Fig. 5)."*

Line 485-487: One specific question to Table 2 is that, why the decomposition and uptake rates for different latitudes are different, given that the temperature response function already accounts for the difference in temperature? If they are calibrated separately, what is the meaning for choosing a climatic gradient??? Table 3: I would recommend running a full sensitivity test with parameters in this table

-   *We have now made a few global sensitivity analysis with all the newly added P parameters and some N relevant parameters, and documented those in detail in supplement to the paper.*
-   *We have added Svensson et al. 2008 and He et al. 2018 in the main texts.*

Line 519-521: difficult to follow the sentence

-   *We have changed into "Thus, the measured P leaching also contains the P leaching from upstream. DOP were not measured for the regions. Thus the so call measured DOP was calculated as the difference between the measured total P and PO$_4$. This*

*means the "measured DOP" may contain both our simulated fractions DOP and particular phosphorus." to be easier to follow what we mean.*

4. Results

Line 561: confusing, please rephrase

- *We have revised into "The total N uptake of the short-cut uptake from the organic N pools decreased from North to South (Table 4)." to make it more clear*

Line 564: why the new Coup-CNP C sequestration rates are so different from previous studies of the same regions?

- *We thank referee's comment on soil sequestration rate but we respectively disagree with the interpretation of these C sequestration rates. First, Previous model results did not consider P, and we had discussed our newly introduced P cycle has clear impacts on C sequestration rates (Fig 5b). Second, we also had different set up with previous settings (see response above), thus a different C sequestration rate could be expected, given the soil sequestration rate were a net result of a number of C fluxes. However, the general trends were the same where an increasing soil C sequestration rate moving towards the south and we consider these results were rather in according with previous results not differs.*

Line 573-575: I only see that the P leaching is very small, which may infer that it has a small effect. But the fact that P leaching accounts for 30% of P deposition does not lead to the conclusion that "a small effect on the system". I guess the key point here is that both P deposition and P loss are very small compared to other fluxes, e.g. plant P uptake

- *We agree and the intention was to compare to the outflow flux to the internal flow. We have revised the texts to "The data show that P losses through leaching were small compared to the internal fluxes, i.e., they account for approximately one-third of the annual deposition input".*

Section 5.2: the rotation period,10 years to10 years after the final felling, makes it a bit difficult to understand the results in figure 4, particularly the plant growth and change in plant in panel A. For me it is very difficult to judge how much of the changes in plant and soil pools are due to the very short spin-up time (10 years)? Is it possible to run a real spin-up to ensure a more stable state of the soil pool? Also, I did not fully understand why the pool size of 10-year-old trees differ so much in N and P size, to me it seems to be the effect of model initialization and spin-up.

- *We have made it clear about our model design and its rational at the beginning section of 3.3: "The 10 years after final harvest were included to cover the potential nutrient leaching during the regeneration phase as in Gärdenäs et al. (2003)."To avoid misunderstanding, we also added "Plant growth in a) represent the net primary production." In Fig.4 captions to explain the figure more in detail.*
- *For spin up, see response above*
- *For initialization, we have a new section 3.4 model forcing, initial and boundary conditions to describe those in more detail, also see response above*

5. Discussion

Section 6.1: all the studies that are compared to in the section are modeling studies, which should be made very clear.

- *We have added "other modelling studies" in the section title to make clear that are modelling studies compared.*

Section 6.2: In general, the discussion is interesting and the findings are encouraging. However, I do have an understanding problem regarding the soil N/P ratio. From the description in the method part, the soil N/P ratio seems to be a parameter in the sensitivity analysis. But its value is not reported in Table 3, and it seems that it is also not a constant value from Figure 3d. A more methodological problem is, only three parameters were tested in the sensitivity analysis, and the result for one parameter was presented. How could one conclude that this one is the most important parameter for the ecosystem? As I mentioned before, if this is the first study of the Coup-CNP, a better-designed sensitivity test should be performed. I am very convinced by the authors that soil N/P is an important indicator of Swedish forests, but I am convinced by the way it was accidentally chosen in this study.

- *We have now conducted global sensitivity analysis for all the newly parameters and summarized the results in the supplement. Please also see the response above for the sensitivity and extra model experiments comments.*
- *The global sensitivity analysis show the initial soil organic P (thus soil N/P ratio), together with the short-cut N and P uptake coefficients show the highest importance in determining the modelled C, N and P cycling (see supplement section 3).*
- *We have now added a separated section in the main paper, "Section 3.5 sensitivity analysis" to briefly describe the global sensitivity analysis and links to the key parameter sensitivity analysis reported in the paper as following: "We conducted a global sensitivity analysis of the new Coup-CNP model to its parameterization (n=34) using a Monte-Carlo based sensitivity analysis method to assess the stability and robustness of the model with respect to its parameter values. The sampled parameters and their ranges (Table S.1), model design and global sensitivity results (Table S.2, S.3, S.4) were reported in detail in the supplement. Based on these simulations and parameter sensitivity rankings, we select three most important parameters (n=3), which has a strong effect on the model outcome to further form a new set of model runs, which are used for the model sensitivity analysis presented in this paper. The selected three parameters are initial soil humus P, short-cut N uptake rate and short-cut P uptake rate (Table S.2, S.3), all strongly regulates the soil N and P availability."*

Line 676: where does this conclusion come from? increasingly P limited with time or latitude, or another gradient?

- *We have added with "decreasing latitude" to make it more clear*

Line 682: have you checked if the threshold is the same for pine and spruce? if not, please be specific about tree species

645

- *We have now added the tree species in the main texts: "In Swedish spruce forests (Picea abies L. Karst.), leaf N/P ratios below 7 are normally considered an indicator of N limitation, while ratios above 12 signal P limitation (Rosengren-Brinck and Nihlgård, 1995; Yu et al., 2018). Linder (1995) has previously reported an optimal N/P ratio of 10 for spruce forests in northern Sweden. Similar optimal N/P ratio for pine forest (Pinus silvestris) (Ingestad, 1979; Tarvainen et al., 2016).".*

650

**Reply to Anonymous Referee #2**

We thank Anonymous Referee #2 for your positive comments and constructive suggestions to our manuscript. Below are our changes made to address the comments; The referee comments in normal font and our response in italics.

655

The paper of He et al., brings us a model that couples P into an existing CN model. It is an interesting study with special focus on mycorrhizal fungi, which is important in P dynamics, but has yet to be adequately represented in current literature. My major concern though, is

660    that the model is heavily parameterised with great details and many parameters, but the model performance is systematically biased. Figure 2 and Table 4 evaluated modelled tree biomass, leaf C:N, leaf C:P, leaf N:P and P leaching against measurements. First of all, for a model that covers many aspects of C, N, P dynamics, variables evaluated here are not adequate to show the model performance. Secondly, the model systematically overestimate

665    leaf C:P (all sites) and leaf C:N (3 out of 4 sites), and underestimate P leaching (all sites). I am not convinced that the model does a good job in capturing the system. Additional work and data are needed to understand the model dynamics and thoroughly assess the model performance.

- *We have now conducted global sensitivity tests including all the P parameters and*
670        *few N related parameter (n=34). The tested parameters, model approach and sensitivity analysis results, including parameter sensitivity for different selected model variables, are given in the newly added supplement (details see section 1-2 in supplement). Briefly, the global sensitivity results (i.e., Table S.2; Table S.3; Table S.4) show that parameter importance/sensitivity differs in terms of selected out*
675        *variables, also in terms of studied region. Overall, our global sensitivity analysis show the stability and robustness of the model (see section 3 in supplement). More importantly, the global sensitivity analysis results further elucidate the model behavior and demonstrate the model ability of capturing the system response to the four regions. Specifically, C and N change in plant and soil is shown to be mostly*
680        *sensitive to the short-cut N uptake rate ($o_{uptNhumus}$) (Table S.2) for the northernmost 64ºN region,. This is expected as the region is identified as being N limited. But for the southernmost 56ºN region, C and N change in plant and soil is shown to be more sensitive to the initial soil organic P in humus (Table S.3), as the region is P limited. This confirms our results from the previous model default runs and solidate that the*
685        *model can capture the system response to the four regions.*
- *We have now revised the datasets for model evaluation section 3.2 to describe the data more in detail, "Thus, the measured P leaching also contains the P leaching from upstream. DOP were not measured for the regions. Thus the so call measured DOP was calculated as the difference between the measured total P and $PO_4$. This means the*
690        *"measured DOP" may contain both our simulated fractions DOP and particular phosphorus."*
- *Again we would like to highlight our aim was to demonstrate model behavior and the implication of the newly added P in the model structure. The intention was not to make a site-specific detailed model calibration.*

695 In addition, I feel it is quite difficult to follow the model description. Sometimes there are logical issues related to terminology and the separation among system compartments (please see detailed comments below). Sometimes it is due to lack of critical information in P cycling in the main text, for example, P dynamics in vegetation (allocation, resorption etc.), through mineralization etc. It might be better to put part of the information in the appendix into the

700 main text, or at least have some overall description of these processes in the main text and point to the appendix for detailed information. The goal is to give the reader a complete picture of P cycling the model tracks.

- *We have merged previous section 2 and 3 and substantially rewritten the entire section. We have changed the model description organization by starting with*

705 *describing the new P model concepts and defining its pools and assumptions in section 2.2. Then starting the detailed process and equation description in sub-sections 2.2.1-2.2.6. "Soil inorganic phosphorus dynamics and nutrient short-cut uptake" has now been moved to section 2.2.3. We have briefly described the processes that detailed described in appendix, and added linkages (i.e. equation*

710 *numbers, sections in appendix) to the appendix when possible to make it easier for readers to follow.*

The novel part of this model, from my perspective, is related to symbiotic mycorrhizal fungi. I did not find any observations to initialise, evaluate model performance or constrain model parameters related to this part. It is also not clear what is the advantage of incorporating

715 detailed symbiotic mycorrhizal fungi, how it affects system dynamics, what are the novel model behaviours due to this part? I feel these questions are worth answering to persuade the reader that the model is advantageous and worth the great details..

- *We have added the following in the introduction section to briefly summarize the theory and few previous model studies that highlights the important of symbiotic*

720 *fungi. This is also used to explain and motivate the organic nutrient uptake concept (closely linked with fungi module) in Coup-CNP: "The organic nutrient uptake in their model was to mimic the additional pathway that plant can utilize organic nutrients by biochemical mineralization either in symbiosis with mycorrhizal fungi or root exudates (e.g. Schachtman et al., 1998, Gärdenäs et al. 2011, Richardson et al.*

725 *2005). However, in Orwin's model, plant growth was static; thus plant-soil or plant-environment interactions were largely ignored. Our model (Eckersten and Beier, 1998; He et al., 2018) also includes a nutrient-short-cut uptake as a process in the rhizosphere. The assumption is that nutrients released by biochemical mineralization are instantly taken up by the symbiotic microbes and/ or the plants, thereby by-*

730 *passing the soil matrix solution."*

-

Detailed comments:

BeforeLine65-70, CMIP6 model results are openly available now. One model (probably the only one) that has land P component is from CSIRO, Australia. The name of the earth system

735 model is ACCESS and land component is CABLECNP.

- *We have deleted this sentence but rephrased the sentence into the following to be precise "Nevertheless, the P cycle is seldom incorporated into ecosystem model*

*structures. Incorporating the P cycle is essential in improving global models as a tool for assessment of climate-C cycling interactions"*

740    Lines70-75, whether CNP models from Goll et al., 2012; Wang et al., 2010; Yang et al., 2014 are simplified are context dependent. As far as I know, these models incorporated key processes in C, N, P, water and energy dynamics and take into account coupling and interactions across spatial-temporal scales. They are not necessarily simpler than the model presented here.

745    - *We have deleted the misleading word "simple", and also have added the missing references of the global vegetation models that have phosphorus cycle.*

Line 75-80. Models in Medlyn et al 2016 are not earth system models per se. They are process-based vegetation models. ESMs have coupled land, atmosphere, ocean etc. Some models might be used as the land component of some ESMs. Some models may not be
750    directly coupled.

- *We will changed to "global vegetation model" to be precise.*

Line 80-85. Low eco2 response do not imply "In other words, the vegetation is rather inflexible to increase P uptake". There are many factors come into play. Without CNP, the models have difficulties in capturing nutrient limitation on CO2 response. In nutrient limited
755    locations, nutrient limitation is likely to reduce eco2 responses. And it is not only about the uptake capability. It is also related to nutrient availability.

- *We have deleted the misleading statements and keep it as: "The CNP models that explicitly considered the P dependency of C assimilation predicted the lowest $eCO_2$ response".*

760    Line 140-150, "The main model structure is a one-dimensional, vertical layered soil profile including plants." This sentence is confusing. How vertical soil profile could include plants ?

- *We have rephrased this into "The main model structure is a one-dimensional, vertical model, with one or two layers of vegetation (for example a tree and field layer as in this application) on a multi-layered soil profile".*

765    Line 150-155, the concept of "big leaf" model assumes canopy carbon fluxes have the same relative responses to the environment as any single unshaded leaf in the upper canopy. You have two layers, trees and understory. Normally when people talk about "big leaf" model, it does not simulate light competition between up- vs. understory plants.

- *We have reworded this into "Multiple-big leaves" model concept, i.e. two vegetation*
770    *layers, trees and understory plants was used.*

Line 170-171, the naming convention is quite confusing. By common definition, inorganic P is part of soil mineral P.

- *We have re-described the pools as "The soil inorganic P has new and renewed state variables. New is the soil solid inorganic $P_{solid}$, a lumped pool containing primary and*
775    *secondary mineral compounds containing P such as apatite (and occluded P) (Smeck,*

 *1985; Wang et al., 2007). $P_{ilab}$ is the sum of phosphate ions absorbed and those in soil solutions, analog to the mineral pool in salt tracer representation in CoupModel (Gärdenäs et al., 2006). Instantaneous equilibrium between adsorbed and soil solution are assumed. Plant and microbes take up phosphate ions from the $P_{ilab}$. $P_{isol}$ can be compared with the sum of N state variables $NH_4^+$ and $NO_3^-$ while being an intrinsic part of $P_{ilab}$ (Fig. 1)."*

- *We have renamed the previous mineral P pool as "soil solid inorganic $P_{solid}$" in the entire paper to avoid confusion*

Line 180. The description of different P pools is rather confusing. If "soil mineral P is the total soil P without organic Po and labile P", how could you estimate it with total P content and bulk density. When we measure bulk density, we do not exclude the contribution from the organic matter.

- *We have revised the naming of the different P pools and clarified that we mean. Please also see response above. The calculation was given by using equ 2 and the content in Table 1.*

Line 180-185. What do you mean by "fresh plant residues"? If plant residue that stays above soil, but it is not fresh (e.g., it is from the last year), do you exclude it from litter?

- *We have redescribed the soil organic pools as: "Soil organic P is divided into three state variables by litter ($P_{Litter}$), humus ($P_{Humus}$), and dissolved organic ($P_{DOP}$) in every soil layer analog to C and N in CoupModel v2.0 (Fig. 1)".*

Line 180-185, "In CoupModel, soil litter could be further divided into two litter pools: one which contains readily decomposing materials (e.g., plant leaves and fine roots) and another for decomposition-resistant litter (e.g., stems and coarse roots)". If you do not represent these in your model, please skip these texts to reduce confusion.

- *We deleted this to reduce confusion.*

Line 190-195. Do you take into account the hysteresis in P adsorption/desorption?

- *No, we assume instant equilibrium between the labile and soluble P. We have described this assumption in section 2.2.2*

Line 170-205, you talked about litter pool, how do you treat soil organic matter/P pool? Do you only have humus pool? If so, non-symbiotic soil microbes are classified as litter in your model?

- *We have revised our definition of soil organic matter pools in section 2.2. "Soil organic P is divided into three state variables by litter ($P_{Litter}$), humus ($P_{Humus}$), and dissolved organic ($P_{DOP}$) in every soil layer analog to C and N in CoupModel v2.0 (Fig. 1). In this study, non-symbiosis microbes is included in the litter."*

Lines 210-215, "During certain seasons, plants can also capture mobile P (as well as mobile N) to prepare for rapid growth in the spring". What do you mean here? You mean plants take

815    up more P in other seasons other than Spring, store it and use it in Spring? How does it occur? What do you mean by mobile P(N)?

- *The following section was added in A4 plant litterfall to describe the pool and assumption more clearly "During litterfall seasons, plants can reallocate P and N from leaves to an internal, mobile storage to prepare for rapid growth in the spring, a known*
820      *mechanism to increase efficient use of nutrients (e.g. Aerts, 1996; Niemien and Helmisaari, 1996) (also see $m_{retain}$ in Table S.1 in supplementary)".*
- *Note the $m_{retain}$ is also included in the global sensitivity analysis and showed the modeled ecosystem C change is sensitive to this parameter in Table S.4*

825    Lines 220-225, I don't understand what do you mean by "In Coup-CNP, biochemical mineralization is defined as organic uptake". Biochemical mineralization and organic uptake are different processes.

- *We have added the following in the introduction section to explain the organic nutrient uptake concept better in Coup-CNP: "The organic nutrient uptake in their*
830      *model was to mimic the additional pathway that plant can utilize organic nutrients by biochemical mineralization either in symbiosis with mycorrhizal fungi or root exudates (e.g. Schachtman et al., 1998, Gärdenäs et al. 2011, Richardson et al. 2005). However, in Orwin's model, plant growth was static; thus plant-soil or plant-environment interactions were largely ignored. Our model (Eckersten and Beier,*
835      *1998; He et al., 2018) also includes a nutrient-short-cut uptake as a process in the rhizosphere. The assumption is that nutrients released by biochemical mineralization are instantly taken up by the symbiotic microbes and/ or the plants, thereby by-passing the soil matrix solution."*
- *The section 2.2.3 further describe the concepts and calculations as "Biochemical*
840      *mineralization, on the other hand, describes the release of $P_i$ through extracellular enzymatic releases (e.g., phosphatases by root exudates), which are driven by plant demand for nutrients (Richardson and Simpson, 2011). In Coup-CNP, biochemical mineralization is conceptually included in the nutrient short-cut uptake (called organic uptake in earlier CoupModel publications) and assumed to be driven by the*
845      *unfulfilled plant P demand after $P_{ilab}$ root uptake (equ A.8) but regulated by the availability (i.e. short-cut uptake coefficients in equ A.4)."*
- *Further, to avoid confusion with the uptake of organic molecules, we rename the organic P/N uptake into "short-cut P/N uptake" in the entire paper.*

850    Line 316, "wais" to "was"

- *Corrected.*

Line 535 – 540 and Figure 2. From Figure 2, the model systematically over-estimate Leaf C/P and leaf C/N ratio (except one site). Is it because an over-estimation of the leaf biomass? If there are coherent bias for all or most sites, it is not a neglectable issue. Figure 4. Why do
855    you plot plant growth in C flux but change in plant for P flux, please be coherent and consistent. Table 6, systematically underestimation of P leaching

- *To avoid misunderstanding, we also added the following texts "Plant growth in a) represent the net primary production." in Fig.4 captions to explain the figure more in detail.*

860 - *We have now revised the datasets for model evaluation section 3.2 to describe the compared leaching data more in detail, and make it clear that the leaching data also contains P leaching from upstream more. The texts have been modified into following: "Thus, the measured P leaching also contains the P leaching from upstream. DOP were not measured for the regions. Thus the so call measured DOP was calculated as the*

865 *difference between the measured total P and $PO_4$. This means the "measured DOP" may contain both our simulated fractions DOP and particular phosphorus."*

- *For model performance, see response above.*

 **Reply to editor comments**

We thank editor for your positive comments and constructive suggestions to our manuscript. Here are our responses to the comments; The editor comments are in normal font and our response in italics.

875 Thanks for preparing a revised version of the manuscript addressing my previous comments. I will accept now the manuscript for publication in the discussion forum and formally start the peer review process. However, your answer to my question on the type of dynamic update, with your respective answer about coupled partial differential equations, suggests that your presentation of equations in the text is not adequate, and that you would have to rewrite
880 many of the equations to make explicit the use of partial differential equations. You also would have to state more explicitly the boundary conditions and the initial conditions since these are factors that strongly influence the solution of the system of equations.
I accept the current version for the review process, but keep this comment in mind when preparing a revised version addressing reviewers' comments.

885 - *The equations in current paper is given in a form that reflects one time step and one of the layers that represent the entire soil profile. We think current forms have the advantage of being easier to follow. Therefore, we have added the following texts in the main texts at section 2.2 to describe the presentation of the equations, "For simplicity, the equations are given in a form that reflects one time step and one of the*
890 *layers that represent the entire soil profile."*
  - *We have further added texts to describe how CoupModel solve the partial differential equations in section 2.1 and give reference where more information is available as "They are numerically solved using an explicit forward difference model scheme (Euler integration, for more details see pp 400-401 in Jansson and Karlberg (2011)).*
895 *Explicit forward difference model means that at current time step, the size of a state variable is updated with the fluxes to and from the state variable during previous time step."*
  - *We have now added a separate sections of 3.4 Model forcing, initial and boundary conditions to describe these into more detail. The initial conditions for soil and plant*
900 *and the boundary conditions for water and heat are all given, if not then refer to the previous studies.*

**CoupModel (v6.0): an ecosystem model for coupled phosphorus, nitrogen, and carbon dynamics – evaluated against empirical data from a climatic and fertility gradient in Sweden**

Hongxing He[1], Per-Erik Jansson[2], Annemieke I. Gärdenäs[1]

[1]Department of Biological and Environmental Sciences, University of Gothenburg, POo Box 460, Gothenburg 40530, Sweden

[2]Department of Land and Water Resources Engineering, Royal Institute of Technology (KTH), 100 44 Stockholm, Sweden

*Correspondence to:* Hongxing He ([hongxing-he@hotmail.com](mailto:hongxing-he@hotmail.com)); and Annemieke Gärdenäs ([annemieke.gardenas@bioenv.gu.se](mailto:annemieke.gardenas@bioenv.gu.se))

**Abstract**

This study presents the integration of the phosphorus (P) cycle into CoupModel (Coup-CNP). The extended Coup-CNP, (which explicitly consider symbiosis between soil microbes and plant roots,) which explicitly consider mycorrhizal interactions enables simulations of coupled carbon (C), nitrogen (N) and P dynamics for terrestrial ecosystems with an explicit consideration of symbiosis between soil microbes and plant roots which explicitly consider mycorrhizal interactions. The model was evaluated against observed forest growth and measured leaf C/P, C/N and N/P ratios in four managed forest regions in Sweden. The four regions form a climatic and fertility gradient from 64°N in the North to 56°N in South Sweden with the mean annual temperature varying between 0.7-7.1 °C and the soil C/N and C/P ratios between 19.8-31.5 and 425-633, respectively. The growth of the southern forests was found to be P-limited, with harvested biomass representing the largest P loss over the studied rotation period. The simulated P budgets revealed that southern forests are losing P, while northern forests have their P budget in balance are close to in balance for Pare close to a steady state in P availabilityclose to in balance for P. SymbioticMycorrhizal fungi account for half of the total plant P uptake across all four regions, which highlights the importance of fungal-tree interactions in Swedish forests. Sensitivity analysis results demonstrated that the highest forest

growth occurs at a soil N/P ratio of 15 to 20. A soil N/P ratio above 15-20 resulted in decreased soil C sequestration and  P leaching, but significantly increased N leaching. With the  evaluation of the new Coup-CNP model, we demonstrate that P fluxes need to be further considered in studies of how climate change can influence C turnover and ecosystem responses. We conclude that the  inclusion of the P cycle is necessary  to make biogeochemical models more reliable tools for assessing long-term impacts of climate change and N deposition on C sequestration and nutrient leaching.

**1 Introduction**

Phosphorus (P) is an essential element for all photosynthesizing plants  in terrestrial ecosystems, with the P cycle coupled to Carbon (C) and Nitrogen (N) fluxes especially through  processes such as  decomposition of soil organic matter and  soil nutrient uptake  (Lang et al., 2016; Vitousek et al., 2010). A steep increase in the anthropogenic release of C and N to the atmosphere relative to P release has altered plant and soil nutrient stoichiometry, leading to new forcing conditions (Elser et al., 2007; Penuelas et al., 2013). For instance, numerous monitoring studies have revealed increasing N/P ratios in plants and soils, especially in forests from North America (Crowley et al., 2012; Gress et al., 2007; Tessier and Raynal, 2003) and Central and Northern Europe (Braun et al., 2010; Jonard et al., 2015; Talkner et al., 2015). Such trends are generally assumed to indicate that these ecosystems are shifting from being N limited to either co-limited by both N and P or P limited (Elser et al., 2007; Saito et al., 2008; Vitousek et al., 2010; Du et al. 2020). Human activities are expected to continue increasing the atmospheric N  deposition. from the atmosphere and, As such P availability and its dynamics will become progressively more important in regulating the biogeochemistry of terrestrial ecosystems and amplifying feedback s relevant to climate change, e.g. limiting the growth response of plants to increased temperature (Deng et al., 2017; Fleischer et al., 2019; Goll et al., 2017).

Nevertheless, the P cycle is seldom incorporated into ecosystem model structures. Incorporating the P cycle is essential in improving global models as a tool for assessment of climate-C cycling interactions ( Reed et al., 2015). Most of the process-based models that can simulate P cycling were specifically developed for agricultural

systems and focus on the soil processes, e.g., EPIC (Jones et al., 1984, Gassman et al., 2005), ANIMO ( Groenendijk et al., 2005), and GLEAMS (Knisel and Turtola, 2000). A few catchment-scale models  focus on surface water quality, e.g. SWAT (Arnold et al., 2012), HYPE (Arheimer et al., 2012), and INCA-P (Jackson-Blake et al., 2016). However, none of these models explicitly consider dynamics of plant litter inputs, nutrient mineralization, or how nutrient uptake may  influence photosynthesis.  However, the C response to P limitation has recently been studied through several empirical and field studies (Van Sundert et al., 2019; Du et al., 2020). For example, Van Sundert et al. (2019) showed that  the productivity of European beech (*Fagus sylvestris*) forests  is negatively related to the soil organic carbon concentrations and mineral C/P ratios. A few global vegetation models have  included a  P cycle to study how it affects the C cycle (Goll et al., 2012, 2017; Wang et al., 2010; Yang et al., 2014; Zhu et al., 2016; Thum et al., 2019). These P enabled models differ in how they describe soil P dynamics (implicitly or explicitly of symbiotic mycorrhiza and other soil microbes), plant P use and acquisition strategies, which results in considerable uncertainty in the C response (Fleischer et al., 2019; Medlyn et al., 2016; Reed et al., 2015). Medlyn et al. (2016) applied six  global vegetation models including two coupled Carbon-Nitrogen-Phosphorus (CNP) models (CABLE and CLM4.0-CNP) to study the response to elevated $CO_2$ ($eCO_2$) of the C cycle of the Eucalyptus-Free-Air $CO_2$ Enrichment experiment and found large variations, ranging from 0.5 to 25%, in predicted net primary productivity. The CNP models that explicitly considered  P dependency of C assimilation predicted the lowest $eCO_2$ response.  Yu et al. (2018)  included the P cycle into the field-scale biogeochemical model – ForSAFE and applied the model to study the P budget of a southern Swedish Spruce forest site. They concluded that the P supply by weathering  was small compared to the internal turnover by mineralization of soil organic matter . Fleischer et al. (2019) demonstrated that four CNP models, when

[revised manuscript text omitted]

**2 Description of mModel structure and  phosphorus model  linked to the phosphorus pool**

**2.1 Brief description of CoupModel (v5)**

The CoupModel platform (coupled heat and mass transfer model for soil–plant–atmosphere systems) is a  process-based model designed to simulate water and heat fluxes, along with C and N cycles, in terrestrial ecosystems (Jansson, 2012). The main model structure is a one-dimensional, vertical model, with one or two layers of vegetation (for example a tree and field layer as in this application) on  a multi-layered soil profile . The core of the model consists of five sets of coupled partial differential equations, one for  water, heat, C, N, and P  cycles (the later one in v6.0), respectively. They are numerically solved using an explicit forward difference model scheme (Euler integration, for more details see pp 400-401 in Jansson and Karlberg (2011)). Explicit forward difference model means that at current time step, the size of a state variable is updated with the fluxes to and from the state variable during previous time step. In this application, we used a daily  time step for all five, but a smaller time step was applied for the water and heat calculations during specific events with peaks in water and/or heat flow such as during snow melting  to ensure the numerical stability and accuracy. The model is driven by climatic data – precipitation, air temperature, relative humidity, wind speed, and global, i.e. sum of direct and diffuse short-wave incoming radiation – and can simulate ecosystem dynamics with daily resolution. Vegetation is described using the "multiple-big leaves" concept, i.e. two vegetation layers, trees and understory plants, are simulated taking into account mutual competition for light interception, water uptake and soil N (Jansson and Karlberg, 2011). The model and technical description (Jansson and Karlberg (2011) is freely available at www.coupmodel.com. A presentation of CoupModel use, calibration and validation is given in Jansson (2012).. He et al. (2018) introduced an explicit plant-mycorrhizal representation  (CoupModel v5).

**2.2 Phosphorus cycle representation in CoupModel (v6.0)**

CoupModel (v6.0), hereafter called Coup-CNP, was extended with P cycle representation to enable simulations of coupled C, N and P dynamics for terrestrial ecosystems with an explicit consideration of symbiosis between soil microbes and plant roots.  Coup-CNP has P state variables

and fluxes representing different plant parts, symbiotic microbes, soil organic P forms ($P_o$, P that is bound to organic C in the soil) and soil inorganic P, $P_i$, forms (Figure 1). For clarity of the coupling between C, N and P cycles, also the C and N state variables and major N and N+P fluxes are given in Figure 1.

P in the plants is partitioned into grain, leaf, stem, coarse root, fine root, and P in symbiotic microbes, analog as for C and N in CoupModel v5.0 (Fig. 1). In this paper, we use mycorrhizal fungi as the role model of plant-microbe symbiosis, the same concept is also applicable for other symbiosis microbes. Soil organic P is divided into three state variables by litter ($P_{Litter}$), humus ($P_{Humus}$), and dissolved organic ($P_{DOP}$) in every soil layer analog to C and N in CoupModel v2.0 (Fig. 1). In this study, non-symbiosis microbes is included in litter. The soil inorganic P has new and renewed state variables. New is the soil solid inorganic $P_{solid}$, a lumped pool containing primary and secondary mineral compounds containing P such as apatite (and occluded P) (Smeck, 1985; Wang et al., 2007). $P_{ilab}$ is the sum of phosphate ions absorbed and those in soil solutions, analog to the mineral pool in salt tracer representation in CoupModel (Gärdenäs et al. 2006). Instantaneous equilibrium between adsorbed and soil solution are assumed. Plant and microbes take up phosphate ions from the $P_{ilab}$. $P_{isol}$ can be compared with the sum of N state variables $NH_4^+$ and $NO_3^-$ while being an intrinsic part of $P_{ilab}$ (Fig 1.)

~~Coup-CNP, soil phosphorus is conceptually divided into inorganic P ($P_i$, phosphate ions, e.g., $H_3PO_4$, $H_2PO_4^-$, $HPO_4^{2-}$, $PO_4^{3-}$), soil solid mineral P ($P_m$) and soil organic P ($P_o$, P that is bound to organic C). Overall, three inorganic P pools (soluble ($P_{isol}$), labile ($P_{ilab}$), and soil mineral ($P_m$)) and three organic P pools (litter ($P_{olitLitter}$), humus ($P_{ohumHumus}$), and dissolved organic ($P_{dop}$)) are used to represent the soil P (Fig. 1). The labile $P_{ilab}$ defines the available $P_i$ for plants including the readily $P_i$ exchanges with soil solutions, $P_{isol}$. Soluble pool ($P_{isol}$) is thus a part of labile pool ($P_{ilab}$) that are dissolved and not adsorbed. The soil solid mineral $P_m$ is defined as the total soil P without organic $P_o$ and labile $P_{ilab}$ (Hedley and Stewart, 1982). Thus, soil solid mineral $P_m$ is a lumped pool containing primary and secondary mineral compounds containing P such as apatite (and occluded P) (Smeck, 1985; Wang et al., 2007). The three organic P pools follow the division of the C and N cycles as that in Svensson et al. (2008). Soil litter consists of plant residues (both current year and years before) and non-symbiotic microbes, while humus is the organic residue from litter decomposition. A fraction of the $P_o$ in the $P_{olitLitter}$ and $P_{ohumHumus}$ pools may also form the dissolved $P_{dop}$ (Fig. 1).~~

The mycorrhiza describes a symbiotic association between fungus and the plants' fine roots: as such, it consists of C, N, and P pools that are separate from those of the plant. The mycorrhiza is further distinguished into the mycelia, which is responsible for N and P uptake (both in inorganic forms and nutrient short-cut from organic pools), and the fungal mantle, which covers the fine-root tips (He et al., 2018). Through plant litterfall, P is recycled and released back into the soil through mineralization. During litterfall seasons, plants can also capture mobile P (as well as mobile N) to prepare for rapid growth in the spring (also see $m_{retain}$ in Table S.1 in supplementary). The mobile P pool (not shown in Fig. 1) act as an internal storage in the plant. Coup-CNP use the mobile pool to mimic the nutrient reallocation or retranslocation process, a known mechanism to reduce dependence on nutrient uptake and increase nutrient recycle (e.g. Aerts, 1996; Niemien and Helmisaari, 1996).

We developed the P model in a way that 1) focusconcentrates on the key P processes that are most relevant processes for biogeochemical cyclingassessmentsdynamics, e.g., dynamic plant growth and P leaching, and 2) follows the conceptual structure of CoupModel as closely as possible. The key P processes are described in detail below. In appendix A are described processes that are analogous to those of the N cycle, e.g., atmospheric deposition, fertilization (A.1), mineralization-immobilization (A.2), plant growth and uptake (A.3), litterfall (A.4), leaching and surface runoff (A.5), and removal of plant harvest (A.6). The P model runs at the same time step as the models for C and N cycles, which can differ from the time step of the models for water and heat. The discretization of the soil includes common compartments thatfor P are linked follows that of to all elements C and N and abiotic conditions. For simplicity, the equations are given in a form that reflects one time step and For simplicity, the following description of the model concerns one of the layers that represent the entire soil profile. The symbols in this paper were designed to conform the CoupModel nomenclature in the following way: uppercase P means state variables, lowercase p means parameters related in P processes. 
[revised manuscript text omitted]

 .

**2.2.1 Weathering**

 By weathering Soil solid mineral $P_{solid\text{m}}$, is transformed into labile $P_{ilab}$.  (Fig. 1; equ 1). proportional to the pH response of the soil water solution (Guidry and Machenzie, 2000).~~ The weathering rate depends on soil pH and temperature (Guidry and Machenzie, 2000) and is calculated as,

$$P_{solid \to ilab} = k_w \times f(T) \times f(pH) \times P_{solid} \tag{1}$$

Where $P_{solid\text{m} \to ilab}$ is the flux rate of weathering (g P m$^{-2}$ day$^{-1}$), $k_w$ is a first-order integrated weathering rate coefficient (day$^{-1}$) which depends on lithology, rates of physical erosion and soil properties (Table 3), the erosion affect the weathering rate by reducing the pool size of $P_{solid}$ (equ A.14). $f_{\text{w}}(T_s)$ and $f_{\text{w}}(pH)$ are response functions of soil temperature, $T_s$, and soil $pH$, $P_{solid\text{m}}$ is the size of the $P_{solid\text{m}}$ pool (g P m$^{-2}$), determined by,

$$P_{solid} = \delta_P \times \rho_{bulk} \times \Delta z_{layer} \times 10^6 \tag{2}$$

Where $\delta_P$ is the prescribed $P_{solid}$ content for each soil layer (g P g dry soil$^{-1}$), with reported ranges from $10^{-4}$ to $1.5 \times 10^{-3}$ g P g soil$^{-1}$ (Yang et al., 2014), $\rho_{bulk}$ is the dry bulk density for each soil layer (g cm$^{-3}$), and $\Delta z_{layer}$ is the thickness of the simulated soil layer (m).

The temperature effect can be expressed as an Arrhenius function (3), where $E_{a,wea}$ is the activation energy parameter (J mol$^{-1}$) for minerals (i.e., apatite), available from empirical studies, $R$ is the gas constant (J K$^{-1}$ mol$^{-1}$), $T_s$ is the simulated soil temperature in °C, $T_{s,0}$ is a parameter (°C) which normalize the function $f_{\text{w}}(T_s)=1$ and $T_{abszero}$ is -273.15 °C.

$$f\left(T\right) = e^{\left(-\frac{E_{a,wea}}{R} \times \left(\frac{1}{T_s + T_{abszero}} - \frac{1}{T_{s,0} + T_{abszero}}\right)\right)} \tag{3}$$

Alternatively, the existing Ratkowsky function, O'Neill function or $Q_{10}$ method can be used to determine the temperature response in CoupModel.

The effect of soil pH on weathering can be calculated through (equ 4), where $n_H$ is a parameter that describes the sensitivity soil pH when it differs from an optimal value $pH_{opt}$ for weathering (Table 3).

$$f(pH) = 10^{n_H \times |pH_{opt} - pH|} \tag{4}$$

**2.2.2 Inorganic soluble phosphorus dynamics**

~~The sizes of the $P_{isol}$ and $P_{ilab}$ pools are largely determined by chemical soil properties, e.g., anion exchange capacity and pH. The dynamics of these pools are regulated by physiochemical, e.g., adsorption/desorption, as well as biochemical processes, e.g., mineralization/immobilization. Part of the $P_{ilab}$ pool would quickly be adsorbed by soil water and colloids (Buendía et al., 2010; Stewart and Tiessen, 1987). These $P_i$ ions are normally loosely adsorbed to surfaces and can thus easily re-enter the $P_{ilab}$ pool through the desorption process (McGechan and Lewis, 2002). Both Cole et al. (1977) and Olander and Vitousek (2005) showed that whenpoolsAs the CoupModel provides daily resolution, w$P_{ilab}$ pools(Barrow, 1979). The relationship between the $P_{isol}$ and $P_{ilab}$ pools is normally represented by empirical equations, i.e., Freundlich and Langmuir isotherms (McGechan and Lewis, 2002). In this study, t~~The modified Langmuir isotherm (Barrow, 1979) was used to model the fast and reversible sorption process within $P_{ilab}$.

$$P_{ilab,con} = p_{max,ads} \times \frac{P_{isol}}{c_{50,ads} + P_{isol}} \tag{5}$$

Where $P_{ilab,con}$ is the concentration of labile pool (g P g soil$^{-1}$) calculated similarly by equation (2) as $P_{ilab,con} = P_{ilab} / (\rho_{bulk} \times \Delta z_{layer} \times 10^6)$ , $p_{max,ads}$ is the maximum sorption capacity of the labile pool (g P g

soil$^{-1}$), and $c_{50,ads}$ is an empirical parameter corresponding to 50% of P saturation (g P m$^{-2}$) (Table 3).

**2.2.3 Soil inorganic phosphorus dynamics and nutrient short-cut uptake**

Atmospheric P deposition is assumed to directly flow to the $P_{ilab}$ pool in the uppermost soil layer (equ A.1 in Appendix A). If mineral $P_i$ fertilizer is applied at the soil surface, the $P_i$ first enters an undissolved fertilizer pool, after which $P_i$ from this pool gradually dissolves into the labile P pool following a decay-type function (equ A.1). P could also be added as an external organic substrate (faeces or manure). In this case, P moves to the surface faeces ($P_{ofae}$), litter ($P_{olitLitter}$), and labile ($P_{ilab}$) P pools according to the composition of the manure. $P_i$ in the $P_{isol}$ pool and dissolved organic $P_{dopDOP}$ can be transported by water flows between layers or from a layer to a drainage outlet (equ A. 12-13). The soil surface layer may also lose solid mineral $P_{solid}$ by erosion, which is driven by surface runoff (equ A.14).

P mineralization is conceptually divided into biological and biochemical mineralization (equ A.2-A.6) following McGill and Cole (1981). Biological mineralization, which is regulated by temperature and moisture, represents microbe-mediated oxidation of organic matter, during which nutrients (P and N) are immobilized by implicit non-symbiotic microbes or transferred from litter to humus (Fig. 1; equ A.2). Biochemical mineralization, on the other hand, describes the release of $P_i$ through extracellular enzymatic releases (e.g., phosphatases by root exudates), which are driven by plant demand for nutrients (Richardson and Simpson, 2011). In Coup-CNP, biochemical mineralization is conceptually included in the nutrient short-cut uptake (called organic uptake in earlier CoupModel publicationsliterature) and assumed to be driven by the unfulfilled plant P demand after $P_{ilab}$ root uptake (equ A.8) but regulated by the availability (equ A.4). The assumption is that under P limited conditions, plant roots and symbiotic fungi bypass the $P_{ilab}$ pool, and obtain mineralized $P_i$ directly from the organic $P_{olitLitter}$ and $P_{ohumHumus}$ pools (Fig. 1; equ A.4).

**32.2.43 Plant growth under phosphorus and nitrogen limitationstress**

Plant photosynthesis is modelled by a "light use efficiency" approach (Monteith, 1965, equ 6)). We adopted Liebig's law of minimum to simulate the effects of multiple nutrient stress on plant growth (Liebig, 1840)). This approach assumes that the nutrient (N, P) which has a the smallestsmaller supply relative to the corresponding plant demand will limit growth (equ. 7). Plant demand was estimated through defined optimum ratios (equ A.9).

$$C_{a \to plant} = \varepsilon_L \times f(T_{leaf}) \times f(nutrient) \times f\left(\frac{E_{ta}}{E_{tp}}\right) \times R_S \qquad (6)$$

$$f(nutrient) = min\left(f\left(C/N_{leaf}\right); f\left(C/P_{leaf}\right)\right) \qquad (7)$$

Where $C_{a \to plant}$ is the plant carbon assimilation rate (g C m$^{-2}$ day$^{-1}$), $\varepsilon_L$ is the coefficient for radiation use efficiency (g C J$^{-1}$), $f(T_{leaf})$, $f(nutrient)$ and $f(E_{ta}/E_{tp})$ are response functions of leaf temperature, leaf nutrient status (N$_{leaf}$, P$_{leaf}$) in proportion to its C content, and water, respectively, and $R_s$ represents radiation absorbed by the canopy (J m$^{-2}$ day$^{-1}$). Details concerning $f(T_{leaf})$, $f(E_{ta}/E_{tp})$, as well as growth and maintenance respiration, can be found in Jansson and Karlberg (2011). Plant demand was estimated through defined optimum ratios (equ A.9). The nutrient response function $f(nutrient)$ which includes P is described below.

As is the case with N, the photosynthesis process responds to the leaf C/P ratio was modelled according to the work of Ingestad and Ågren (1992). Hence, below an optimum C/P ratio ($p_{CP,opt}$), the photosynthesis is not limited by P, while between $p_{CP,opt}$ and $p_{CP,th}$ the response function decrease as a linear function from one to zero,

$$f(C/P_{leaf}) = \begin{cases} 1 & C/P_{leaf} < p_{CP,opt} \\ 1 + \left(\dfrac{C/P_{leaf} - p_{CP,opt}}{p_{CP,opt} - p_{CP,th}}\right) & p_{CP,th} \leq C/P_{leaf} \geq p_{CP,opt} \\ 0 & C/P_{leaf} > p_{CP,th} \end{cases} \qquad (8)$$

Where C/P$_{leaf}$ is the actual leaf C/P ratio and $p_{CP,opt}$ and $p_{CP,th}$ are parameters that vary between plant species (Table 3). The leaf C/P ratio is calculated at each time step with the leaf state variables pool sizes of C and P. The pool sizes are iterated with the inflow and outflow fluxes.
* * *

[revised manuscript text omitted]

**4̶.̶2̶ ̶M̶o̶d̶e̶l̶ ̶d̶e̶s̶i̶g̶n̶ ̶a̶n̶d̶ ̶s̶e̶t̶u̶p̶**

T̶h̶e̶ ̶d̶e̶v̶e̶l̶o̶p̶m̶e̶n̶t̶ ̶o̶f̶ ̶m̶a̶n̶a̶g̶e̶d̶ ̶f̶o̶r̶e̶s̶t̶s̶ ̶w̶a̶s̶ ̶s̶i̶m̶u̶l̶a̶t̶e̶d̶ ̶i̶n̶ ̶d̶a̶i̶l̶y̶ ̶r̶e̶s̶o̶l̶u̶t̶i̶o̶n̶ ̶o̶v̶e̶r̶ ̶a̶ ̶r̶o̶t̶a̶t̶i̶o̶n̶ ̶p̶e̶r̶i̶o̶d̶ ̶f̶r̶o̶m̶ ̶s̶t̶a̶n̶d̶ ̶a̶g̶e̶ ̶c̶l̶a̶s̶s̶ ̶1̶0̶ ̶u̶n̶t̶i̶l̶ ̶1̶0̶ ̶y̶e̶a̶r̶s̶ ̶a̶f̶t̶e̶r̶ ̶f̶i̶n̶a̶l̶ ̶h̶a̶r̶v̶e̶s̶t̶i̶n̶g̶ ̶t̶o̶ ̶c̶o̶v̶e̶r̶ ̶t̶h̶e̶ ̶p̶o̶t̶e̶n̶t̶i̶a̶l̶ ̶n̶u̶t̶r̶i̶e̶n̶t̶ ̶l̶e̶a̶c̶h̶i̶n̶g̶ ̶d̶u̶r̶i̶n̶g̶ ̶t̶h̶e̶ ̶r̶e̶g̶e̶n̶e̶r̶a̶t̶i̶o̶n̶ ̶p̶h̶a̶s̶e̶ ̶a̶s̶ ̶i̶n̶ ̶G̶ä̶r̶d̶e̶n̶ä̶s̶ ̶e̶t̶ ̶a̶l̶.̶ ̶(̶2̶0̶0̶3̶)̶.̶ ̶T̶h̶e̶ ̶t̶r̶e̶e̶s̶ ̶i̶n̶ ̶a̶l̶l̶ ̶r̶e̶g̶i̶o̶n̶s̶ ̶w̶e̶r̶e̶ ̶a̶s̶s̶u̶m̶e̶d̶ ̶t̶o̶ ̶b̶e̶ ̶p̶l̶a̶n̶t̶e̶d̶ ̶i̶n̶ ̶1̶9̶6̶1̶;̶ ̶t̶h̶u̶s̶,̶ ̶t̶h̶e̶ ̶p̶e̶r̶i̶o̶d̶ ̶1̶9̶6̶1̶ ̶t̶o̶ ̶1̶9̶7̶0̶ ̶r̶e̶p̶r̶e̶s̶e̶n̶t̶e̶d̶ ̶t̶h̶e̶ ̶s̶p̶i̶n̶-̶u̶p̶ ̶p̶e̶r̶i̶o̶d̶.̶ ̶T̶h̶e̶ ̶h̶a̶r̶v̶e̶s̶t̶i̶n̶g̶ ̶i̶n̶t̶e̶n̶s̶i̶t̶i̶e̶s̶ ̶a̶n̶d̶ ̶r̶o̶t̶a̶t̶i̶o̶n̶ ̶l̶e̶n̶g̶t̶h̶s̶ ̶w̶e̶r̶e̶ ̶s̶e̶t̶ ̶s̶p̶e̶c̶i̶f̶i̶c̶a̶l̶l̶y̶ ̶f̶o̶r̶ ̶e̶a̶c̶h̶ ̶r̶e̶g̶i̶o̶n̶ ̶f̶o̶l̶l̶o̶w̶i̶n̶g̶ ̶r̶e̶c̶o̶m̶m̶e̶n̶d̶a̶t̶i̶o̶n̶s̶ ̶f̶r̶o̶m̶ ̶S̶L̶U̶ ̶(̶2̶0̶1̶2̶)̶.̶ ̶T̶h̶e̶ ̶m̶a̶n̶a̶g̶e̶m̶e̶n̶t̶ ̶o̶f̶ ̶f̶o̶r̶e̶s̶t̶ ̶s̶t̶a̶n̶d̶s̶ ̶r̶a̶n̶g̶e̶d̶ ̶f̶r̶o̶m̶ ̶t̶w̶o̶ ̶t̶h̶i̶n̶n̶i̶n̶g̶s̶ ̶d̶u̶r̶i̶n̶g̶ ̶a̶ ̶r̶o̶t̶a̶t̶i̶o̶n̶ ̶p̶e̶r̶i̶o̶d̶ ̶o̶f̶ ̶1̶2̶0̶ ̶y̶e̶a̶r̶s̶ ̶f̶o̶r̶ ̶t̶h̶e̶ ̶l̶e̶a̶s̶t̶ ̶p̶r̶o̶d̶u̶c̶t̶i̶v̶e̶ ̶s̶t̶a̶n̶d̶s̶ ̶i̶n̶ ̶n̶o̶r̶t̶h̶e̶r̶n̶ ̶r̶e̶g̶i̶o̶n̶s̶ ̶t̶o̶ ̶f̶o̶u̶r̶ ̶t̶h̶i̶n̶n̶i̶n̶g̶s̶ ̶d̶u̶r̶i̶n̶g̶ ̶a̶ ̶r̶o̶t̶a̶t̶i̶o̶n̶ ̶p̶e̶r̶i̶o̶d̶ ̶o̶f̶ ̶7̶0̶ ̶y̶e̶a̶r̶s̶ ̶i̶n̶ ̶t̶h̶e̶ ̶m̶o̶s̶t̶ ̶p̶r̶o̶d̶u̶c̶t̶i̶v̶e̶ ̶s̶t̶a̶n̶d̶s̶ ̶i̶n̶ ̶s̶o̶u̶t̶h̶e̶r̶n̶ ̶S̶w̶e̶d̶e̶n̶ ̶(̶T̶a̶b̶l̶e̶ ̶1̶)̶.̶ ̶A̶c̶c̶o̶r̶d̶i̶n̶g̶ ̶t̶o̶ ̶g̶e̶n̶e̶r̶a̶l̶ ̶f̶o̶r̶e̶s̶t̶ ̶m̶a̶n̶a̶g̶e̶m̶e̶n̶t̶ ̶g̶u̶i̶d̶e̶l̶i̶n̶e̶s̶,̶ ̶i̶t̶ ̶w̶a̶s̶

assumed that during thinnings 20% of the stem is removed while 5% transforms into litter (Swedish Forest Agency, 2005). For leaves and roots, it was assumed that 25% transforms into litter. For all of the regions, one clearance – during which 60% of the stands is removed - was applied at the end of the first year after spin-up. During final felling, 5% of trees are remained intact, 90% of the stem is removed and 5% becomes litter, while it is assumed that 95% of all other plant components become litter. The surface cover parameters and litterfall rates of understory vegetation were modified from Svensson et al. (2008) to achieve a more realistic understory in those regions (Table 2).

Historical weather data were derived from the nearby SMHI weather station data through spatial interpolation for each region. Projections of future weather data were generated by the climate change and environmental objective (CLEO) project, using ECHAM5 projections and bias correction of regional climatic data (Personal communication to Thomas Bosshard, SMHI). N-concentration in the deposition was kept constant for each region throughout the simulation as in Svensson et al. (2008) and Gärdenäs et al. (2003).

Literature data, both soil and biomass, were compiled from sites with coniferous forests on Podzols soil within the major moisture classes (mesic and moist), according to the Swedish National Forest Soil Inventory (NFSI) (Olsson et al. 2009, Stendahl et al., 2010.) The corresponding forest biomass data were based on measured standing stock volumes of different age classes presented in the Swedish Forest Inventory (SFI) data (SLU, 2003), for more details, see Svensson et al. (2008).

Part of the model design and setup such as soil physical properties, soil depth (1 m), initial soil C content and C/N ratio followed what was reported by Svensson et al. (2008), who in turn had NSFI as the main source. He et al. (2018) additionally described explicit mycorrhizal fungi settings. The following section will only chronicle the setup for the newly developed P model, as well as describes parts of the model that differed from the aforementioned studies (Svensson et al., 2008, He et al., 2018).

The P content in soil organic matter, i.e., soil organic matter C/P ratios, was based on measurements performed at the sites during a Swedish Throughfall Monitoring Network (SWETHTRO) project (Pihl Karlsson et al., 2015). Only the organic C/P ratio at the O horizon was measured at most sites. Thus, in our calculations of the total stock of soil organic P in the soil profile, we assumed that the C/P ratio measured for the O horizon also extends to the other horizons (uncertainties associated with this assumption will be assessed by including a range in the soil N/P ratios, 10-25, in the sensitivity analysis). The initial labile $P_i$ concentrations were set according to previous data from similar Swedish forest sites (Kronnäs et al., 2019; Fransson

and Bergkvist, 2000). Soil pH was set according to the NSFI data. Initial soil organic P pools were partitioned between soil litter (5%) and humus pools (95%) analog to N partition in Svensson et al. (2008), and decreased exponentially with depth (Fransson and Bergkvist, 2000). The sensitivity of plant growth, soil C and leaching loss responses to soil N and P availability was assessed by varying the soil N/P ratio from 10 to 25 for the study regions (see Table 2 and Figure 5). These ranges were set according to previously published Swedish forest soil data (Lagerström et al., 2009; Giesler et al., 2002; Kronnäs et al., 2019). Previous modelling studies (Eckersten et al., 1995; He et al., 2018) have reported that the parameter 'fungal organic N uptake rate' strongly affects N availability. For this reason, both fungal N and P uptake rates were included in the sensitivity analysis so that we could determine how fungal N and P uptake influence the response to soil N and P availability between soil N/P ratio 10 to 25 (Table 2). The range between the regional lowest and highest values of fungal organic uptake rates for the four regions was used for the sensitivity analysis (Table 2). The newly introduced parameters of P processes were mostly based on values from the literature (Table 3). For instance, the optimal leaf C/P ratios for forest growPrevious order is more logicth, C/P ratios of individual plant components were obtained from empirical measurements from Swedish forests (e.g. Thelin et al., (1998; 2002)). The weathering and surface runoff parameters were defined according to laboratory empirical data (e.g. Guidry and Machenzie, (2000)). The fungi related parameters were mainly obtained from the previous CoupModel calibrations for the same regions (He et al., 2018).

**43.3 2 Datasets for model evaluation**

Literature data of tree biomass, leaf nutrient content as well as water flow and P leaching, both soil and biomass, were compiled from sites with coniferous forests on Podzols soil within the major moisture classes (mesic and moist), according to the Swedish National Forest Soil Inventory (NFSI) (Olsson et al. 2009, Stendahl et al., 2010.) The corresponding forest biomass data were based on measured standing stock volumes of different age classes presented in the Swedish Forest Inventory (SFI) data (SLU, 2003), for more details, see Svensson et al. (2008). Data used to evaluate the newly developed model include SFI biomass values from forest stands aged 10 to 100 years (SLU, 2003).

The measured leaf nutrient data used in the evaluation were obtained from managed forests sites from Swedish Forest Agencywithin the SWETHTRO project (Pihl Karlsson et al., 2011; Pihl Karlsson et al., 2015)The measured leaf nutrient data used in the evaluation were obtained from managed forests sites within the forest monitoring sites of the Swedish Forest Agency

(Wijk, 1997; Akselsson et al. 2015) in the studied regions (some forest sites are also part of the ICP FOREST LEVEL II monitoring program, www.icp-forests.org). Data used in the North 64°N region include two Scots pine stand sites, Gransjö (N 64°30, E17°24 ) and Brattfors (N64°29', E18°28'). For the 61°N region, two sites with Scots pine stands - Kansbo (N61°7', E 14°21') and Furudalsbruk (N61°12', E15°11') were used. Data describing the Fagerhult (N57°30', E15°20') site, dominated by Norway spruce, and the Gynge Scots pine stand (N57°52', E14°44') were used in the 57°N region. Three sites, including a Scots pine stand in Bjäärsgård (N56°10', E13°8'), a Norway spruce stand in Västra Torup (N56°8', E13°30') and a European Beech stand in Kampholma (N56°6', E13°30'), represented the 56°N region.

To compare model outputs with measured P leaching, PO$_4$ and total P data in stream water were obtained from the open database of environmental monitoring data (MVM, https://miljodata.slu.se/mvm/). Thus, the measured P leaching also contains the P leaching from upstream. DOP data were not available measured for the regions. thus iThus the so call measured DOP t was calculated as the difference between the measured total P and PO$_4$. Thus This means the "measured DOP" may contain boths our simulated fractions DOP andbut also particular phosphorus. We used measured water outflow rates from the regional outlet from the Swedish Meteorological and Hydrological Institute (SMHI, https://vattenwebb.smhi.se/station/) to convert the concentrations into fluxes.

**3.3 Model design and setup**

The results were based on simulated forest development withof managed forests was simulated in daily resolution over a rotation period from stand age elass 10 years until 10 years after final harvesting. The 10 years after final harvest were included to cover the potential nutrient leaching during the regeneration phase as in Gärdenäs et al. (2003). The trees in all regions were assumed to be planted in 1961, and thus, the period 1961 to 1970 was used as represented the a spin-up period. The harvesting intensities and rotation lengths were specifiedset specifically for each region following recommendations from SLU (2012). The simulated rotational period was 120, 110, 90 and 70 years from the Nnorth to Ssouth region, respectively. Two thinnings were conducted infor the two northern region andwith three thinnings infor the two southern regions (Table 1). FollowingAccording to general forest management guidelines, it was assumed that during thinnings 20% of the stem is removed while 5% transforms into litter (Swedish Forest Agency, 2005). For leaves and roots, it was assumed that 25% transforms into litter. For all of the regions, one clearance – during which 96060% of the stands is removed - was applied at the end of the stand age year 10 after spin-up. During final felling, 5% of trees

are remained intact, and it is assumed that 90% of the stem is harvested and 5% becomes litter, while all leaves and roots become litter.

**3.4 Model forcing, initial and boundary conditions**

Historical weather data were derived from the nearby SMHI weather station data through spatial interpolation for each region. Projections of future weather data were generated by the climate change and environmental objective (CLEO) project, using ECHAM5 projections and bias correction of regional climatic data (Personal communication with Thomas Bosshard, SMHI). For the P deposition, the measured P deposition rate from each region (Table 1) was kept constant over the simulation period, similar as that for N deposition.

An 11.3-meter deep soil profile of 20 layers were simulated for all the four regions as that in Svensson et al. (2008). An assumed constant heat flow was used to define the lower boundary condition for heat and no water flow was assumed at the bottom soil layer. Part of the model setup and initial conditions such as soil physical properties, drainage, initial soil C content and C/N ratio followed what was reported by Svensson et al. (2008), who in turn had NSFI as the main source. He et al. (2018) additionally described explicit mycorrhizal fungi settings. The following section will only describe the initial conditions for the newly developed P model.

The two vegetation layers were initialized as bare ground with a small amount of C, N and P mass in seedling to start vegetation growth. Initial conditions for solid mineral P content, soil organic matter content, soil stoichiometry were reported in Table 1. Initial soil organic P pools (Table 1) were partitioned between soil litter (5%) and humus pools (95%) analog to N partition in Svensson et al. (2008), and total amount of soil organic P decreased exponentially with depth (Fransson and Bergkvist, 2000). Litter was assumed to be distributed down to 0.5 m while humus down to 1 m depth. The initial labile $P_i$ concentrations were set according to previous data from similar Swedish forest sites (Kronnäs et al., 2019; Fransson and Bergkvist, 2000). Soil pH was set according to the NSFI data and kept constant over the simulation period (Table 1). The initial value of soil organic P for the soil profile was estimated by the available measurements of soil organic matter N/P ratios, performed at the same forest monitoring sites of the Swedish Forest Agency (Wijk, 1995; Akselsson et al. 2015) where leaf nutrient content were sampled.

. However, only the organic N/P ratio at the O horizon was measured at most sites. Thus, in our calculations of the total stock of soil organic P, we assumed that the mean N/P ratio measured for the O horizon also extends to the other horizons in the default model run. Model uncertainties associated with this assumption were assessed

by including a range in the soil N/P ratios, 10-25, in the sensitivity analysis (e.g. Fig5).

**3.5 Sensitivity analysis**

The C and N parameters for these regions in CoupModel were previously tested and calibrated in a number of studies (Svensson et al., 2008) and those of fungi by (He et al., 2018) (Table 2). The surface cover parameters and litterfall rates of understory vegetation were modified from Svensson et al. (2008) to achieve a more realistic understory dynamics in those regions (Table 3).  Most of the newly introduced P parameters were derived from literature (Table 2, 3) . For instance, the optimal leaf C/P ratios for forest growth, C/P ratios of individual plant components were obtained from empirical measurements from Swedish forests (e.g. Thelin et al., (1998; 2002)). The weathering and surface runoff parameters were defined according to laboratory empirical data (e.g. Guidry and Machenzie, (2000)).

We conducted a global sensitivity analysis of the new Coup-CNP model to its parameterization (n=34) using a Monte-Carlo based sensitivity analysis method to assess the stability and robustness of the model with respect to its parameter values. The sampled parameters and their ranges (Table S.1), model design and global sensitivity results (Table S.2, S.3, S.4) were reported in detail in the supplement. Based on these simulations and parameter sensitivity rankings, we select three most important parameters (n=3), which has a strong effect on the model outcome to further form a new set of model runs, which are used for the model sensitivity analysis presented in this paper. The selected three parameters are initial soil humus P, short-cut N uptake rate and short-cut P uptake rate (Table S.2, S.3), all strongly regulates the soil N and P availability. The sensitivity of plant growth, soil C and leaching loss responses to soil N and P availability was then assessed by varying the soil N/P ratio from 10 to 25 for the study regions (see Table 2 and Figure 5). These ranges were set according to previously published Swedish forest soil data (Lagerström et al., 2009; Giesler et al., 2002; Kronnäs et al., 2019) and the additional soil P data from Swedish Forestry Agency inventory (Table 1). The ranges of

short-cut uptake coefficient for N and P were based on the regional lowest and highest values of short-cut uptake rates for the four regions (Table 2).

~~The sensitivity of plant growth, soil C and leaching loss responses to soil N and P availability was then assessed by varying the soil N/P ratio from 10 to 25 for the study regions (see Table 2 and Figure 5). These ranges were set according to previously published Swedish forest soil data (Lagerström et al., 2009; Giesler et al., 2002; Kronnäs et al., 2019) and the additional soil P data from Swedish Forestry Agency inventory (Table 1). Furthermore previous modelling studies (Eckersten and Beier, 1998; He et al., 2018) have reported that the parameter 'fungal short-cut N uptake rate' strongly affects N availability. High sensitivity of this parameter was again shown in our comprehensive parameter sensitivity analysis (Fig. S1, S.3 in the supplementary). Thus, both fungal N and P uptake rates were included in the sensitivity analysis so that we could determine how fungal N and P uptake influence the response to soil N and P availability between soil N/P ratio 10 to 25 (Table 2). The range of fungal N and P uptake rates was based on the regional lowest and highest values of fungal short-cut uptake rates for the four regions (Table 2).~~

 ~~The sensitivity of plant growth, soil C and leaching loss responses to soil N and P availability was then assessed by varying the soil N/P ratio from 10 to 25 for the study regions (see Table 2 and Figure 5). These ranges were set according to previously published Swedish forest soil data (Lagerström et al., 2009; Giesler et al., 2002; Kronnäs et al., 2019) and the additional soil P data from Swedish Forestry Agency inventory (Table 1). Previous modelling studies (Eckersten and Beier, 1998; He et al., 2018) have reported that the parameter 'fungal short-cut N uptake rate' strongly affects N availability. High sensitivity of this parameter was again shown in our comprehensive parameter sensitivity analysis (Fig. S1, S.3 in the supplementary). Thus, both fungal N and P uptake rates were included in the sensitivity analysis so that we could determine how fungal N and P uptake influence the response to soil N and P availability between soil N/P ratio 10 to 25 (Table 2). The range between the regional lowest and highest values of fungal short-cut uptake rates for the four regions was used for the sensitivity analysis (Table 2).~~

**4 Results**

**4.1 Model assessment**

[revised manuscript text omitted]

---

## Author Response (AR2)

Topical Editor Decision: Publish subject to technical corrections (12 Dec 2020) by Carlos Sierra

Comments to the Author:

Dear authors,

Your revised version of the manuscript addresses well most comments from the reviewers. I appreciate your effort in preparing a global sensitivity analysis, and editing your manuscript with a professional proof-reading service. This new version reads much better than the previous versions. As pointed out by the reviewers, the model makes a number of assumptions that may need additional support from empirical studies. It is also highly parameterized, and from this version it is hard to know what would be the effect of parameter uncertainty in predictions.

I also have reservations on your use of a simple Euler scheme to solve the system of partial differential equations. It is well known that this method has serious issues with numerical stability and large approximation errors.

From my point of view, these are important issues of the current implementation of your model. However, GMD is a scientific forum for the description of models, and we are open to publish model descriptions that may be re-evaluated and improved in the future. So I will accept your current model description, but I would appreciate if you make two modifications to the current version:

- Please add a paragraph at the end of your Discussion identifying potential limitations and areas of improvement in your model. For example, the high number of parameters is one potential limitation for the use of this model in other ecosystems, and it is an issue that may translate in high uncertainties in predictions. Other implicit assumptions identified by the reviewers could also be mentioned here.

- Avoid very definite statements in the Abstract and the Conclusions about the 'need for biogeochemical models to comprehensively consider the P cycle'. Although, we all would agree that the P cycle is very important for global scale biogeochemistry, your study does not address directly this topic. You do not provide evidence that a global scale model would not be able to predict effects of global change on C sequestration globally. You have some interesting results for a gradient in Sweden, but this is far from a modeling study that compares models with and without P cycle at the global scale. So, please re-phrase or remove these definitive statements from your manuscript.

Best regards,

Carlos A. Sierra

The following revisions has been made:

**Abstract**:

Current version: "Simulations from the new Coup-CNP model provide strong evidence that P fluxes need to be further considered in ecosystem studies of how C turnover react to climate change. We conclude that biogeochemical models should include the P cycle to reliably assess how both climate change and N deposition will affect C sequestration and nutrient leaching over long time periods."

- Revised to: "The simulations showed that Coup-CNP can describe shifting from being most N to most P limited and vice versa. The potential P-limitation of terrestrial ecosystems highlights the need for biogeochemical ecosystem models to consider the P cycle. We conclude that the inclusion of the P cycle enabled the COUP-CNP to account for various

feedback mechanisms that have a significant impact on ecosystem C sequestration and leaching under climate change and/or elevated N deposition"

**Conclusion (section 6)**

Current version: "We conclude that the potential P-limitation of terrestrial ecosystems highlights the need for biogeochemical ecosystem models to comprehensively consider the P cycle. The inclusion of the P cycle will enable a model to account for various feedback mechanisms that have a significant impact on C sequestration and N leaching under climate change and/or elevated N deposition".

- Revised to: "The simulations showed that Coup-CNP is able to reproduce shifting from being most N to most P limited and vice versa during a rotation period. We conclude that the potential P-limitation of terrestrial ecosystems highlights the need for biogeochemical ecosystem models to consider the P cycle. The inclusion of the P cycle enabled the COUP-CNP to account for various feedback mechanisms that have a significant impact on ecosystem C sequestration and N leaching under climate change and/or elevated N deposition".

Add the following model limitation and future research was added at the end of discussions:

"This paper presents the newest version of the Coup-CNP model. The evaluation data from this study offer a partial picture of the entire P cycle, and further validation should focus on the internal P fluxes and its interaction with C and N. As such, the global sensitivity analysis presented here provides an example for future use of the model. A user can choose which modules to include depending on the specific research question. The Coup-CNP was evaluated using data of Swedish forest ecosystems and model results suggest soil organic matter C/P ratio is a good indicator of plant P availability. However, in the long run, the weathering rate provides the ultimate source for P. Most of the P model concepts builds on well-established concepts. However, there are few model assumptions and parameters, which would benefit from further research and more experimental evidence to test and evaluate its more general validity. Our results show the importance of the P short-cut uptake to sustain the forest growth and thereby highlighting the role of microbes. The plant P availability is regulated by the competition between mineral P uptake, short-cut P uptake, and soil adsorption. Coup-CNP simulates such competition by different coefficients or parameters, which are largely unconstrained by observations. Similarly, while a plant-mycorrhizal symbiosis interaction scheme is suggested, it relies on several parameters or coefficients, which are largely unconstrained by independent observations. We recommend further testing of the model for agricultural, wetland and other ecosystems with a wide range of plant P availability to reduce uncertainties in the model outputs. For example, tropical ecosystems that are known to be P-limited.